# Unlocking Global Optimality in Bilevel Optimization: A Pilot Study

**Quan Xiao**
Rensselaer Polytechnic Institute
Troy, NY 12180, United States
quanx1808@gmail.com

**Tianyi Chen**
Rensselaer Polytechnic Institute
Troy, NY 12180, United States
chentianyi19@gmail.com

## Abstract

Bilevel optimization has witnessed a resurgence of interest, driven by its critical role in trustworthy and efficient AI applications. Recent focus has been on finding efficient methods with provable convergence guarantees. However, while many prior works have established convergence to stationary points or local minima, obtaining the global optimum of bilevel optimization remains *an important yet open problem*. The difficulty lies in the fact that unlike many prior non-convex single-level problems, bilevel problems often do not admit a "benign" landscape, and may indeed have multiple spurious local solutions. Nevertheless, attaining the global optimality is indispensable for ensuring reliability, safety, and cost-effectiveness, particularly in high-stakes engineering applications that rely on bilevel optimization. In this paper, we first explore the challenges of establishing a global convergence theory for bilevel optimization, and present two sufficient conditions for global convergence. We provide *algorithm-dependent* proofs to rigorously substantiate these sufficient conditions on two specific bilevel learning scenarios: representation learning and data hypercleaning (a.k.a. reweighting). Experiments corroborate the theoretical findings, demonstrating convergence to global minimum in both cases.

## 1 Introduction

Bilevel optimization aims to handle two interconnected problems, where one problem is nested within another (Bracken & McGill, 1973). Recently, bilevel optimization has gained significant attention due to their relevance in various machine learning, signal processing and wireless communication applications, including hyperparameter optimization (Maclaurin et al., 2015; Franceschi et al., 2017; 2018; Pedregosa, 2016), meta-learning (Finn et al., 2017), representation learning (Arora et al., 2020), reinforcement learning with human feedback (Stadie et al., 2020; Shen et al., 2024), continual learning (Pham et al., 2021; Borsos et al., 2020; Hao et al., 2023), adversarial learning (Zhang et al., 2022; Robey et al., 2024) and neural architecture search (Liu et al., 2019); see recent survey (Liu et al., 2021a; Sinha et al., 2017). In this paper, we focus on the *optimistic bilevel optimization* problem as

$$\min_{u,v \in \mathcal{S}(u)} f(u,v), \quad \text{s.t.} \quad \mathcal{S}(u) = \arg\min_v g(u,v) \tag{1}$$

where both the upper-level objective function $f : \mathbb{R}^{d_1} \times \mathbb{R}^{d_2} \to \mathbb{R}$ and the lower-level objective function $g : \mathbb{R}^{d_1} \times \mathbb{R}^{d_2} \to \mathbb{R}$ are continuously differentiable.

To tackle the above bilevel problems, various efficient algorithms have been proposed with rigorous guarantees on iteration and sample complexity, but most of them are only guaranteed to converge to the stationary points (Ji et al., 2021; Chen et al., 2021; Hong et al., 2023; Dagréou et al., 2022; Ghadimi & Wang, 2018; Kwon et al., 2023) or locally optimal solution (Huang et al., 2022; Dempe, 2019; Chen et al., 2023a) instead of the global optima. However, identifying the global optimal solution for bilevel optimization is vital in various real-world applications where the quality of solutions can have significant impacts. For instance, in policy-making (Dempe et al., 2019), energy systems (Wu et al., 2019; Razmara et al., 2016), resource allocation (Gao et al., 2020; Huang et al., 2019; Shi & Luo, 2017) and network design (Gao et al., 2005), globally optimal solutions can lead to more cost-efficient and sustainable outcomes than local solutions. Furthermore, in high-stakes fields like healthcare, law, and robotics, attaining global optima ensures that the AI models are aligned with

human values with minimal risks of harmful generation (Modares et al., 2015; Bıyık et al., 2022). Thus, the *goal of this paper* is to study the global convergence (in contrast to convergence to local optima) of bilevel optimization for certain (not all) machine learning applications.

## 1.1 OUR MAIN RESULTS

We summarize our main results to tackle the global optimality of bilevel optimization.

### C1) The penalty reformulation of bilevel optimization has a more benign landscape.

Recent advances in bilevel optimization algorithms can be generally classified into two categories.

**Nested approaches** solve the problem (1) from its nested formulation $\mathsf{F}(u) := \min_{v \in \mathcal{S}(u)} f(u, v)$ by optimizing first over one variable $v$ and then over the order $u$; see e.g., (Ghadimi & Wang, 2018; Ji et al., 2021; Hong et al., 2023; Chen et al., 2021).

**Constrained approaches** incorporate the optimality condition of the lower-level problem in (1) as a constraint in the upper-level problem and then optimize over $u$ and $v$ jointly; see e.g., (Sow et al., 2022a; Liu et al., 2022; Kwon et al., 2023; 2024; Chen et al., 2023a; Shen et al., 2023).

In this paper, we first investigate the loss landscape of the bilevel problem (1) through the lens of *nested* and *constrained* bilevel reformulations, demonstrating that the constrained formulation is easier to yield a *benign landscape*.

### C2) Benign properties of the penalty reformulation ensure convergence to global optimum.

We analyze the landscape of the penalty reformulation of the constrained version of (1), and derive two sufficient conditions that ensure a favorable landscape for bilevel problems. These conditions, inspired by practical applications, provide a stepping stone for global convergence of bilevel algorithms.

Specifically, we define the penalized objective as $\mathsf{L}_\gamma(u, v) := f(u, v) + \gamma(g(u, v) - g^*(u))$, where $g^*(u) := \min_v g(u, v)$ is the value function and $\gamma > 0$ is a penalty constant, and generalize the definition of the standard Polyak-Lojasiewicz (PL) condition to the two-variable case below.

**Definition 1** (Joint and blockwise PL condition). $\mathsf{L}_\gamma(u, v)$ *satisfies **joint PL** with* $\mu_l = \mathcal{O}(\gamma)$ *if*

$$\|\nabla \mathsf{L}_\gamma(u, v)\|^2 \geq 2\mu_l(\mathsf{L}_\gamma(u, v) - \min_{u,v} \mathsf{L}_\gamma(u, v)). \tag{2}$$

*In addition,* $\mathsf{L}_\gamma(u, v)$ *is said to be **blockwise PL** with* $\mu_u, \mu_v = \mathcal{O}(\gamma)$ *if both of the followings hold*

$$\|\nabla_u \mathsf{L}_\gamma(u, v)\|^2 \geq 2\mu_u(\mathsf{L}_\gamma(u, v) - \min_u \mathsf{L}_\gamma(u, v)) \tag{3a}$$

$$\|\nabla_v \mathsf{L}_\gamma(u, v)\|^2 \geq 2\mu_v(\mathsf{L}_\gamma(u, v) - \min_v \mathsf{L}_\gamma(u, v)). \tag{3b}$$

We use the term 'benign landscape' to represent penalty function $\mathsf{L}_\gamma(u, v)$ satisfying either of the above conditions, or nested objective $\mathsf{F}(u) = \min_{v \in \mathcal{S}(u)} f(u, v)$ satisfying PL condition over $u$.

**From benign landscape to global convergence.** Under either of the above conditions, we establish that the penalized bilevel gradient descent (PBGD) algorithm (Kwon et al., 2023; Shen et al., 2023; Kwon et al., 2024; Chen et al., 2023a), a fully first-order method, globally converges to the optimal solutions of (1). Under the joint PL condition, updating $(u, v)$ in a Jacobi manner (c.f. (7b)–(7c)) will ensure the global convergence. Under the blockwise PL condition, updating $u$ and $v$ in a Gauss-Seidel manner (c.f. (7d)–(7e)) ensures the global convergence.

### C3) Two representative bilevel learning problems guarantee benign properties.

We validate two benign landscape conditions in Definition 1 through two representative bilevel applications: *representation learning* and *data hyper-cleaning* (a.k.a. reweighting) with a least squares loss, respectively. We rigorously prove that the loss surfaces reached by PBGD, when employing a Jacobi update and a Gauss-Seidel update, respectively, adhere to the joint PL condition and the blockwise PL condition throughout the optimization trajectory for these two problems. As a result, we prove for *the first time* that PBGD converges to the *global optimum* for both problems.

**Choice of problems and models.** The particular choice of two bilevel applications exemplify two different types of bilevel interactions between the upper and lower-level problems, are suited to different update dynamics of PBGD. Compared with the landscape analysis of single-level problems,

the uniqueness of the bilevel landscape lies in the intricate coupling structures between the upper-level and lower-level problems. Recognizing this, it is evident that even analyzing the *linear models* can capture the essence of the problem structure and exclude other confounding factors which might be less relevant to the bilevel landscape analysis. Moreover, as a challenging yet important problem, the global convergence analysis for single-level optimization also starts with linear models to gain insights such as in matrix completion (Sun & Luo, 2016; Ye & Du, 2021), phrase retrieval (Ma et al., 2018), and linear neural networks (Xu et al., 2023; Zou et al., 2020). Therefore, our efforts will be put into understanding how coupling structure affects the bilevel landscape with a linear model.

## 1.2 RELATED WORKS

**Stationary point and local convergence.** The recent interest in developing efficient gradient-based bilevel methods with nonasymptotic convergence guarantees has been stimulated by (Ghadimi & Wang, 2018; Ji et al., 2021; Hong et al., 2023; Chen et al., 2021). Based on different Hessian inversion approximation techniques, these algorithms can be categorized into iterative differentiation (Franceschi et al., 2017; 2018; Grazzi et al., 2020) and implicit differentiation-based approaches (Chen et al., 2021; Ghadimi & Wang, 2018; Hong et al., 2023; Ji et al., 2021; Pedregosa, 2016). Recent works have reformulated the bilevel optimization problem as a single-level constrained problem, and solved it via the penalty-based gradient method (Shen et al., 2023); see also (Liu et al., 2022; Kwon et al., 2023; 2024; Chen et al., 2023a; Lu & Mei, 2023). While the nonasymptotic analysis of stationary convergence has been extensively studied in bilevel optimization recently, finite-time convergence guarantee for local minimum remains under-explored. Recently, (Huang et al., 2022) and (Chen et al., 2023a) have found the benefit of adding noise to gradient-based bilevel methods, which helps them to efficiently escape from saddle points and converge to local minima. However, none of them analyze the landscape and the convergence to global optimal solutions.

**Global optimum convergence.** In general, finding the global optimal solution for bilevel optimization is *NP-hard* (Vicente et al., 1994). Historically, globally convergent algorithms for bilevel optimization (Gümüş & Floudas, 2001; Muu & Quy, 2003) were built upon the branch and bound method, a globally convergent single-level algorithm. Despite its theoretical soundness, the branch and bound method is generally inefficient due to its exhaustive search nature. Another line of research focused on the bilevel problems with specific structures. A semi-definite relaxation method has been introduced in (Jeyakumar et al., 2016) for polynomial bilevel problems, and a dual reformulation has been developed in (Wang et al., 2007) for quadratic bilevel problems with a linear lower-level. More recent efforts by (Wang et al., 2021; 2022) solved Stackelberg prediction games with quadratic regularized least squares problems at the lower level by applying semi-definite relaxation and spherical constraints. However, these methods are problem-specific and are not suitable for more complex machine-learning scenarios, particularly where the lower-level problem presents multiple solutions.

## 1.3 NOVELTY AND TECHNICAL CHALLENGES

We highlight the novelty and technical challenges for the analysis in our work as follows.

T1) We carefully examine the challenges posed by the complex landscape of nested optimization and the general non-additivity of PL functions (see Examples 1–5), underscoring the importance of analyzing the landscape of the penalized problem and the additivity properties of specific PL functions. Even for the special case highlighted in Observation 2, the additivity remains non-trivial due to differences in the strongly convex and matrix mappings.

T2) Global convergence under the generic benign landscape conditions in Definition 1 is still insufficient for the two applications, even in the linear model case. This is because only local PL and smoothness conditions are satisfied, with constants that vary along the optimization trajectory of PBGD. By leveraging an induction-based proof and the advanced acute matrix perturbation theory, we establish the boundedness of the local PL and smoothness constants throughout the trajectory of PBGD, and thus this leads to the global convergence results over the penalized objective $\mathsf{L}_\gamma(u, v)$.

T3) To bridge the gap in achieving global convergence to the optimal solution of the original bilevel problem for these two applications, we establish application-specific approximate equivalence between the penalized problem and the original problem, relying solely on local PL and local smoothness conditions, which has not yet been explored in the existing literature. The conditions are also validated along the optimization trajectory of PBGD.

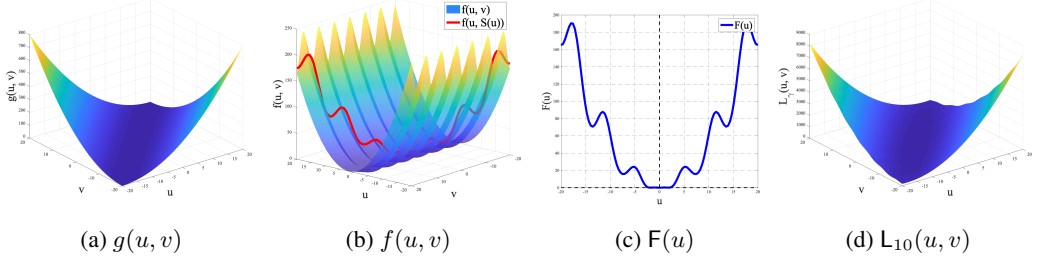

(a) $g(u, v)$      (b) $f(u, v)$      (c) $\mathsf{F}(u)$      (d) $\mathsf{L}_{10}(u, v)$

Figure 1: Visualization of $g(u, v)$, $f(u, v)$, $\mathsf{F}(u)$ and $\mathsf{L}_\gamma(u, v)$ in Example 1. In (b), $f(u, v)$ is PL but $F(u)$ is distorted by $\mathcal{S}(u)$. In (c), saddle points exist for $\mathsf{F}(u)$, suggesting that $\mathsf{F}(u)$ is not PL. In (d), the penalty objective $\mathsf{L}_\gamma(u, v)$ has better landscape because of additional dimension of $v$.

**Notation.** Let $\mathbb{R}, \mathbb{R}_{\geq 0}, \mathbb{R}_{>0}, \mathbb{Z}$ be the sets of real, nonnegative real, positive real and integer numbers. For a given matrix $A \in \mathbb{R}^{p \times q}$, let $\lambda_i(A)$ and $\sigma_i(A)$ be the $i$-th eigenvalue and singular value in descending order, respectively. Denote $\sigma_{\max}(A) = \sigma_1(A)$, $\sigma_{\min}(A) = \sigma_{\min\{p,q\}}(A)$ and $\sigma_*(A) = \sigma_{\max\{i|\sigma_i(A)>0\}}$ as the maximal, minimal, and minimal nonzero singular values. Let $A_{ij}$ represent the entry at the $i$-th row and $j$-th column of $A$. Let $\|A\|$ and $\|A\|_2$ be the Frobenius and spectral norms of $A$. Let $\theta_{>0}$ be $\theta$ if $\theta > 0$ and infinity otherwise. Let $\mathbb{1}(\cdot)$ denote the indicator function.

## 2   CHALLENGES AND TARGET OF CONVERGENCE

In this section, we will reveal the complicated landscape of the nested objective $\mathsf{F}(u)$ and then state the target of global convergence.

### 2.1   CHALLENGES IN THE NESTED FORMULATION OF BILEVEL OPTIMIZATION

Establishing the global convergence for bilevel optimization algorithms is fundamentally challenging because the nested bilevel objective $\mathsf{F}(u)$ exhibits different properties compared to the upper- and lower-level objectives $f(u, v)$ and $g(u, v)$. This is primarily due to the distortion induced by $\mathcal{S}(u)$. To gain some intuition on this distortion, consider the case where both upper and lower-level objectives satisfy the PL condition jointly over $(u, v)$: there exists $\mu_f, \mu_g > 0$ such that

$$\|\nabla f(u, v)\|^2 \geq 2\mu_f(f(u, v) - \min_{u,v} f(u, v)) \text{ and } \|\nabla g(u, v)\|^2 \geq 2\mu_g(g(u, v) - \min_{u,v} g(u, v)). \quad (4)$$

The following example shows that the PL condition on both levels is not sufficient to guarantee the PL condition over the bilevel objective $\mathsf{F}(u)$, even if the lower-level solution mapping $\mathcal{S}(u)$ is linear.

**Example 1.** *With $u \in \mathbb{R}$ and $v \in \mathbb{R}$, consider the following upper and lower-level objectives*

$$f(u, v) = \frac{1}{2}(u - 2\sin(v))^2 \quad and \quad g(u, v) = \frac{1}{2}(u - v)^2.$$

*We can verify that both $f(u, v)$ and $g(u, v)$ satisfy the joint PL condition in (4) and the lower-level problem parameterized by $u$ yields the unique solution $\mathcal{S}(u) = u$. However, the overall bilevel function $\mathsf{F}(u) = f(u, \mathcal{S}(u)) = \frac{1}{2}(u - 2\sin(u))^2$ violates the PL condition over $u$. The graph of $\mathsf{F}(u)$ is shown in Figure 1 and the formal proof is deferred to Appendix C.1.*

Similarly, joint convexity of both levels can not ensure the convexity of the bilevel objective $\mathsf{F}(u)$ even if the lower-level solution is unique; see the Example 4 in Appendix C.3.

These suggest that the landscape of the nested bilevel objective $\mathsf{F}(u)$ can be easily distorted by the lower-level solution mapping $\mathcal{S}(u)$, which we do not have the direct access to control.

### 2.2   SEEKING GLOBAL OPTIMUM VIA PENALTY REFORMULATION

To avoid directly dealing with $\mathcal{S}(u)$, some of recent works have focused on solving the bilevel optimization (1) from the perspective of constrained optimization, where the lower-level problem is treated as a constraint of the upper-level problem; see e.g., (Kwon et al., 2023; Shen et al., 2023; Liu et al., 2022; Mehra & Hamm, 2021; Kwon et al., 2024) . Defining the value function as $g^*(u) = \min_v g(u, v)$, the bilevel problem in (1) can be equivalently reformulated as

$$\min_{u,v} f(u, v), \quad \text{s.t.} \quad g(u, v) - g^*(u) \leq 0. \quad (5)$$

An $(\epsilon_1, \epsilon_2)$ optimal solution of the bilevel problem (1) can then be defined as follows.

**Definition 2** (An $(\epsilon_1, \epsilon_2)$ solution of bilevel problem). *Given $u^* \in \mathbb{R}^{d_1}, v^* \in \mathbb{R}^{d_2}$, we say point $(u^*, v^*)$ is an $(\epsilon_1, \epsilon_2)$ global solution to bilevel problem* (1) *if $g(u^*, v^*) - g^*(u^*) \leq \epsilon_2$ and for any $u \in \mathbb{R}^{d_1}, v \in \mathbb{R}^{d_2}$ satisfying $g(u, v) - g^*(u) \leq \epsilon_2$, we have $f(u^*, v^*) \leq f(u, v) + \epsilon_1$.*

To find an $(\epsilon_1, \epsilon_2)$ solution of (5), one can resort to optimize its penalized problem (Shen et al., 2023)

$$\min_{u,v} \mathsf{L}_\gamma(u, v) := f(u, v) + \gamma(g(u, v) - g^*(u)). \tag{6}$$

It was shown in (Shen et al., 2023) that, the $\epsilon$-solution for (6) with a penalty parameter $\gamma = \mathcal{O}(\epsilon^{-0.5})$ is the $(\epsilon, \mathcal{O}(\epsilon))$ solution for bilevel problem in Definition 2. Therefore, analyzing the landscape of the penalized objective $\mathsf{L}_\gamma(u, v)$ also leads to the global convergence to an $(\epsilon, \epsilon)$ solution of (5).

**Benefits of $\mathsf{L}_\gamma(u, v)$ instead of $\mathsf{F}(u)$.** Analyzing $\mathsf{L}_\gamma(u, v)$ bypasses the need to study the PL-preserving property under composition in Section 2.1, making it easier to establish a benign landscape property; see Figure 1. Besides, the continuity and differentiablity of $g^*(u)$ is generally easier to be satisfied than that of $\mathcal{S}(u)$; see e.g., (Dontchev & Rockafellar, 2014, Example 3B.6). This makes $\mathsf{L}_\gamma(u, v)$ more likely to be differentiable than $\mathsf{F}(u)$. On the other hand, as $\min_v \mathsf{L}_\gamma(u, v)$ is an equivalent-dimensional proxy of $\mathsf{F}(u)$ (Kwon et al., 2024), $\mathsf{L}_\gamma(u, v)$ offers a high dimensional approximation of $\mathsf{F}(u)$, smoothing out the ravine of $\mathsf{F}(u)$ in high-dimensional space; see Figure 1–2.

# 3 GLOBAL CONVERGENCE CONDITION IN BILEVEL OPTIMIZATION

In this section, we will propose counterparts to the global convergence condition from single-level optimization for bilevel optimization based on the penalized constrained formulation (6).

## 3.1 BENIGN LANDSCAPE CONDITIONS

To characterize the global convergence in bilevel optimization, we generalize the PL condition (Karimi et al., 2016), and define the joint PL and blockwise PL conditions of $\mathsf{L}_\gamma(u, v)$ in Definition 1.

**Two "bilevel PL" conditions.** The joint PL condition in (2) extends the standard PL condition to a two-variable setting by treating $(u, v)$ as a new single variable, while the blockwise PL condition in (3) reflects the hierarchical structure of bilevel problems by treating the optimization over $u$ and $v$ as separate blocks. For simplicity, we define condition (3a) for the whole space, but it is also sufficient for global convergence if the condition (3a) holds only for $v \in \arg\min_v \mathsf{L}_\gamma(u, v)$. It is also important to note that the joint PL condition and the blockwise PL condition can not imply each other.

**Rationale of two PL conditions.** The two PL conditions correspond to bilevel problems with isomorphic and heterogeneous levels, respectively. Specifically, in representation learning (Section 4), the upper-level and lower-level variables are model weights from different layers, maintaining a similar nature; and in data hyper-cleaning (Section 5), the lower-level variables are still model weights, but the upper-level variables are the classification parameters for each sample, which is a distinct type of variable from the model weights.

Following (Shen et al., 2023; Kwon et al., 2024), if we choose $\gamma = \mathcal{O}(\epsilon^{-0.5})$, the $\epsilon$-global convergence to the penalized problem (6) implies the convergence to $(\epsilon, \mathcal{O}(\epsilon))$ global solution of the bilevel problem in Definition 2. Therefore, we can then focus on developing a globally convergent gradient-based algorithm for the penalized problem $\mathsf{L}_\gamma(u, v)$ under the joint or blockwise PL condition.

## 3.2 A GLOBALLY CONVERGENT ALGORITHM: PENALTY-BASED BILEVEL GRADIENT DESCENT

We revisit the penalty-based bilevel gradient descent (PBGD) algorithm (Shen et al., 2023; Kwon et al., 2023; 2024; Chen et al., 2023a), a fully first-order method with provable stationary convergence. We will demonstrate that PBGD reaches the global optimum when the benign conditions are satisfied.

As the name implies, PBGD employs gradient descent on the penalized objective (6). Thanks to the Danskin type theorem (Nouiehed et al., 2019), $\nabla \mathsf{L}_\gamma(u, v)$ can be calculated by

$$\nabla_u \mathsf{L}_\gamma(u, v) = \nabla_u f(u, v) + \gamma(\nabla_u g(u, v) - \nabla_u g(u, w)), \ \nabla_v \mathsf{L}_\gamma(u, v) = \nabla_v f(u, v) + \gamma \nabla_v g(u, v)$$

where $w \in \mathcal{S}(u)$ is an auxiliary variable used for estimating $\nabla g^*(u)$. Therefore, at iteration $k$, PBGD first updates $w^k$ to track $\mathcal{S}(u^k)$ by $T_k$ step gradient descent on $g(u^k, \cdot)$ initialized by $w^{k,0} = 0$

$$w^{k,t+1} = w^{k,t} - \beta \nabla_v g(u^k, w^{k,t}) \tag{7a}$$

and update $w^{k+1} = w^{k,T_k}$. Then we update $u^k$ and $v^k$ via the gradient descent update on $\mathsf{L}_\gamma(u,v)$ simultaneously that we term the Jacobi version if $\mathsf{L}_\gamma(u,v)$ is jointly PL; that is

$$u^{k+1} = u^k - \alpha\left(\nabla_u f(u^k, v^k) + \gamma\left(\nabla_u g(u^k, v^k) - \nabla_u g(u^k, w^{k+1})\right)\right) \tag{7b}$$

$$v^{k+1} = v^k - \alpha(\nabla_v f(u^k, v^k) + \gamma\nabla_v g(u^k, v^k)) \tag{7c}$$

or alternatingly that we term the Gauss-Seidel version if $\mathsf{L}_\gamma(u,v)$ is blockwise PL (with $v^{k,0} = v^0$ as the initialization and $v^{k+1} = v^{k,T_k}$ as the output); that is

$$v^{k,t+1} = v^{k,t} - \tilde{\beta}(\nabla_v f(u^k, v^{k,t}) + \gamma\nabla_v g(u^k, v^{k,t})), \; t = 1, 2, \cdots, T_k \tag{7d}$$

$$u^{k+1} = u^k - \alpha\left(\nabla_u f(u^k, v^{k+1}) + \gamma\left(\nabla_u g(u^k, v^{k+1}) - \nabla_u g(u^k, w^{k+1})\right)\right). \tag{7e}$$

The Jacobi and Gauss-Seidel versions of PBGD are summarized in Algorithm 1 and 2, respectively.

---

**Algorithm 1** PBGD in Jacobi fashion

1: Initialization $\{u^0, v^0, w^0\}$, stepsizes $\{\alpha, \beta\}$, penalty constants $\gamma$
2: **for** $k = 0$ **to** $K - 1$ **do**
3:     initialize $w^{k,0} = w^0$
4:     **for** $t = 0$ **to** $T_k - 1$ **do**
5:         update $w^{k,t+1}$ by (7a)
6:     **end for**
7:     set $w^{k+1} = w^{k,T_k}$
8:     update $u^{k+1}$ by (7b)
9:     update $v^{k+1}$ by (7c)
10: **end for**

**Algorithm 2** PBGD in Gauss-Seidel fashion

1: Initialization $\{u^0, v^0, w^0\}$, stepsizes $\{\alpha, \beta, \tilde{\beta}\}$, penalty constants $\gamma$
2: **for** $k = 0$ **to** $K - 1$ **do**
3:     **for** $t = 0$ **to** $T - 1$ **do**
4:         update $w^{k,t+1}$ by (7a)
5:     **end for**
6:     **for** $t = 0$ **to** $T - 1$ **do**
7:         update $v^{k,t+1}$ by (7d)
8:     **end for**
9:     update $u^{k+1}$ by (7e)
10: **end for**

---

To prove the convergence, we make the following assumptions; see also in (Shen et al., 2023; Kwon et al., 2024; Ghadimi & Wang, 2018; Hong et al., 2023; Chen et al., 2021; Ji et al., 2021).

**Assumption 1.** *Assume $f(u,v)$ and $g(u,v)$ are $\ell_f$- and $\ell_g$-smooth over $(u,v)$, $f(u,\cdot)$ is $\ell_{f,0}$-Lipschitz continuous over $v$, and $g(u,\cdot)$ is $\mu_g$-PL over $v$.*

The following theorem shows that PBGD is globally convergent to the bilevel problem with almost linear convergence rate when the penalized problem has a benign landscape.

**Theorem 1** (Global convergence of PBGD). *Suppose Assumption 1 holds, then $g^*(u)$ is smooth with $L_g := \ell_g(1 + \ell_g/2\mu_g)$. Given a target accuracy $\epsilon$, we set $\gamma = \mathcal{O}(\epsilon^{-0.5})$, stepsizes $\beta \leq \frac{1}{\ell_g}$, inner loop $T_k = \mathcal{O}\left(\log\left(\gamma^2\epsilon^{-1}\right)\right)$. Then if $\mathsf{L}_\gamma(u,v)$ satisfies the joint PL condition (2) with $\mu_l > 0$ and we choose $\alpha \leq \frac{1}{\ell_f + \gamma(\ell_g + L_g)}$, there exists $\epsilon_\gamma = \mathcal{O}(\epsilon)$ s.t. the iterates of PBGD in Algorithm 1 satisfies*

$$f(u^K, v^K) - f(u, v) \leq \mathcal{O}((1 - \alpha\mu_l)^K) + \mathcal{O}(\epsilon) \quad \text{and} \quad g(u^K, v^K) - \min_v g(u^K, v) \leq \epsilon_\gamma = \mathcal{O}(\epsilon)$$

*for any $(u,v)$ with $g(u,v) - g^*(u) \leq \epsilon_\gamma = \mathcal{O}(\epsilon)$.*

*Alternatively, if $\mathsf{L}_\gamma(u,v)$ satisfies the blockwise PL condition (3) with $\mu_u, \mu_v > 0$, $\arg\min_v \mathsf{L}_\gamma(u,v)$ is independent of $u$ and we choose the stepsizes $\tilde{\beta} \leq \frac{1}{\ell_f + \gamma\ell_g}, \alpha \leq \frac{1}{L_\gamma}$ where $L_\gamma := (\ell_f + \gamma\ell_g)(1 + (\ell_f + \gamma\ell_g)/2\mu_v) + L_g$, then the iterates of PBGD in Algorithm 2 satisfies*

$$f(u^K, v^{K+1}) - f(u, v) \leq \mathcal{O}((1 - \alpha\mu_u)^K) + \mathcal{O}(\epsilon) \quad \text{and} \quad g(u^K, v^{K+1}) - \min_v g(u^K, v) \leq \epsilon_\gamma$$

*for any $(u,v)$ with $g(u,v) - g^*(u) \leq \epsilon_\gamma = \mathcal{O}(\epsilon)$.*

Theorem 1 shows that, with properly selected stepsizes, PBGD employing either the Jacobi update under the joint PL condition or the Gauss-Seidel update under the blockwise PL condition, converges to an $(\mathcal{O}(\epsilon), \mathcal{O}(\epsilon))$ solution for the bilevel problem within $\mathcal{O}(K \max T_k) = \mathcal{O}(\log(\epsilon^{-1})^2)$ iterations. The proof of Theorem 1 is provided in Appendix D.

**Remark 1** (Other choices of algorithms). *It is worth mentioning that other first-order bilevel algorithms based on the penalty formulation (6), such as $F^2SA$ (Kwon et al., 2023; 2024; Chen et al., 2023a) and BOME (Liu et al., 2022), could also have global convergence. We provided the global convergence based on the PBGD algorithm (Shen et al., 2023) here as an example.*

### 3.3 Ensuring Global Convergence Conditions

In this section, we will provide some key observations to help us establish the global convergence conditions in Section 3.1. Let us first revisit the definition of the penalized objective

$$
\mathsf{L}_\gamma(u, v) \quad = \quad \overbrace{f(u, v)}^{} \quad + \quad \underbrace{\gamma(g(u, v) - g^*(u))}_{\text{Observation 1: joint PL \& blockwise PL over } v}.
$$

$$\text{Additivity of PL functions holds? Observation 2}$$

$$\text{joint PL \& blockwise PL}$$

We have the first observation on the landscape of $g(u, v) - g^*(u)$ over $(u, v)$ and that over $v$.

**Observation 1.** *Under Assumption 1, $g(u, v) - g^*(u)$ is joint PL over $(u, v)$ and blockwise PL over $v$.*

Therefore, if the upper-level objective additionally satisfies joint PL or blockwise PL condition, then whether $\mathsf{L}_\gamma(u, v)$ satisfies joint PL or blockwise PL condition over $v$ depends on the additivity of PL functions. In general, the sum of two joint/blockwise PL functions is not necessarily a joint/blockwise PL function. For the counterexamples in joint/blockwise PL condition, see Appendix C.2.

The *lack of the additivity* of PL functions in general impedes the development of a unified global convergence theory for bilevel problems. However, we highlight another key observation to establish the PL condition of $\mathsf{L}_\gamma(u, v)$ for a special class of problem. This observation implies that a strongly convex function composite with a linear mapping preserves the PL condition under additivity.

**Observation 2.** *Let $h_1 : \mathbb{R}^m \to \mathbb{R}$ and $h_2 : \mathbb{R}^n \to \mathbb{R}$ be strongly convex functions with constant $\mu_1$ and $\mu_2$, respectively. Given any matrix $A \in \mathbb{R}^{m \times d}, B \in \mathbb{R}^{n \times d}$, $h_1(Az)$ and $h_2(Bz)$ satisfy the PL condition over $z \in \mathbb{R}^d$ with constant $\mu_1 \sigma_*(A)$ and $\mu_2 \sigma_*(B)$, respectively. Moreover, $h_1(Az) + h_2(Bz)$ satisfies the PL condition with constant $\min\{\mu_1 \sigma_*^2(A), \mu_2 \sigma_*^2(B)\}$.*

This observation is particularly valuable for analyzing global convergence in bilevel optimization problems involving linear neural networks with strongly convex loss functions, such as the least squares loss or cross-entropy loss within a bounded region.

**Dealing with problem-specific challenges.** In Sections 4 and 5, we will show how to use our bilevel global convergence framework in two problems: representation learning and data hyper-cleaning. In these two applications, we choose linear or bilinear models with least-squares losses. Although these models are relatively simple, only local (non-uniform) versions of the joint PL and blockwise PL conditions are satisfied due to the coupling within the linear network, and the local constants vary with the algorithm's iterations. Nevertheless, we will judiciously verify the joint PL condition and blockwise PL condition **along the optimization trajectory of PBGD** by carefully bounding these constants and establish the optimality equivalence of (1) and (6) at the limit point of PBGD.

## 4 Global Convergence in Representation Learning

In representation learning, we are given a training dataset $\{x_i, y_i\}_{i=1}^N$ and a validation dataset $\{\tilde{x}_i, \tilde{y}_i\}_{i=1}^{N'}$ with data sample $x_i, \tilde{x}_i \in \mathbb{R}^m$ and label $y_i, \tilde{y}_i \in \mathbb{R}^n$. We aim to train a model capable of excelling with unseen data by adjusting only the top layer's weights $W_1$ while keeping the backbone model $W_2$ fixed. For simplicity, we consider the two-layer linear neural network and the mean square loss in this paper, so the resultant bilevel problem can be formulated as

$$
\min_{W_1, W_2 \in \mathcal{S}(W_1)} \frac{1}{2} \|Y_{\text{val}} - X_{\text{val}} W_1 W_2\|^2, \quad \text{s.t.} \quad \mathcal{S}(W_1) = \arg\min_{W_2} \frac{1}{2} \|Y_{\text{trn}} - X_{\text{trn}} W_1 W_2\|^2 \quad (8)
$$

where $X_{\text{trn}} = [x_1, \cdots, x_N]^\top \in \mathbb{R}^{N \times m}, X_{\text{val}} = [\tilde{x}_1, \cdots, \tilde{x}_{N'}]^\top \in \mathbb{R}^{N' \times m}$ are the training and validation data matrix, $Y_{\text{trn}} = [y_1, \cdots, y_N]^\top \in \mathbb{R}^{N \times n}, Y_{\text{val}} = [\tilde{y}_1, \cdots, \tilde{y}_{N'}]^\top \in \mathbb{R}^{N' \times n}$ are the training and validation label matrix and $W_1 \in \mathbb{R}^{m \times h}, W_2 \in \mathbb{R}^{h \times n}$ are the weight matrix. We consider the overparameterized case with $m \geq \max\{N, N'\}$ so that $\mathcal{S}(W_1)$ has multiple solutions. We also assume the neural network is wide $h \geq \max\{m, n\}$ and $X_{\text{trn}}, X_{\text{val}}$ are of full row rank [1].

Let us denote $\mathsf{L}_{\text{trn}}(W_1, W_2) = \frac{1}{2} \|Y_{\text{trn}} - X_{\text{trn}} W_1 W_2\|^2$, $\mathsf{L}_{\text{val}}(W_1, W_2) = \frac{1}{2} \|Y_{\text{val}} - X_{\text{val}} W_1 W_2\|^2$ as the loss of two-layer linear neural network. We are interested in the scenario where the backbone

---

[1]Our theory also works for rank-deficient case by changing $\sigma_{\min}(X_{\text{trn}}), \sigma_{\min}(X_{\text{trn}})$ to $\sigma_*(X_{\text{trn}}), \sigma_*(X_{\text{trn}})$.

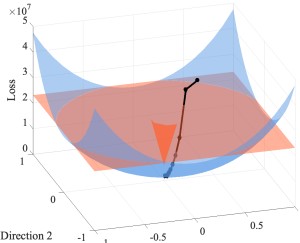 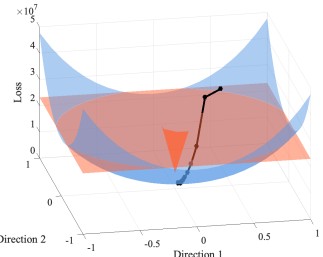 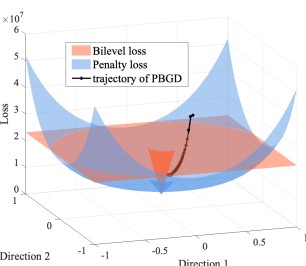

Figure 2: The landscape of $\mathsf{F}(W_1)$ and $\mathsf{L}_\gamma(W_1, W_2)$ with different penalty constant $\gamma = 0.1, 1, 50$ in representation learning. The orange terrain is $\mathsf{F}(W_1)$, while the blue surface is $\mathsf{L}_\gamma(W_1, W_2)$. The black line is the trajectory of PBGD which converges to the global optimum of bilevel loss.

model, when learned from the validation dataset, also achieves a good fit on the training dataset. This ensures that there are hidden patterns applicable across training and validation datasets so that a generalizable backbone model exists. Note that although we employ a two-layer neural network structure to model both the backbone and adaptation layer for representation learning, our analysis is adaptable to multi-layer neural networks as well. To this end, we make the following assumption.

**Assumption 2.** *For any $\epsilon_1, \epsilon_2 \in \mathbb{R}_{\geq 0}$, there exists $(W_1^*, W_2^*)$ being the $(\epsilon_1, \epsilon_2)$ solution to the bilevel representation learning problem (8) such that $\mathsf{L}_{\mathrm{trn}}(W_1^*, W_2^*) - \min_{W_1, W_2} \mathsf{L}_{\mathrm{trn}}(W_1, W_2) \leq \epsilon_2$.*

Assumption 2 ensures that the bilevel problem has at least one full-rank solution $W_1^*$ so that is not degenerate. We present some *sufficient conditions* for it in Appendix F.1.

To solve (8), we resort to its penalized function as stated (6) with the upper-level function $f(u, v) = \mathsf{L}_{\mathrm{val}}(W_1, W_2)$, and the lower-level objective $g(u, v) = \mathsf{L}_{\mathrm{trn}}(W_1, W_2)$, which is defined as

$$\mathsf{L}_\gamma(W_1, W_2) := \mathsf{L}_{\mathrm{val}}(W_1, W_2) + \gamma(\mathsf{L}_{\mathrm{trn}}(W_1, W_2) - \mathsf{L}_{\mathrm{trn}}^*(W_1)) \tag{9}$$

where $\mathsf{L}_{\mathrm{trn}}^*(W_1) = \min_{W_2} \mathsf{L}_{\mathrm{trn}}(W_1, W_2)$. Then we can run PBGD 1 with $(u, v, w) = (W_1, W_2, W_3)$ where $W_3 \in \mathbb{R}^{h \times n}$ is an axillary variable used to estimate the gradient of the value function.

To verify the global convergence condition (2), we first observe that both $\mathsf{L}_{\mathrm{val}}(W_1, W_2)$ and $\mathsf{L}_{\mathrm{trn}}(W_1, W_2)$ are in the form of a quadratic function composite with a linear mapping. Moreover, we find that $\mathsf{L}_{\mathrm{trn}}^*(W_1) = 0$ if $W_1$ is of full row rank and $W_1^k$ will maintain full row rank on the trajectory generated by PBGD, so that $\mathsf{L}_{\mathrm{trn}}(W_1, W_2) - \mathsf{L}_{\mathrm{trn}}^*(W_1)$ will be in the form of a quadratic function composite with a linear mapping along the optimization trajectory. Therefore, Observation 2 implies that $\mathsf{L}_\gamma(W_1, W_2)$ is a PL function. Figure 2 shows the landscape comparison of $\mathsf{F}(W_1) := \min_{W_2 \in \mathcal{S}(W_1)} \mathsf{L}_{\mathrm{val}}(W_1, W_2)$ and $\mathsf{L}_\gamma(W_1, W_2)$ in representation learning. From Figure 2, it can be seen that the nested bilevel function $\mathsf{F}(W_1)$ is nonconvex and discontinuous, while the penalized loss $\mathsf{L}_\gamma(W_1, W_2)$ has a smoother landscape, effectively steering the weight matrix updated by PBGD towards the global optimal solution. Formally, we have the following lemma.

**Lemma 1** (Joint PL condition and descent lemma over trajectory). *Let $X_\gamma = [X_{\mathrm{val}}; \sqrt{\gamma} X_{\mathrm{trn}}]$ be the concatenated data matrix. Then under Assumption 2, if $\sigma_{\min}^2(W_1^k) > 0$, $\sigma_{\min}^2(W_2^k) > 0$, the joint PL inequality holds with $\mu_k = (\sigma_{\min}^2(W_1^k) + \sigma_{\min}^2(W_2^k))\sigma_*^2(X_\gamma)$, that is*

$$\|\nabla \mathsf{L}_\gamma(W_1^k, W_2^k)\|^2 \geq 2\mu_k(\mathsf{L}_\gamma(W_1^k, W_2^k) - \min_{W_1, W_2} \mathsf{L}_\gamma(W_1, W_2)).$$

*The descent lemma holds with $L_k$ defined in (51) and $\|\delta_k\|$ being the estimation error of $\nabla \mathsf{L}_\gamma^*(W_1^k)$*

$$\mathsf{L}_\gamma(W_1^{k+1}, W_2^{k+1}) \leq \mathsf{L}_\gamma(W_1^k, W_2^k) - \left(\frac{\alpha}{2} - \alpha^2 L_k\right) \|\nabla \mathsf{L}_\gamma(W_1^k, W_2^k)\|^2 + \left(\frac{\alpha}{2} + \alpha^2 L_k\right) \|\delta_k\|^2.$$

To this end, using the joint PL condition together with the standard descent lemma leads to one-step contraction in the optimality gap of the penalized objective. Through induction, we can then establish the lower bounds of $\sigma_{\min}^2(W_1^k)$ and $\sigma_{\min}^2(W_2^k)$, as well as the upper bounds of $\sigma_{\max}^2(W_1^k)$ and $\sigma_{\max}^2(W_2^k)$. Consequently, these yield $k$-independent lower and upper bounds for $\mu_k$ and $L_k$, which enable us to demonstrate the almost linear convergence of PBGD.

**Theorem 2** (Global convergence of PBGD for representation learning). *Under Assumption 2 and choose $\gamma = \mathcal{O}(\epsilon^{-0.5}), T_k = \mathcal{O}(\log(\epsilon^{-1}))$, there exists $\mu = \mathcal{O}(\gamma)$, $L = \mathcal{O}(\gamma), \alpha = \mathcal{O}(\gamma^{-1})$, such that $\mu_k \geq \mu, L_k \leq L$, and the iteration complexity of PBGD in Algorithm 1 to achieve $(\epsilon, \epsilon)$ global optimal point of the bilevel representation learning problem (8) is $\mathcal{O}(\log^2(\epsilon^{-1}))$.*

## 5   GLOBAL CONVERGENCE IN DATA HYPER-CLEANING

Data hyper-cleaning aims to leverage a small, clean validation dataset to enhance the quality of a larger training dataset, which may be hindered by some noisy or unreliable data points. This process is widely used when the access to clean data is costly but the noisy data is not like in recommendation systems (Wang et al., 2023; Chen et al., 2022). To do so, we are given a training dataset $\{x_i, y_i\}_{i=1}^N$ with data sample $x_i \in \mathbb{R}^m$ and label $y_i \in \mathbb{R}^n$, where each label may be corrupted. We are also given the clean validation data $\{\tilde{x}_i, \tilde{y}_i\}_{i=1}^{N'}$ to guide the training. Let $u \in \mathbb{R}^N$ be the classification vector trained to label the noisy data and $W \in \mathbb{R}^{m \times n}$ be the model weight, the bilevel problem of data hyper-cleaning is given by $\min_{u \in \mathcal{U}, W \in S(u)} \frac{1}{2} \sum_{i=1}^{N'} (\tilde{y}_i - \tilde{x}_i^\top W)^2$, s.t. $\mathcal{S}(u) = \arg\min_W \frac{1}{2} \sum_{i=1}^N \psi(u_i)(y_i^\top - x_i^\top W)^2$ where $\mathcal{U} = [-\bar{u}, \bar{u}]^N$ is used to avoid trivial solution $u_i = -\infty$ and $\psi(\cdot)$ is the sigmoid function.

Let $X_{\text{trn}} = [x_1, \cdots, x_N]^\top \in \mathbb{R}^{N \times m}, X_{\text{val}} = [\tilde{x}_1, \cdots, \tilde{x}_{N'}]^\top \in \mathbb{R}^{N' \times m}, Y_{\text{trn}} = [y_1, \cdots, y_N]^\top \in \mathbb{R}^{N \times n}, Y_{\text{val}} = [\tilde{y}_1, \cdots, \tilde{y}_{N'}]^\top \in \mathbb{R}^{N' \times n}$ be the training and validation data and label matrix, $\psi_N(\cdot)$ denotes the diagnal matrix of element wise sigmoid function and $\sqrt{\cdot}$ denote the elementwise square root operator. We can formulate the data hyper-cleaning problem as the following bilevel problem

$$\min_{u \in \mathcal{U},\ W \in S(u)} \frac{1}{2} \|Y_{\text{val}} - X_{\text{val}} W\|^2, \quad \text{s.t.} \quad \mathcal{S}(u) = \arg\min_W \frac{1}{2} \left\| \sqrt{\psi_N(u)} \left(Y_{\text{trn}} - X_{\text{trn}} W\right) \right\|^2. \quad (10)$$

Note that for data hyper-cleaning, where the primary concern involves the interaction between the classification parameter $u$ and the model weight $W$, we adopt a one-layer neural network structure for simplicity in notations. Nonetheless, our results can also be extended to multi-layer linear networks.

We consider the overparameterized case with $m \geq \max\{N, N'\}$. To model a corrupted training setting, we assume the concatenate data matrix $[X_{\text{val}}; X_{\text{trn}}]$ is rank-deficient so that the training and validation objectives do not share a common weight minimizer, but there exists a parameter $u^* \in \mathcal{U}$ such that the selected concatenated matrix $[X_{\text{val}}; \sqrt{\psi_N(u^*)} X_{\text{trn}}]$ is nearly full rank by neglecting the rows with small $\psi(u_i)$, meaning that $\text{rank}([X_{\text{val}}; \sqrt{\psi_N(u^*)} X_{\text{trn}}]) = N + N' - \left|\{i : \psi(u_i^*) < \tilde{\psi}\}\right|$, where $\tilde{\psi} > 0$ is a threshold close to 0 and $|\cdot|$ denotes the cardinality of the set $\{i : \psi(u_i^*) < \tilde{\psi}\}$.

Let us denote $\ell_{\text{trn}}(u, W) = \frac{1}{2}\|\sqrt{\psi_N(u)}(Y_{\text{trn}} - X_{\text{trn}} W)\|^2$ and the value function in data hyper-cleaning as $\ell_{\text{trn}}^*(u) = \min_W \ell_{\text{trn}}(u, W)$. To solve (10), we resort to its penalized reformulation

$$\ell_\gamma(u, W) := \ell_{\text{val}}(W) + \gamma(\ell_{\text{trn}}(u, W) - \ell_{\text{trn}}^*(u)). \quad (11)$$

Then we can run PBGD 2 with $(u, v, w) = (u, W, Z)$ where $Z \in \mathbb{R}^{m \times n}$ is an axillary variable used to estimate $\nabla \ell_{\text{trn}}^*(u)$. Since both $\ell_{\text{val}}(W)$ and $\ell_{\text{trn}}(u, W)$ are in the form of strongly convex functions composite with a linear mapping, $\ell_\gamma(u, W)$ is blockwise PL with respect to $W$ from Observation 2. To prove the blockwise PL condition over $u$, we first show that $\mathcal{S}(u)$ is independent of $u$ and obtain the form of $\ell_{\text{trn}}^*(u)$ by plugging in the closed form of $\mathcal{S}(u)$, for which we defer the details in Lemma 23 in Appendix G.1. As a result, the penalized objective has the closed-form expression as

$$\ell_\gamma(u, W) = \ell_{\text{val}}(W) + \frac{\gamma}{2} \sum_{i=1}^N \psi(u_i) \left[ \|y_i^\top - x_i^\top W\|^2 - \|y_i\|^2 \mathbb{1}([X_{\text{trn}} X_{\text{trn}}^\dagger]_i \neq 1) \right]. \quad (12)$$

Thus the landscape of $\ell_\gamma(u, W)$ over $u$ is fully characterized by the sigmoid function, which is PL on $u \in \mathcal{U}$. Consequently, $\ell_\gamma(u, W)$ is blockwise PL. We formalize our findings in the next lemma.

**Lemma 2** (Blockwise PL condition)**.** *If* $X_{\text{trn}} X_{\text{trn}}^\dagger$ *is a diagonal matrix, then for any* $u \in \mathcal{U}$ *and* $W$, *there exists* $L_w^\gamma, \mu_w^\gamma = \mathcal{O}(\gamma)$ *such that* $\ell_\gamma(u, W)$ *is* $L_w^\gamma$-*smooth and* $\mu_w^\gamma$-*PL over* $W$. *Moreover,* $\ell_\gamma(u, W)$ *is* $\gamma \ell_{\text{trn}}(W)$ *smooth and* $\frac{\gamma c(W) \psi(\bar{u})(1-\psi(\bar{u}))^2}{4}$-*PL over* $u \in \mathcal{U}$, *where*

$$c(W) = \min_i \left\{ \|y_i^\top - x_i^\top W\|^2 - \|y_i\|^2 \mathbb{1}([X_{\text{trn}} X_{\text{trn}}^\dagger]_i \neq 1) \right\}_{>0}$$

*is defined as the lower bound of the **positive mismatch** in the training loss.*

Although the blockwise PL constant of $\ell_\gamma(u, W)$ depends on $W$, we can derive a uniform positive lower bound of $\mu_u := \min_{u \in \mathcal{U}} c(W_\gamma^*(u))$ based on the acute matrix perturbation theory, and thus, we can obtain the global convergence results for PBGD (Algorithm 2) on data hyper-cleaning problem.

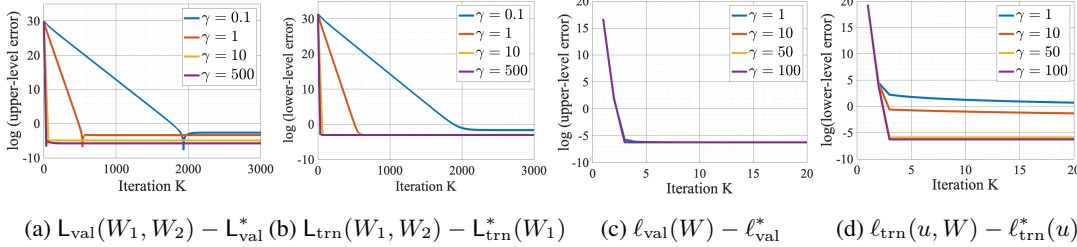

(a) $\mathsf{L}_{\mathrm{val}}(W_1, W_2) - \mathsf{L}^*_{\mathrm{val}}$ (b) $\mathsf{L}_{\mathrm{trn}}(W_1, W_2) - \mathsf{L}^*_{\mathrm{trn}}(W_1)$ (c) $\ell_{\mathrm{val}}(W) - \ell^*_{\mathrm{val}}$ (d) $\ell_{\mathrm{trn}}(u, W) - \ell^*_{\mathrm{trn}}(u)$

Figure 3: Relative errors at upper-level and lower-level in $\log$ scale of PBGD versus iteration $K$ under different $\gamma$, where $\mathsf{L}^*_{\mathrm{val}} = \min_{W_1, W_2 \in \mathcal{S}(W_1)} \mathsf{L}_{\mathrm{val}}(W_1, W_2)$ and $\ell^*_{\mathrm{val}} = \min_{u, W \in \mathcal{S}(u)} \ell_{\mathrm{val}}(W)$. (a)–(b) are for PBGD 1 in representation learning, and (c)-(d) are for PBGD 2 in data hyper-cleaning.

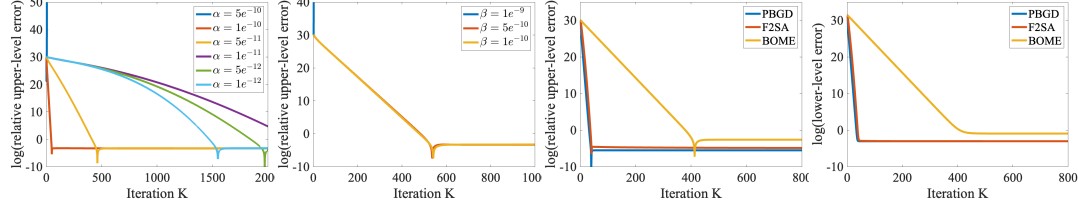

Figure 4: Relative errors in $\log$ scale of different methods under different stepsizes in representation learning. (a)–(b) are ablation study for stepsizes $\alpha, \beta$ in PBGD, and (c)-(d) are for different methods.

**Theorem 3** (Global convergence for data hyper-cleaning). *Suppose* $[X_{\mathrm{trn}}; X_{\mathrm{val}}][X_{\mathrm{trn}}; X_{\mathrm{val}}]^\dagger$ *is a diagonal matrix. If the stepsizes in Algorithm 2 satisfy* $\alpha \leq \frac{1}{L_u} = \mathcal{O}(1/\gamma)$, $\beta \leq \frac{1}{L_w}$, $\tilde{\beta} \leq \frac{1}{L^\gamma_w}$, $T_k = \mathcal{O}(\log(\epsilon^{-1}))$, *and* $\bar{u} \geq 1$, *then the iteration complexity of PBGD in Algorithm 2 to achieve* $(\epsilon, \epsilon)$ *global optimal point of the data hyper-cleaning problem* (10) *is* $\mathcal{O}(\log(\epsilon^{-1})^2)$.

## 6 NUMERICAL EXPERIMENTS AND CONCLUSIONS

We verify the global convergence property of PBGD for representation learning and data hyper-cleaning with different penalty constant $\gamma$; see results in Figure 3 and the detailed setup in Appendix H. We also conduct ablation studies of stepsizes $\alpha, \beta$ for PBGD and test the performance of other state-of-the-art fully first order bilevel methods (F$^2$SA (Kwon et al., 2023), BOME (Liu et al., 2022)). The results are shown in Figure 4. We summarized empirical findings below.

**Almost linear onvergence of PBGD.** PBGD with proper stepsizes converges almost linearly to the optimum of bilevel representation learning and data hyper-cleaning, which is aligned with the convergence rate we obtained from Theorem 1–3.

**Impact of $\gamma$ on optimality gap.** Enlarging the penalty constant $\gamma$ reduces the target optimality gap $\epsilon$, consistent with Theorems 1–3, which establish that $\gamma$ is inversely propotional to the target error $\epsilon$.

**Stepsize sensitivity.** Stepsizes $\alpha, \beta$ of PBGD should be carefully chosen. Excessively large step sizes may cause divergence, while small step sizes result in slow convergence. Theoretically, Theorems 1–3 provide an upper bound for $\alpha, \beta$ to guarantee the convergence of PBGD.

**Convergence of penalty-based first-order bilevel methods.** All tested first-order bilevel methods (PBGD, F$^2$SA, BOME) successfully converge to the global optimum in representation learning task in an almost linear rate. This observation suggests that our local PL-based analysis can be extended to other penalty reformulation-based algorithms.

**Conclusion.** In this paper, we proposed two benign landscape conditions for bilevel problems, tailored to isomorphic and heterogeneous bilevel learning problems. We proved that PBGD, in either Jacobi or Gauss-Seidel fashions, converges to the global optimal solution at an almost linear rate under these conditions respectively. These global conditions were rigorously verified in representation learning and data hyper-cleaning along the optimization trajectory of PBGD by local non-uniform analysis. The global convergence property of PBGD over two applications is thus guaranteed. Numerical results validate our theory and confirm that PBGD is globally convergent.

ACKNOWLEDGMENT

The work was supported by National Science Foundation (NSF) project 2134168, NSF CAREER project 2047177, NSF ECCS 2412486, and the IBM-Rensselaer Future of Computing Research Collaboration. We also thank Ziqing Xu and Liuyuan Jiang for their insightful feedback and constructive suggestions, which are instrumental in enhancing the clarity and quality of our paper.

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

# Supplementary Material for " Unlocking Global Optimality in Bilevel Optimization: A Pilot Study "

## Table of Contents

## A ADDITIONAL RELATED WORKS

**Stationary point convergence.** The recent interest in developing efficient gradient-based bilevel methods and analyzing their nonasymptotic convergence rate has been stimulated by (Ghadimi & Wang, 2018; Ji et al., 2021; Hong et al., 2023; Chen et al., 2021). Based on different Hessian inversion approximation techniques, these algorithms can be categorized into iterative differentiation (Franceschi et al., 2017; 2018; Grazzi et al., 2020) and approximate implicit differentiation-based approaches (Chen et al., 2021; Ghadimi & Wang, 2018; Hong et al., 2023; Ji et al., 2021; Pedregosa, 2016). Later on, the research has been extended to variance reduction and momentum based methods (Khanduri et al., 2021; Yang et al., 2021; Dagréou et al., 2022); warm-started algorithms (Arbel

& Mairal, 2022; Li et al., 2022; Liu et al., 2023); distributed approaches (Tarzanagh et al., 2022; Lu et al., 2022; Yang et al., 2022); and methods that are able to tackle with constrained (Xiao et al., 2023b; Khanduri et al., 2023; Xu & Zhu, 2023) or non-strongly-convex lower-level problem (Xiao et al., 2023a; Liu et al., 2021b;a; Chen et al., 2023b; Sow et al., 2022b). Recent works have reformulated the bilevel optimization problem as a single-level constrained problem, and solved it via the penalty-based gradient method (Shen et al., 2023); see also (Liu et al., 2022; Kwon et al., 2023; 2024; Chen et al., 2023a; Lu & Mei, 2023). This method not only further enhanced the computational efficiency by using only first-order information, but also broadened the applicability of the bilevel framework to a wider class of problems. Nevertheless, none of these attempts the issue of global optimum convergence.

## B  AUXILIARY LEMMAS

**Additional Notations.**     Let matrix $A \in \mathbb{R}^{m \times n}$. The range space of $A$ is denoted as $\text{Ran}(A) = \{Ay : y \in \mathbb{R}^n\}$, while the null space of $A$ is given by $\text{Ker}(A) = \{y \in \mathbb{R}^n : Ay = 0\}$. Let $A_{ij}$ represent the element located at the $i$-th row and $j$-th column of matrix $A$. Let $\|A\|$ and $\|A\|_2$ be the Frobenius norm and spectral norm of $A$, respectively. $\mathbb{B}(A, r) := \{A' \in \mathbb{R}^{p \times q} \mid \|A' - A\| \leq a\}$ denotes the closed ball with center $A$ and radius $a$. Let $\theta \in \mathbb{R}^d$ be a point and $S \subset \mathbb{R}^d$ be a set, we define the distance of $\theta$ to $S$ as $d(\theta, S) = \inf\{\|\theta - a\| \mid a \in S\}$. Let us denote $\ell_{\text{trn}}(W) = \frac{1}{2} \|Y_{\text{trn}} - X_{\text{trn}}W\|^2, \ell_{\text{val}}(W) = \frac{1}{2} \|Y_{\text{val}} - X_{\text{val}}W\|^2$ as the square loss.

**Lemma 3** (Matrix Inequality).

$$\langle A, B \rangle \leq \|A\| \cdot \|B\|, \tag{13}$$

$$2\|AB\| \leq \|A\|^2 + \|B\|^2, \tag{14}$$

$$\|AB + CD\|^2 \leq \left[\sigma_{\max}^2(A) + \sigma_{\max}^2(C)\right]^2 \cdot \left[\|B\|^2 + \|D\|^2\right], \tag{15}$$

$$\|A\|^2 + \|B\|^2 \leq 2\|A + B\|^2, \tag{16}$$

$$\sigma_{\max}(AB) \leq \sigma_{\max}(A)\sigma_{\max}(B). \tag{17}$$

*Moreover, if $A$ has full column rank and $B \neq 0$ or if $A \neq 0$ and $B$ has full row rank, it holds that*

$$\sigma_{\min}(AB) \geq \sigma_{\min}(A)\sigma_{\min}(B) \tag{18}$$

**Lemma 4** ((Zou et al., 2020, Lemma B.3)). *Let $A \in \mathbb{R}^{p \times s}$ be a rank-$s$ matrix. Then for any $B \in \mathbb{R}^{s \times q}$, we have*

$$\sigma_{\min}(A)\|B\| \leqslant \|AB\| \leqslant \sigma_{\max}(A)\|B\|.$$

**Lemma 5.** *In representation learning, we consider the overparameterized wide neural network case with $m \geq \max\{N, N'\}, h \geq \max\{m, n\}$ and $X_{\text{trn}}, X_{\text{val}}$ are in full row rank. Therefore, we have,*

$$\min_W \ell_{\text{trn}}(W) = 0, \quad \min_{W_1, W_2} \mathsf{L}_{\text{trn}}(W_1, W_2) = 0, \quad \min_W \ell_{\text{val}}(W) = 0, \quad \min_{W_1, W_2} \mathsf{L}_{\text{val}}(W_1, W_2) = 0$$

$$\min_W (\ell_{\text{trn}} + \gamma \ell_{\text{val}})(W) = \min_{W_1, W_2} (\mathsf{L}_{\text{trn}} + \gamma \mathsf{L}_{\text{trn}})(W_1, W_2)$$

*holds for any $\gamma$. Moreover, for $W_1$ with $\sigma_{\min}(W_1) > 0$, it holds that $\mathsf{L}_{\text{trn}}^*(W_1) = \min_{W_2} \mathsf{L}_{\text{trn}}(W_1, W_2) = 0$.*

**Lemma 6** ((Karimi et al., 2016, Theorem 2)). *We say a function $h(\theta)$ is $\mu_h$ PL if it satisfies*

$$\|\nabla h(\theta)\|^2 \geq 2\mu_h(h(\theta) - h^*)$$

*where $h^* = \min h(\theta)$. If $h(\theta)$ is $L_h$-Lipschitz smooth and PL in $\theta$ with $\mu_h$, then it satisfies the error bound condition with $\mu_h$, i.e.*

$$\|\nabla h(\theta)\| \geq \mu_h d(\theta, S). \tag{19}$$

*where $S = \arg\min h(\theta)$. Moreover, it also satisfies the quadratic growth condition with $\mu_h$, i.e.*

$$h(\theta) - h^* \geq \frac{\mu_h}{2} d(\theta, S)^2. \tag{20}$$

*where $h^* = \min h(\theta)$. Conversely, if $h(\theta)$ is $L_h$-Lipschitz smooth and satisfies the error bound condition with $\mu_h$, then it is PL in $\theta$ with $\mu_h/L_h$.*

**Lemma 7** ((Oymak & Soltanolkotabi, 2019, Theorem 5.2)). *Suppose that $h(\theta)$ is $L_h$ Lipschitz smooth and $\mu_h$ PL over $\theta$. If we execute $T$-step gradient descent following $\theta^{t+1} = \theta^t - \beta \nabla h(\theta^t)$ with $\beta \le \frac{1}{L_h}$, the output satisfies*

$$h(\theta^T) - \min h(\theta) \le (1 - \beta\mu_h)^T \left( h(\theta^0) - \min h(\theta) \right)$$

*Moreover, the iterates generated by gradient descent is bounded, i.e.*

$$\|\theta_T - \theta_0\| \le \sum_{t=0}^{\infty} \|\theta_{t+1} - \theta_t\|_{\ell_2} \le \sqrt{\frac{8(h(\theta_0) - \min h(\theta_0))}{\mu_h}}$$

The following lemma gives the gradient of the value function.

**Lemma 8** ((Nouiehed et al., 2019, Lemma A.5)). *Suppose $g(u,v)$ is $\mu_g$ PL over $v$ and is $\ell_{g,1}$-Lipschitz smooth, then $g^*(u)$ is differentiable with the gradient*

$$\nabla g^*(u) = \nabla_u g(u,v), \quad \forall v \in \mathcal{S}(u).$$

*Moreover, $g^*(u)$ is $L_g$-smooth with $L_g := \ell_g(1 + \ell_g/2\mu_g)$.*

**Lemma 9.** *If $\|\delta_k\| \le \delta$ and $\sigma_{\min}(W_1^k) > 0, \sigma_{\min}(W_2^k) > 0$, then the following inequalities hold*

$$2 \left\| \left( \nabla_{W_1} \mathsf{L}_\gamma(W_1^k, W_2^k) + \delta_k \right) \nabla_{W_2} \mathsf{L}_\gamma(W_1^k, W_2^k) \right\|$$
$$\le \|\nabla \mathsf{L}_\gamma(W_1^k, W_2^k) + \delta_k\|^2 + \|\nabla \mathsf{L}_\gamma(W_1^k, W_2^k)\|^2 \tag{21}$$

$$\left\| \left( \nabla_{W_1} \mathsf{L}_\gamma(W_1^k, W_2^k) + \delta_k \right) \nabla_{W_2} \mathsf{L}_\gamma(W_1^k, W_2^k) \right\|$$
$$\le 2\sigma_{\max}^2(X_\gamma)\sigma_{\max}(W^k)(\mathsf{L}_\gamma(W_1^k, W_2^k) - \mathsf{L}_\gamma^*) + \delta\sigma_{\max}(W_1^k)\sqrt{2\sigma_{\max}^2(X_\gamma)(\mathsf{L}_\gamma(W_1^k, W_2^k) - \mathsf{L}_\gamma^*)} \tag{22}$$

$$\left\| \left( \nabla_{W_1} \mathsf{L}_\gamma(W_1^k, W_2^k) + \delta_k \right) W_2^k + W_1^k \nabla_{W_2} \mathsf{L}_\gamma(W_1^k, W_2^k) \right\|$$
$$\le \left( \sigma_{\max}^2(W_1^k) + \sigma_{\max}^2(W_2^k) \right) \sqrt{2\sigma_{\max}^2(X_\gamma)(\mathsf{L}_\gamma(W_1^k, W_2^k) - \mathsf{L}_\gamma^*)} + \delta\sigma_{\max}(W_2^k) \tag{23}$$

**Proof:** (21) holds from (14) by letting $A = \nabla_{W_1} \mathsf{L}_\gamma(W_1^k, W_2^k) + \delta_k$ and $B = \nabla_{W_2} \mathsf{L}_\gamma(W_1^k, W_2^k)$.

Note that when $\sigma_{\min}(W_1^k) > 0, \sigma_{\min}(W_2^k) > 0$, we have $\mathsf{L}_\gamma(W_1^k, W_2^k) = \widetilde{\mathsf{L}}_\gamma(W_1^k, W_2^k) = \ell_\gamma(W^k)$ where $\ell_\gamma = \ell_{\mathrm{val}} + \gamma\ell_{\mathrm{trn}}$. Thus (22) can be derived from

$$\left\| \left( \nabla_{W_1} \mathsf{L}_\gamma(W_1^k, W_2^k) + \delta_k \right) \nabla_{W_2} \mathsf{L}_\gamma(W_1^k, W_2^k) \right\|$$
$$\le \left\| \nabla_{W_1} \mathsf{L}_\gamma(W_1^k, W_2^k) \nabla_{W_2} \mathsf{L}_\gamma(W_1^k, W_2^k) \right\| + \delta \left\| \nabla_{W_2} \mathsf{L}_\gamma(W_1^k, W_2^k) \right\|$$
$$\le \left\| \nabla\ell_\gamma(W^k)(W_2^k)^\top (W_1^k)^\top \nabla\ell_\gamma(W^k) \right\| + \delta \left\| \nabla_{W_2} \mathsf{L}_\gamma(W_1^k, W_2^k) \right\|$$
$$\le \left\| \nabla\ell_\gamma(W^k) W^{k,\top} \nabla\ell_\gamma(W^k) \right\| + \delta \left\| (W_1^k)^\top \nabla\ell_\gamma(W^k) \right\|$$
$$\le \sigma_{\max}(W^k) \|\nabla\ell_\gamma(W^k)\|^2 + \delta\sigma_{\max}(W_1^k) \|\nabla\ell_\gamma(W^k)\|$$
$$\overset{(a)}{\le} 2\sigma_{\max}^2(X_\gamma)(\mathsf{L}_\gamma(W_1^k, W_2^k) - \mathsf{L}_\gamma^*) + \delta\sigma_{\max}(W_1^k)\sqrt{2\sigma_{\max}^2(X_\gamma)(\mathsf{L}_\gamma(W_1^k, W_2^k) - \mathsf{L}_\gamma^*)}$$

where $W^k = W_1^k W_2^k$ and (a) holds because $\ell_\gamma$ is $\sigma_{\max}^2(X_\gamma)$- smooth so that

$$\|\nabla\ell_\gamma(W^k)\|^2 \le 2\sigma_{\max}^2(X_\gamma)(\ell_\gamma(W^k) - \ell_\gamma^*) \overset{(41),(44)}{\le} 2\sigma_{\max}^2(X_\gamma)(\mathsf{L}_\gamma(W_1^k, W_2^k) - \mathsf{L}_\gamma^*). \tag{24}$$

Likewise, we can derive (23) similarly by

$$\left\| \left( \nabla_{W_1} \mathsf{L}_\gamma(W_1^k, W_2^k) + \delta_k \right) W_2^k + W_1^k \nabla_{W_2} \mathsf{L}_\gamma(W_1^k, W_2^k) \right\|$$
$$\le \left\| \nabla_{W_1} \mathsf{L}_\gamma(W_1^k, W_2^k) W_2^k \right\| + \delta\sigma_{\max}(W_2^k) + \|W_1^k \nabla_{W_2} \mathsf{L}_\gamma(W_1^k, W_2^k)\|$$
$$\le \left\| \nabla\ell_\gamma(W^k)(W_2^k)^\top W_2^k \right\| + \delta\sigma_{\max}(W_2^k) + \|W_1^k (W_1^k)^\top \nabla\ell_\gamma(W^k)\|$$
$$\le \left( \sigma_{\max}^2(W_1^k) + \sigma_{\max}^2(W_2^k) \right) \|\nabla\ell_\gamma(W^k)\| + \delta\sigma_{\max}(W_2^k)$$
$$\overset{(24)}{\le} \left( \sigma_{\max}^2(W_1^k) + \sigma_{\max}^2(W_2^k) \right) \sqrt{2\sigma_{\max}^2(X_\gamma)(\mathsf{L}_\gamma(W_1^k, W_2^k) - \mathsf{L}_\gamma^*)} + \delta\sigma_{\max}(W_2^k).$$

**Lemma 10** (Pseudoinverse of matrix product). *For two real matrix $M_1 \in \mathbb{R}^{m \times m}$ and $M_2 \in \mathbb{R}^{m \times n}$, if $M_1$ is a diagonal matrix and is invertible and $M_2 M_2^\dagger$ is diagonal, then we have $(M_1 M_2)^\dagger = M_2^\dagger M_1^{-1}$.*

**Proof:** According to (Greville, 1966, Theorem 2), $(M_1 M_2)^\dagger = M_2^\dagger M_1^\dagger = M_2^\dagger M_1^{-1}$ holds if and only if both $M_1^\dagger M_1 M_2 M_2^\top$ and $M_1^\top M_1 M_2 M_2^\dagger$ are Hermitian matrix. As $M_1$ and $M_2$ are real matrix, it remains to prove that both $M_1^\dagger M_1 M_2 M_2^\top$ and $M_1^\top M_1 M_2 M_2^\dagger$ are symmetric. By the invertiability of $M_1$,

$$M_1^\dagger M_1 M_2 M_2^\top = M_1^{-1} M_1 M_2 M_2^\top = M_2 M_2^\top$$

so $M_1^\dagger M_1 M_2 M_2^\top$ is symmetric.

For $M_1^\top M_1 M_2 M_2^\dagger$, the starting point is a fact that diagonal matrix is commutative with diagonal matrix. This is because for diagonal matrix $\Lambda_1$ and diagonal matrix $\Lambda_2$, it holds that

$$\Lambda_1 \Lambda_2 = (\Lambda_2^\top \Lambda_1^\top)^\top = (\Lambda_2 \Lambda_1)^\top = \Lambda_2 \Lambda_1$$

where the last equation is due to the symmetricity of $\Lambda_2 \Lambda_1$.

Since both $M_1^\top M_1$ and $M_2 M_2^\dagger$ are diagonal matrix, we then have

$$M_1^\top M_1 M_2 M_2^\dagger = M_2 M_2^\dagger M_1^\top M_1 = (M_1^\top M_1 M_2 M_2^\dagger)^\top$$

so that $M_1^\top M_1 M_2 M_2^\dagger$ is also symmetric, which completes the proof.

**Definition 3** ((Stewart, 1977, Acute matrix)). *Let $P_A = AA^\dagger, P_B = BB^\dagger$ be the orthogonal projection matrix onto the range space $\mathrm{Ran}(A)$ and $\mathrm{Ran}(B)$. We say matrix $A \in \mathbb{R}^{m \times n}$ and $B \in \mathbb{R}^{m \times n}$ are acute if $\|P_A - P_B\| < 1$. Moreover, a class of parameterized matrix family $\{A(u)\}, u \in \mathcal{U}$, is said to be acute if for any $u^1, u^2 \in \mathcal{U}$, $A(u^1)$ and $A(u^2)$ are acute.*

**Proposition 1** ((Stewart, 1977)). *If $A \in \mathbb{R}^{m \times n}$ and $B \in \mathbb{R}^{m \times n}$ are acute, then $\mathrm{Ran}(A) = \mathrm{Ran}(B)$.*

**Lemma 11** ((Stewart, 1977, Theorem 2.5)). *Matrix $A \in \mathbb{R}^{m \times n}$ and $B \in \mathbb{R}^{m \times n}$ are acute if and only if*

$$\mathrm{rank}(A) = \mathrm{rank}(B) = \mathrm{rank}(P_A B R_A)$$

*where $P_A = AA^\dagger, R_A = A^\dagger A$ are the orthogonal projection matrix onto the column space $\mathrm{Ran}(A)$ and the row space $\mathrm{Ran}(A^\top)$.*

**Lemma 12** (Rank equations (Gardner)). *For $A \in \mathbb{R}^{m \times n}, B \in \mathbb{R}^{v \times p}, C \in \mathbb{R}^{p \times q}$, if $A$ is full column rank, and $C$ is full row rank, then it holds that*

$$\mathrm{rank}(AB) = \mathrm{rank}(B), \quad \mathrm{rank}(BC) = \mathrm{rank}(B).$$

*Moreover, for any $D \in \mathbb{R}^{m \times n}$, $\mathrm{rank}(D^\top D) = \mathrm{rank}(D)$.*

## C    PROOF FOR EXAMPLES 1 AND 3

### C.1    PROOF FOR EXAMPLE 1

The gradients of the two objectives in Example 1 can be computed as

$$\nabla f(u, v) = \begin{bmatrix} u - 2\sin(v) \\ -2(u - 2\sin(v))\cos(v) \end{bmatrix} \quad \text{and} \quad \nabla g(u, v) = \begin{bmatrix} u - v \\ -(u - v) \end{bmatrix}.$$

Besides, also using the fact that $\min_{u,v} f(u, v) = \min_{u,v} g(u, v) = 0$, we have

$$\|\nabla f(u, v)\|^2 = (u - 2\sin(v))^2 \times (1 + 4\cos(v)^2)$$

$$\geq 2 \times \frac{1}{2}(u - 2\sin(v))^2 = 2 \times \left( f(u, v) - \min_{u,v} f(u, v) \right)$$

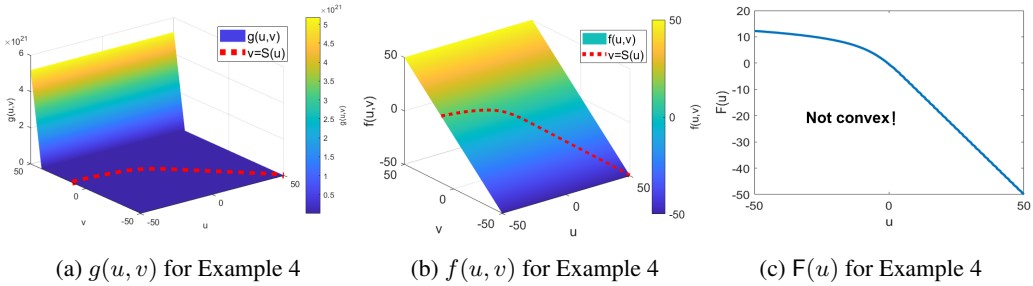

(a) $g(u, v)$ for Example 4       (b) $f(u, v)$ for Example 4       (c) $\mathsf{F}(u)$ for Example 4

Figure 5: Visualization of $g(u, v)$, $f(u, v)$ and $\mathsf{F}(u)$ in Example 4.

$$\|\nabla g(u, v)\|^2 = 2(u - v)^2 \geq 4 \times \frac{1}{2}(u - v)^2 = 4 \times \left( g(u, v) - \min_{u,v} g(u, v) \right)$$

which suggests $f(u, v)$ is 2-PL and $g(u, v)$ is 1-PL. The lower-level solution set is earned by setting $u = v$. In this way, the overall objective is $\mathsf{F}(u) = f(u, \mathcal{S}(u)) = \frac{1}{2}(u - 2\sin(u))^2$ and it is minimized when $u = 0$. By calculating its gradient $\nabla F(u) = (u - 2\sin(u))(1 - 2\cos(u))$, we have

$$\|\nabla F(u)\|^2 = (u - 2\sin(u))^2(1 - 2\cos(u))^2.$$

Since $(1 - 2\cos(u))^2 = 0$ when $u = \frac{\pi}{3} + 2k\pi, k \in \mathbb{Z}$. Thus, there does not exist $\mu > 0$ such that

$$\|\nabla F(u)\|^2 = (u - 2\sin(u))^2(1 - 2\cos(u))^2 \geq 2\mu(F(u) - \min F(u)) = \mu(u - 2\sin(u))^2$$

holds for any $u$. This means $\mathsf{F}(u)$ is not PL in $u$.

### C.2 PL FUNCTIONS DO NOT PRESERVE PL CONDITION UNDER ADDITIVITY IN GENERAL

Let's take a look at a similar example with Example 1 that is joint PL at both levels.

**Example 2.** *With $u \in \mathbb{R}$ and $v \in \mathbb{R}$, consider the following upper and lower-level objectives*

$$f(u, v) = \frac{1}{2}(u - \sin(v))^2 \quad and \quad g(u, v) = \frac{1}{2}(u - v)^2.$$

*We can verify that both $f(u, v)$ and $g(u, v)$ satisfy the joint PL condition in* (4) *and the lower-level problem parameterized by $u$ yields the unique solution $\mathcal{S}(u) = u$.*

The following lemma shows that joint PL upper- and lower-level objectives can lead to non-joint PL penalized objectives.

**Lemma 13.** *In Example 2, the joint PL condition on both levels does not guarantee the joint PL condition of $\mathsf{L}_\gamma(u, v)$; see also the landscape of the penalized function shown in Figure 6.*

**Proof:** For Example 2, as $g^*(u) = 0$ for any $u$, the gradient of $\mathsf{L}_\gamma(u, v)$ can be computed as

$$\nabla \mathsf{L}_\gamma(u, v) = \begin{bmatrix} u - 2\sin(v) + \gamma(u - v) \\ -2(u - 2\sin(v))\cos(v) - \gamma(u - v) \end{bmatrix}.$$

On the other hand, $\mathsf{L}_\gamma(u, v)$ is minimized when $u = v = 0$ with the minimal value 0. For any $\gamma$, there exists $(\bar{u}, \bar{v})$ with $\bar{u} = \frac{2\gamma\pi}{1+\gamma}, \bar{v} = 2\pi$ such that

$$\nabla \mathsf{L}_\gamma(\bar{u}, \bar{v}) = \begin{bmatrix} \bar{u} - 2\sin(\bar{v}) + \gamma(\bar{u} - \bar{v}) \\ -2(\bar{u} - 2\sin(\bar{v}))\cos(\bar{v}) - \gamma(\bar{u} - \bar{v}) \end{bmatrix} = \begin{bmatrix} 0 \\ 0 \end{bmatrix}.$$

However, $\mathsf{L}_\gamma(\bar{u}, \bar{v}) = \frac{1}{2}(\bar{u} - 2\sin(\bar{v}))^2 + \frac{\gamma}{2}(\bar{u} - \bar{v})^2 > 0 = \min_{u,v} \mathsf{L}(\bar{u}, \bar{v})$. This means that saddle points $(\bar{u}, \bar{v})$ exist for any $\gamma$ so that $\mathsf{L}_\gamma(\bar{u}, \bar{v})$ does not satisfy the joint PL condition.

Similarly, the following example shows that blockwise PL conditions on upper and lower-level objectives are also not sufficient to ensure the blockwise PL condition of $\mathsf{L}_\gamma(u, v)$.

**Example 3.** *With $u \in \mathbb{R}$ and $v = [v_1, v_2] \in \mathbb{R}^2$, consider the bilevel problem with the following blockwise PL upper-level objective $f(u,v)$ and the blockwise PL lower-level objective $g(u,v)$, i.e.,*

$$f(u,v) = \frac{1}{2}\left(\frac{u}{2} + v_1 - \sin(v_2)\right)^2 \quad and \quad g(u,v) = \frac{1}{2}(u + v_1 + \sin(v_2))^2$$

*where $g^*(u) = 0$. However, the penalized loss function $\mathsf{L}_\gamma(u,v)$ is not blockwise PL. The landscape of $\mathsf{L}_\gamma(u,v)$ when $v_1 = 2$ is shown in Figure 6.*

**Proof:** The gradients of two objectives in Example 3 can be computed as

$$\nabla f(u,v) = \begin{bmatrix} \frac{1}{2}\left(\frac{u}{2} + v_1 - \sin(v_2)\right) \\ \left(\frac{u}{2} + v_1 - \sin(v_2)\right) \\ -\left(\frac{u}{2} + v_1 - \sin(v_2)\right)\cos(v_2) \end{bmatrix} \quad and \quad \nabla g(u,v) = \begin{bmatrix} (u + v_1 + \sin(v_2)) \\ (u + v_1 + \sin(v_2)) \\ (u + v_1 + \sin(v_2))\cos(v_2) \end{bmatrix}.$$

We first verify that both $f(u,v)$ and $g(u,v)$ are blockwise PL. As $u, v_1 - \sin(v_2), v_1 + \cos(v_2)$ spread out in $\mathbb{R}$, then both $\frac{u}{2} + v_1 - \sin(v_2) = 0$ and $u + v_1 + \sin(v_2) = 0$ always have a solution. Therefore, we have $\min_u f(u,v) = \min_v f(u,v) = \min_u g(u,v) = \min_v g(u,v) = 0$. On the other hand,

$$\|\nabla_u f(u,v)\|^2 = \frac{1}{4}\left(\frac{u}{2} + v_1 - \sin(v_2)\right)^2 \geq \frac{1}{2} \times \left(f(u,v) - \min_u f(u,v)\right)$$

$$\|\nabla_v f(u,v)\|^2 = \left(\frac{u}{2} + v_1 - \sin(v_2)\right)^2 \times (1 + \cos^2(v_2)) \geq 2 \times \left(f(u,v) - \min_v f(u,v)\right)$$

$$\|\nabla_u g(u,v)\|^2 = (u + v_1 + \sin(v_2))^2 = 2 \times \left(g(u,v) - \min_u g(u,v)\right)$$

$$\|\nabla_v g(u,v)\|^2 = (u + v_1 + \sin(v_2))^2 \times (1 + \cos^2(v_2)) \geq 2 \times \left(f(u,v) - \min_v f(u,v)\right)$$

which suggests both $f(u,v)$ and $g(u,v)$ are blockwise PL.

For any $\gamma$, since $g^*(u) = 0$, $\nabla_u \mathsf{L}_\gamma(u,v) = 0$ gives the equation

$$\left(\gamma + \frac{1}{4}\right)u + \left(\gamma + \frac{1}{2}\right)v_1 + \left(\gamma - \frac{1}{2}\right)\sin(v_2) = 0. \tag{25}$$

It can be verified that both $u = v_1 = v_2 = 0$ and $u = -\frac{4 + 8\gamma}{4\gamma + 1}, v_1 = 2, v_2 = 0$ are solutions to (25), which yields two different objective values $\mathsf{L}_\gamma(u,v) = 0$ and $\mathsf{L}_\gamma(u,v) = \frac{8\gamma^2 + 2}{(4\gamma + 1)^2}$. This means there exists saddle points for $\mathsf{L}_\gamma(u,v)$ so that $\mathsf{L}_\gamma(u,v)$ is not blockwise PL over $u$. Same phenomenon happens for $v$, suggesting that $\mathsf{L}_\gamma(u,v)$ is also not blockwise PL over $u$.

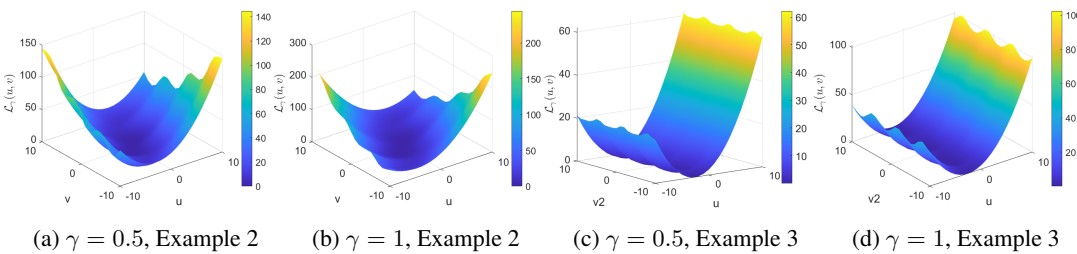

(a) $\gamma = 0.5$, Example 2     (b) $\gamma = 1$, Example 2     (c) $\gamma = 0.5$, Example 3     (d) $\gamma = 1$, Example 3

Figure 6: Visualization of $\mathsf{L}_\gamma(u,v)$ in Example 2 and 3. Saddle points exist for $\mathsf{L}_\gamma(u,v)$ in Example 1 and Example 3, suggesting that penalized objective does not satisfy the PL condition.

### C.3 NONCONVEXITY OF $\mathsf{F}(u)$

**Example 4.** *With $u \in \mathbb{R}$ and $v \in \mathbb{R}$, consider the following upper and lower-level objectives*

$$f(u,v) = v \quad and \quad g(u,v) = e^v + (u + v)^4.$$

*It is obvious that both $f(u,v)$ and $g(u,v)$ are jointly convex with regard to $(u,v)$ and minimizing the lower-level problem parameterized by $u$ yields the unique solution $\mathcal{S}(u) = \{v \mid 4(u + v)^3 + e^v = 0\}$. However, the overall bilevel loss function $\mathsf{F}(u) = f(u, \mathcal{S}(u)) = \mathcal{S}(u)$ is not convex over $u$. A visualization of $\mathsf{F}(u) = \mathcal{S}(u)$ is shown in Figure 5.*

**Proof:** Let us calculate the gradient and Hessian of $f(u,v)$ and $g(u,v)$ in Example 4 by

$$\nabla f(u,v) = \begin{bmatrix} 0 \\ 1 \end{bmatrix} \quad \text{and} \quad \nabla g(u,v) = \begin{bmatrix} 4(u+v)^3 \\ 4(u+v)^3 + \mathrm{e}^v \end{bmatrix}$$

$$\nabla^2 f(u,v) = \begin{bmatrix} 0 & 0 \\ 0 & 0 \end{bmatrix} \quad \text{and} \quad \nabla^2 g(u,v) = \begin{bmatrix} 12(u+v)^2 & 12(u+v)^2 \\ 12(u+v)^2 & 12(u+v)^2 + \mathrm{e}^v \end{bmatrix}.$$

From the Hessian, both $f(u,v)$ and $g(u,v)$ are convex because the determinants of all $k \times k$ submatrix are nonnegative. Besides, the lower-level optimal solution set is obtained by letting $4(u+v)^3 + \mathrm{e}^v = 0$. Since when fixing $u$, $4(u+v)^3 + \mathrm{e}^v$ is strictly increasing with respect to $v$ and spread out over $(-\infty, \infty)$, the solution set $\mathcal{S}(u) = \{v \mid 4(u+v)^3 + \mathrm{e}^v = 0\}$ is unique and nonempty.

We will then prove $\mathsf{F}(u)$ is not convex by contradiction. We choose $u = \sqrt[3]{-\frac{1}{4}}$ and $\tilde{u} = \sqrt[3]{-\frac{\mathrm{e}}{4}} - 1$, then $\mathcal{S}(u) = \{v \mid 4(u+v)^3 + \mathrm{e}^v = 0\} = 0$ and $\mathcal{S}(\tilde{u}) = \{v \mid 4(\tilde{u}+v)^3 + \mathrm{e}^v = 0\} = 1$, so that

$$\mathsf{F}(u) = f(\sqrt[3]{-\frac{1}{4}}, 0) = 0 \quad \text{and} \quad F(\tilde{u}) = f(\sqrt[3]{-\frac{\mathrm{e}}{4}} - 1, 1) = 1.$$

Assume that $\mathsf{F}(u)$ is convex, and thus the function value $\mathsf{F}(\bar{u})$ at the medium point $\bar{u} = \frac{u+\tilde{u}}{2} = \frac{\sqrt[3]{-\frac{1}{4}} + \sqrt[3]{-\frac{\mathrm{e}}{4}} - 1}{2}$ satisfies $\mathsf{F}(\bar{u}) \leq \frac{F(u)+F(\tilde{u})}{2} = \frac{1}{2}$, which means there exists $v \leq \frac{1}{2}$ such that $4(\bar{u}+v)^3 + \mathrm{e}^v = 0$. However, $4(\bar{u}+v)^3 + \mathrm{e}^v$ is strictly increasing and

$$4(\bar{u}+v)^3 + \mathrm{e}^v \mid_{v=1/2} = 4 \left( \frac{\sqrt[3]{-\frac{1}{4}} + \sqrt[3]{-\frac{\mathrm{e}}{4}} - 1}{2} + \frac{1}{2} \right)^3 + \mathrm{e}^{\frac{1}{2}} \approx -0.07 < 0.$$

So for any $v \leq \frac{1}{2}$, $4(\bar{u}+v)^3 + \mathrm{e}^v < 0$, which yields a contradiction, and thus, $\mathsf{F}(u)$ is not convex.

## D   PROOF OF THEOREM 1

**Lemma 14** (Gradient bias). *Under Assumption 1, denoting $d_u^k = (u^{k+1} - u^k)/\alpha$, we have*

$$\|d_u^k - \nabla_u \mathsf{L}_\gamma(u^k, v^k)\|^2 \leq \frac{2\gamma^2 \ell_g^2}{\mu_g} (1 - \beta\mu_g)^{T_k} (g(u^k, w^0) - g^*(u^k)). \tag{26}$$

**Proof:** First, according to Lemma 8, $\nabla g^*(u) = \nabla_u g(u,v), \forall v \in \mathcal{S}(u)$. By the update rule, the bias comes from the inexactness of the value function's gradient, i.e.

$$\|d_u^k - \nabla_u \mathsf{L}_\gamma(u^k, v^k)\| = \gamma \|\nabla g^*(u^k) - \nabla g(u^k, w^{k+1})\| \leq \gamma \ell_g d(w^{k+1}, \mathcal{S}(u^k)). \tag{27}$$

According to Lemma 7 and Lemma 6, we have

$$d(w^{k+1}, \mathcal{S}(u^k))^2 \leq \frac{2}{\mu_g} (1 - \beta\mu_g)^{T_k} (g(u^k, w^0) - g^*(u^k))$$

from which plugging into (27) yields the conclusion.

### D.1   PROOF UNDER THE JOINT PL CONDITION

From Lemma 8, we have the smoothness of $g^*(u)$ and thus $\mathsf{L}_\gamma(u,v)$. Therefore, by defining $d_u^k = (u^{k+1} - u^k)/\alpha$ and by Taylor expansion, we have

$$\mathsf{L}_\gamma(u^{k+1}, v^{k+1}) \leq \mathsf{L}_\gamma(u^k, v^k) + \langle \nabla_u \mathsf{L}_\gamma(u^k, v^k), u^{k+1} - u^k \rangle + \langle \nabla_v \mathsf{L}_\gamma(u^k, v^k), v^{k+1} - v^k \rangle$$

$$+ \frac{\ell_f + \gamma(\ell_g + L_g)}{2} \|u^{k+1} - u^k\|^2 + \frac{\ell_f + \gamma(\ell_g + L_g)}{2} \|v^{k+1} - v^k\|^2$$

$$\leq \mathsf{L}_\gamma(u^k, v^k) - \alpha \langle \nabla_u \mathsf{L}_\gamma(u^k, v^k), d_u^k \rangle - \left( \alpha - \frac{(\ell_f + \gamma(\ell_g + L_g))\alpha^2}{2} \right) \|\nabla_v \mathsf{L}_\gamma(u^k, v^k)\|^2$$

$$+ \frac{(\ell_f + \gamma(\ell_g + L_g))\alpha^2}{2}\|d_u^k\|^2$$

$$= \mathsf{L}_\gamma(u^k, v^k) - \frac{\alpha}{2}\|\nabla_u \mathsf{L}_\gamma(u^k, v^k)\|^2 - \left(\frac{\alpha}{2} - \frac{(\ell_f + \gamma(\ell_g + L_g))\alpha^2}{2}\right)\|d_u^k\|^2$$

$$- \left(\alpha - \frac{(\ell_f + \gamma(\ell_g + L_g))\alpha^2}{2}\right)\|\nabla_v \mathsf{L}_\gamma(u^k, v^k)\|^2 + \frac{\alpha}{2}\|d_u^k - \nabla_u \mathsf{L}_\gamma(u^k, v^k)\|^2$$

$$\overset{(a)}{\le} \mathsf{L}_\gamma(u^k, v^k) - \frac{\alpha}{2}\|\nabla_u \mathsf{L}_\gamma(u^k, v^k)\|^2 - \frac{\alpha}{2}\|\nabla_v \mathsf{L}_\gamma(u^k, v^k)\|^2$$

$$+ \frac{\alpha\gamma^2\ell_g^2}{\mu_g}(1 - \beta\mu_g)^{T_k}(g(u^k, w^0) - g^*(u^k))$$

$$= \mathsf{L}_\gamma(u^k, v^k) - \frac{\alpha}{2}\|\nabla \mathsf{L}_\gamma(u^k, v^k)\|^2 + \frac{\alpha\gamma^2\ell_g^2}{\mu_g}(1 - \beta\mu_g)^{T_k}(g(u^k, w^0) - g^*(u^k))$$

$$\overset{(b)}{\le} \mathsf{L}_\gamma(u^k, v^k) - \alpha\mu_l(\mathsf{L}_\gamma(u^k, v^k) - \min_{u,v}\mathsf{L}_\gamma(u, v)) + \frac{\alpha\gamma^2\ell_g^2}{\mu_g}(1 - \beta\mu_g)^{T_k}(g(u^k, w^0) - g^*(u^k))$$

where (a) is because $\alpha \le \frac{1}{\ell_f + \gamma(\ell_g + L_g)}$ and Lemma 14, and (b) is because of the joint PL condition. Subtracting both sides by $\min_{u,v}\mathsf{L}_\gamma(u, v)$ yields

$$\mathsf{L}_\gamma(u^{k+1}, v^{k+1}) - \min_{u,v}\mathsf{L}_\gamma(u, v) \le (1 - \alpha\mu_l)(\mathsf{L}_\gamma(u^k, v^k) - \min_{u,v}\mathsf{L}_\gamma(u, v))$$

$$+ \frac{\alpha\gamma^2\ell_g^2}{\mu_g}(1 - \beta\mu_g)^{T_k}(g(u^k, w^0) - g^*(u^k)).$$

Letting $T_k = \log\left(\frac{\gamma^2(g(u^k, w^0) - g^*(u^k))}{\epsilon}\right)$ and telescoping the above inequality yields

$$\mathsf{L}_\gamma(u^K, v^K) - \min_{u,v}\mathsf{L}_\gamma(u, v) \le (1 - \alpha\mu_l)^K(\mathsf{L}_\gamma(u^0, v^0) - \min_{u,v}\mathsf{L}_\gamma(u, v)) + \sum_{k=1}^{K-1}(1 - \alpha\mu_l)^\kappa \mathcal{O}(\alpha\epsilon)$$

$$\le (1 - \alpha\mu_l)^K(\mathsf{L}_\gamma(u^0, v^0) - \min_{u,v}\mathsf{L}_\gamma(u, v)) + \mathcal{O}(\epsilon).$$

On the other hand, according to (Shen et al., 2023, Theorem 1) and denoting $\epsilon_\gamma := g(u^K, v^K) - g^*(u^K)$, we have for any $\epsilon_1$ and $\gamma \ge 2\gamma^* = \frac{\ell_{f,0}^2\mu_g}{4}\epsilon_1^{-1}$, the following

$$\epsilon_\gamma \le \frac{\epsilon + \epsilon_1}{\gamma - \gamma^*} \le \frac{8\epsilon_1(\epsilon + \epsilon_1)}{\ell_{f,0}^2\mu_g} \quad \text{and} \quad f(u^K, v^K) - f(u, v) \le (1 - \alpha\mu_l)^K(\mathsf{L}_\gamma(u^0, v^0) - \min_{u,v}\mathsf{L}_\gamma(u, v)) + \mathcal{O}(\epsilon)$$

holds for any $(u, v)$ satisfying $g(u, v) - g^*(u) \le \epsilon_\gamma$. The choice of $\epsilon_1 = \mathcal{O}(\sqrt{\epsilon})$ gives $\epsilon_\gamma = \mathcal{O}(\epsilon)$ and $\gamma = \mathcal{O}(\epsilon^{-0.5})$, which completes the proof.

### D.2 PROOF UNDER THE BLOCKWISE PL CONDITION

Define $\mathsf{L}_\gamma^*(u) = \min_v \mathsf{L}_\gamma(u, v)$ and $\mathcal{S}_\gamma(u) = \arg\min_v \mathsf{L}_\gamma(u, v)$. According to Lemma 8, $\nabla g^*(u) = \nabla_u g(u, v), \forall v \in \mathcal{S}(u)$. Since $\mathsf{L}_\gamma(u, v)$ is $(\ell_f + \gamma\ell_g)$ smooth over $v$, then according to Lemma 8, we also have $\nabla\mathsf{L}_\gamma^*(u) = \nabla_u\mathsf{L}_\gamma(u, v), \forall v \in \mathcal{S}_\gamma(u)$. Moreover, $\mathsf{L}_\gamma^*(u)$ is $L_\gamma$ smooth with $L_\gamma := (\ell_f + \gamma\ell_g)(1 + (\ell_f + \gamma\ell_g)/2\mu_v) + L_g$. Also $\mathsf{L}_\gamma^*(u)$ is PL with $\mu_u$ because $\mathsf{L}_\gamma(u, v)$ is blockwise PL over $u$.

If we denote $d_u^k = (u^{k+1} - u^k)/\alpha$, then by the update rule, the bias comes from the inexactness of the value function's gradient, i.e.

$$\|d_u^k - \nabla\mathsf{L}_\gamma^*(u^k)\| = \|\nabla_u f(u^k, v^{k+1}) - \nabla_u f(u^k, v_\gamma)\| + \gamma\|\nabla g(u^k, v^{k+1}) - \nabla g(u^k, v_\gamma)\|$$

$$+ \gamma\|\nabla g(u^k, w^{k+1}) - \nabla g(u^k, v)\|$$

$$\le (\ell_f + \gamma\ell_g)d(v^{k+1}, \mathcal{S}_\gamma(u^k)) + \gamma\ell_g d(w^{k+1}, \mathcal{S}(u^k))$$

where $v_\gamma \in \mathcal{S}_\gamma(u^k), v \in \mathcal{S}(u^k)$.

According to Lemma 6 and Lemma 7, we have

$$d(w^{k+1}, \mathcal{S}(u^k))^2 \leq \frac{2}{\mu_g}(1 - \beta\mu_g)^{T_k}(g(u^k, w^0) - g^*(u^k))$$

$$d(v^{k+1}, \mathcal{S}_\gamma(u^k))^2 \leq \frac{2}{\mu_v}(1 - \tilde{\beta}\mu_v)^{T_k}(\mathsf{L}_\gamma(u^k, w^0) - \mathsf{L}_\gamma^*(u^k)).$$

As a result, the bias of the gradient estimator can be bounded by

$$\|d_u^k - \nabla\mathsf{L}_\gamma^*(u^k)\|^2 \leq \frac{4(\ell_f + \gamma\ell_g)^2}{\mu_v}(1 - \tilde{\beta}\mu_v)^{T_k}(\mathsf{L}_\gamma(u^k, w^0) - \mathsf{L}_\gamma^*(u^k))$$

$$+ \frac{4\gamma^2\ell_g^2}{\mu_g}(1 - \beta\mu_g)^{T_k}(g(u^k, w^0) - g^*(u^k)). \tag{28}$$

Therefore, by Taylor expansion and the smoothness of $\mathsf{L}_\gamma^*(u)$, we have

$$\mathsf{L}_\gamma^*(u^{k+1}) \leq \mathsf{L}_\gamma^*(u^k) + \langle\nabla\mathsf{L}_\gamma^*(u^k), u^{k+1} - u^k\rangle + \frac{L_\gamma}{2}\|u^{k+1} - u^k\|^2$$

$$\leq \mathsf{L}_\gamma^*(u^k) - \alpha\langle\nabla\mathsf{L}_\gamma^*(u^k), d_u^k\rangle + \frac{L_\gamma\alpha^2}{2}\|d_u^k\|^2$$

$$= \mathsf{L}_\gamma^*(u^k) - \frac{\alpha}{2}\|\nabla\mathsf{L}_\gamma^*(u^k)\|^2 - \frac{\alpha}{2}\|d_u^k\|^2 + \frac{\alpha}{2}\|\nabla\mathsf{L}_\gamma^*(u^k) - d_u^k\|^2 + \frac{L_\gamma\alpha^2}{2}\|d_u^k\|^2$$

$$\overset{(a)}{\leq} \mathsf{L}_\gamma^*(u^k) - \frac{\alpha}{2}\|\nabla\mathsf{L}_\gamma^*(u^k)\|^2 + \frac{2\alpha(\ell_f + \gamma\ell_g)^2}{\mu_v}(1 - \tilde{\beta}\mu_v)^{T_k}(\mathsf{L}_\gamma(u^k, w^0) - \mathsf{L}_\gamma^*(u^k))$$

$$+ \frac{2\alpha\gamma^2\ell_g^2}{\mu_g}(1 - \beta\mu_g)^{T_k}(g(u^k, w^0) - g^*(u^k))$$

$$\overset{(b)}{\leq} \mathsf{L}_\gamma^*(u^k) - \alpha\mu_u(\mathsf{L}_\gamma^*(u) - \min_{u,v}\mathsf{L}_\gamma(u, v)) + \frac{2\alpha(\ell_f + \gamma\ell_g)^2}{\mu_v}(1 - \tilde{\beta}\mu_v)^{T_k}(\mathsf{L}_\gamma(u^k, w^0) - \mathsf{L}_\gamma^*(u^k))$$

$$+ \frac{2\alpha\gamma^2\ell_g^2}{\mu_g}(1 - \beta\mu_g)^{T_k}(g(u^k, w^0) - g^*(u^k))$$

where (a) is because $\alpha \leq \frac{1}{L_\gamma}$ and Lemma 14, and (b) is because of the PL condition of $\mathsf{L}_\gamma^*(u)$.
Subtracting both sides by $\min_{u,v}\mathsf{L}_\gamma(u, v)$ yields

$$\mathsf{L}_\gamma^*(u^{k+1}) - \min_{u,v}\mathsf{L}_\gamma(u, v) \leq (1 - \alpha\mu_u)(\mathsf{L}_\gamma^*(u^k) - \min_{u,v}\mathsf{L}_\gamma(u, v))$$

$$+ \frac{2\alpha(\ell_f + \gamma\ell_g)^2}{\mu_v}(1 - \tilde{\beta}\mu_v)^{T_k}(\mathsf{L}_\gamma(u^k, w^0) - \mathsf{L}_\gamma^*(u^k))$$

$$+ \frac{2\alpha\gamma^2\ell_g^2}{\mu_g}(1 - \beta\mu_g)^{T_k}(g(u^k, w^0) - g^*(u^k)).$$

When $\mu_v = \mathcal{O}(\gamma)$, the last term is dominating over the second term. Letting $T_k = \log\left(\frac{\gamma^2(g(u^k, w^0) - g^*(u^k))}{\epsilon}\right)$ and telescoping the above inequality yields

$$\mathsf{L}_\gamma^*(u^{k+1}) - \min_{u,v}\mathsf{L}_\gamma(u, v) \leq (1 - \alpha\mu_u)^k(\mathsf{L}_\gamma^*(u^k) - \min_{u,v}\mathsf{L}_\gamma(u, v)) + \mathcal{O}(\epsilon)$$

Combining with the results of

$$\mathsf{L}_\gamma(u^k, v^{k+1}) - \mathsf{L}_\gamma^*(u^k) \leq \mathcal{O}(\epsilon) \text{ and } \mathsf{L}_\gamma(u^{k+1}, v^{k+2}) - \mathsf{L}_\gamma^*(u^{k+1}) \leq \mathcal{O}(\epsilon)$$

we know that

$$\mathsf{L}_\gamma(u^K, v^{K+1}) - \min_{u,v}\mathsf{L}_\gamma(u, v) \leq (1 - \alpha\mu_u)^K(\mathsf{L}_\gamma(u^0, v^1) - \min_{u,v}\mathsf{L}_\gamma(u, v)) + \mathcal{O}(\epsilon)$$

On the other hand, according to (Shen et al., 2023, Theorem 1) and denoting $\epsilon_\gamma := g(u^K, v^K) - g^*(u^K)$, we have for any $\epsilon_1$ and $\gamma \geq 2\gamma^* = \frac{\ell_{f,0}^2\mu_g}{4}\epsilon_1^{-1}$, the following

$$\epsilon_\gamma \leq \frac{\epsilon + \epsilon_1}{\gamma - \gamma^*} \leq \frac{8\epsilon_1(\epsilon + \epsilon_1)}{\ell_{f,0}^2\mu_g} \text{ and } f(u^K, v^K) - f(u, v) \leq (1 - \alpha\mu_l)^K(\mathsf{L}_\gamma(u^0, v^0) - \min_{u,v}\mathsf{L}_\gamma(u, v)) + \mathcal{O}(\epsilon)$$

holds for any $(u, v)$ satisfying $g(u, v) - g^*(u) \leq \epsilon_\gamma$. The choice of $\epsilon_1 = \mathcal{O}(\sqrt{\epsilon})$ gives $\epsilon_\gamma = \mathcal{O}(\epsilon)$ and $\gamma = \mathcal{O}(\epsilon^{-0.5})$, which completes the proof.

# E  PROOF OF OBSERVATIONS 1-2

## E.1  PROOF OF OBSERVATION 1

**Proof:** The blockwise PL condition of $\gamma(g(u,v) - g^*(u))$ over $v$ is obvious under Assumption 1 because $g^*(u)$ is independent of $v$. On the other hand, with Assumption 1, the gradient of $\gamma(g(u,v) - g^*(u))$ can be lower bounded by

$$
\begin{aligned}
\|\nabla\gamma(g(u,v) - g^*(u))\|^2 &= \gamma^2\|\nabla_u g(u,v) - \nabla_u g^*(u)\|^2 + \gamma^2\|\nabla_v g(u,v)\|^2 \\
&\geq \gamma^2\|\nabla_v g(u,v)\|^2 \\
&\overset{(a)}{\geq} \gamma\mu_g(\gamma(g(u,v) - g^*(u))) \\
&\overset{(b)}{=} \gamma\mu_g(\gamma(g(u,v) - g^*(u)) - \min_{u,v}\gamma(g(u,v) - g^*(u)))
\end{aligned}
$$

where (a) is because $g(u,v)$ is $\mu_g$ PL in $v$ and (b) is due to $\min_{u,v}\gamma(g(u,v) - g^*(u)) = 0$.

## E.2  PROOF OF OBSERVATION 2

The result that $h_1(Az)$ and $h_2(Bz)$ satisfy the PL condition is well-known; see e.g., (Karimi et al., 2016, Appendix B). We will show $h_1(Az) + h_2(Bz)$ satisfies the PL inequality. For arbitrary $z_1, z_2$, we denote $u_1 = Az_1, u_2 = Az_2, v_1 = Bz_1, v_2 = Bz_2$. By the strong convexity of $h_1$ and $h_2$, we have

$$
h_1(Az_2) \geq h_1(Az_1) + \nabla h_1(u_1)^\top(Az_2 - Az_1) + \frac{\mu_1}{2}\|Az_1 - Az_2\|^2
$$

$$
h_2(Bz_2) \geq h_2(Bz_1) + \nabla h_2(v_1)^\top(Bz_2 - Bz_1) + \frac{\mu_2}{2}\|Bz_1 - Bz_2\|^2.
$$

By the fact of $\nabla h_1(Az_i) = A^\top\nabla h_1(u_i)$ and $\nabla h_2(Bz_i) = B^\top\nabla h_2(v_i)$ for $i = 1, 2$, we get

$$
h_1(Az_2) \geq h_1(Az_1) + \nabla h_1(Az_1)^\top(z_2 - z_1) + \frac{\mu_1}{2}\|Az_1 - Az_2\|^2 \tag{29a}
$$

$$
h_2(Bz_2) \geq h_2(Bz_1) + \nabla h_2(Bz_1)^\top(z_2 - z_1) + \frac{\mu_2}{2}\|Bz_1 - Bz_2\|^2 \tag{29b}
$$

Letting $z_2 = \text{Proj}_{\mathcal{Z}^*}(z_1)$ be the projection of $z_1$ onto the optimal solution set $\mathcal{Z}^* = \arg\min_z h_1(Az) + h_2(Bz)$. Clearly, $z_2$ is the solution of

$$
\min_z \frac{1}{2}\|z - z_1\|^2, \text{ s.t. } Az = Az^*, Bz = Bz^* \tag{30}
$$

where $z^* \in \arg\min_z h_1(Az) + h_2(Bz)$ is an arbitrary optimal point. Strong duality holds for (30) so that KKT conditions must hold at $z_2$, i.e. there exists $\lambda_1 \in \mathbb{R}^m, \lambda_2 \in \mathbb{R}^n$ such that

$$
z_2 - z_1 + A^\top\lambda_1 + B^\top\lambda_2 = 0. \tag{31}
$$

Without loss of generality, we can consider only two cases: i) $\text{Ran}(A^\top) = \text{Ran}(B^\top)$; and, ii) $\text{Ran}(A^\top) \perp \text{Ran}(B^\top)$. Otherwise, we can decompose $\text{Ran}(B^\top)$ into the direct sum of two orthogonal subspaces: one parallel and the other orthogonal to $\text{Ran}(A^\top)$.

For Case i), it is clear that $z_2 - z_1 \in \text{Ran}(A^\top)$ and $z_2 - z_1 \in \text{Ran}(B^\top)$ by (31). Therefore, letting $C' = \mu_1\sigma_*^2(A) + \mu_2\sigma_*^2(B)$ and adding (29a) and (29b) up yields

$$
\begin{aligned}
h_1(Az_2) + h_2(Bz_2) &\geq h_1(Az_1) + h_2(Bz_1) + (\nabla h_1(Az_1) + \nabla h_2(Bz_1))^\top(z_2 - z_1) + \frac{C'}{2}\|z_1 - z_2\|^2 \\
&\geq h_1(Az_1) + h_2(Bz_1) + \min_z\left[(\nabla h_1(Az_1) + \nabla h_2(Bz_1))^\top(z - z_1) + \frac{C'}{2}\|z - z_1\|^2\right] \\
&= h_1(Az_1) + h_2(Bz_1) - \frac{1}{C'}\|\nabla h_1(Az_1) + \nabla h_2(Bz_1)\|^2. \tag{32}
\end{aligned}
$$

For Case ii), since $A^\top\lambda_1, B^\top\lambda_2 \in \text{Ran}(A^\top) \cup \text{Ran}(B^\top)$ by (31), we have

$$
z_1 - z_2 = A^\top\lambda_1 + B^\top\lambda_2 \in \text{Span}(\text{Ran}(A^\top) \cup \text{Ran}(B^\top)).
$$

By definition, there is a finite number of orthogonal $v_i \in \text{Ran}(A^\top) \cup \text{Ran}(B^\top)$ and of $a_i \in \mathbb{R}$ so that $z_1 - z_2 = a_1 v_1 + \cdots + a_k v_k$. As each $v_i$ is either in $\text{Ran}(A^\top)$ or $\text{Ran}(B^\top)$ and they are orthogonal, we have $\|z_1 - z_2\|^2 = \sum_{i=1}^{k} \|a_i v_i\|^2$ and we can focus on each of $v_i$ independently.

Let $C = \min\{\mu_1 \sigma_*^2(A), \mu_2 \sigma_*^2(B)\}$, since $v_i \in \text{Ran}(A^\top) \cup \text{Ran}(B^\top)$, we have that for each $v_i$, either $\|A a_i v_i\|^2 \geq \frac{C}{2}\|a_i v_i\|^2$ or $\|B a_i v_i\|^2 \geq \frac{C}{2}\|a_i v_i\|^2$. Therefore, adding (29a) and (29b) yields

$$h_1(Az_2) + h_2(Bz_2) \geq h_1(Az_1) + h_2(Bz_1) + (\nabla h_1(Az_1) + \nabla h_2(Bz_1))^\top (z_2 - z_1) + \frac{C}{2}\|z_1 - z_2\|^2$$

$$\geq h_1(Az_1) + h_2(Bz_1) + \min_z \left[ (\nabla h_1(Az_1) + \nabla h_2(Bz_1))^\top (z - z_1) + \frac{C}{2}\|z - z_1\|^2 \right]$$

$$= h_1(Az_1) + h_2(Bz_1) - \frac{1}{C}\|\nabla h_1(Az_1) + \nabla h_2(Bz_1)\|^2. \tag{33}$$

According to (32) and (33), as $h_1(Az_2) + h_2(Bz_2) = \min_z h_1(Az) + h_2(Bz)$ and $C \leq C'$, this completes the proof.

## F    PROOF FOR REPRESENTATION LEARNING

In this section, we provide the proof of lemmas and theorems omitted for representation learning.

### F.1    SUFFICIENT CONDITIONS FOR ASSUMPTION 2

Assumption 2 ensures that the bilevel problem has at least one full-rank solution $W_1^*$. Note that the full rank solution $W_1^*$ always exists for the single training or validation problem because the objective value is solely dependent on the product $W_1 W_2$. For any solution pair $(W_1, W_2)$, it is possible to transform $W_1$ into a full rank matrix $W_1^*$ by altering its kernel space components and simultaneously, one can identify a corresponding matrix $W_2^*$ that mirrors $W_2$ within the range space of $W_1$ and is nullified within the kernel space of $W_1$, such that $W_1 W_2 = W_1^* W_2^*$. In this sense, Assumption 2 quantifies the singularity of solving two non-singular problems together. To get more sense of Assumption 2, we present some *sufficient conditions* for it.

**Lemma 15.** *Assumption 2 holds if one of the following conditions holds.*

*(a)* $\exists (W_1^*, W_2^*) \in \arg\min_{W_1, W_2} \mathsf{L}_{\text{val}}(W_1, W_2)$ *s.t.* $(W_1^*, W_2^*) \in \arg\min_{W_1, W_2} \mathsf{L}_{\text{trn}}(W_1, W_2)$.

*(b)* *The concatenated data matrix* $[X_{\text{val}}; X_{\text{trn}}]$ *is of full row rank.*

**Proof:** The condition in (a) indicates that $(W_1^*, W_2^*)$ is a minimizer of the validation loss. Therefore, for any $W_1, W_2$, we have $\mathsf{L}_{\text{val}}(W_1^*, W_2^*) \leq \mathsf{L}_{\text{val}}(W_1, W_2)$. Furthermore, we are given any $\epsilon_1, \epsilon_2$. Then for any $W_2$ in the $\epsilon_2$ lower-level optimal set, $\mathsf{L}_{\text{val}}(W_1^*, W_2^*) \leq \mathsf{L}_{\text{val}}(W_1, W_2) \leq \mathsf{L}_{\text{val}}(W_1, W_2) + \epsilon_1$ holds. Together with the $\epsilon_2$ feasibility of $(W_1^*, W_2^*)$ to the bilevel problem, it means $(W_1^*, W_2^*)$ is an $(\epsilon_1, \epsilon_2)$ solution to the bilevel problem. Moreover, since $(W_1^*, W_2^*)$ is also a minimizer of the training loss, we have

$$\mathsf{L}_{\text{trn}}^*(W_1^*) = \mathsf{L}_{\text{trn}}(W_1^*, W_2^*) = \min_{W_1, W_2} \mathsf{L}_{\text{trn}}(W_1, W_2) \overset{\text{Lemma 5}}{=\!=\!=} 0 \leq \epsilon_2$$

which verifies Assumption 2.

For the condition in (b), it implies there exists

$$(W_1^*, W_2^*) \in \arg\min_{W_1, W_2} (\mathsf{L}_{\text{val}}(W_1, W_2) + \mathsf{L}_{\text{trn}}(W_1, W_2))$$

such that $\mathsf{L}_{\text{val}}(W_1^*, W_2^*) + \mathsf{L}_{\text{trn}}(W_1^*, W_2^*) = 0$. Due to the nonnegativeness of both validation and training loss, it must hold that $\mathsf{L}_{\text{val}}(W_1^*, W_2^*) = \mathsf{L}_{\text{trn}}(W_1^*, W_2^*) = 0$. This suggests $(W_1^*, W_2^*)$ is the joint minimizer of both training and validation loss, which satisfies condition (b), and thus, Assumption 2 is verified.

**Remark 2.** *Condition (a) says that the training and validation problem share at least a global solution, situating us within the interpolating regime between training and validation. This means*

*that minimizing the combined objective leads to a solution that simultaneously minimizes both problems (Fan et al., 2023). In this case, the model trained from representation learning matches the performance attained by optimizing both layers of model directly based on validation loss. **Condition (b)** is a sufficient condition of **Condition (a)** and **Condition (b)** itself means there are no redundant information contained in the training and validation set.*

### F.2 Landscape of Nested Bilevel Objective $\mathsf{F}(W_1)$

According to the loss definition and the knowledge in linear algebra, we can derive the explicit form of the bilevel objective in representation learning as shown in the following lemma.

**Lemma 16** (Analytical form of $\mathsf{F}(W_1)$ and its nonconvexity). *The bilevel objective for two layer linear neural network representation learning in* (8) *can be explicitly written as*

$$F(W_1) = \frac{1}{2} \left\| A - (X_{\mathrm{val}} W_1) \left( I - (X_{\mathrm{trn}} W_1)^\dagger (X_{\mathrm{trn}} W_1) \right) b \right\|^2 \tag{34}$$

*with* $A = Y_{\mathrm{val}} - X_{\mathrm{val}} W_1 (X_{\mathrm{trn}} W_1)^\dagger Y_{\mathrm{trn}}$ *and* $b = \left[ (X_{\mathrm{val}} W_1) \left( I - (X_{\mathrm{trn}} W_1)^\dagger (X_{\mathrm{trn}} W_1) \right) \right]^\dagger A$. *Moreover,* $\mathsf{F}(W_1)$ *is nonconvex even when* $X_{\mathrm{trn}} = X_{\mathrm{val}} = Y_{\mathrm{trn}} = Y_{\mathrm{val}} = I$.

When $X_{\mathrm{trn}} = X_{\mathrm{val}} = Y_{\mathrm{trn}} = Y_{\mathrm{val}} = I$, it is obvious that any invertible matrix $W_1$ and its inverse $W_2 = W_1^{-1}$ are the solutions to the bilevel problem (1) as both upper and lower-level objective are the same. In this sense, the above lemma indicates that the landscape of the bilevel objective is not convex even in the trivial case. A possible explanation to this phenomenon is the discontinuity and nonconvexity of the inverse operator $W_1^\dagger$ at the rank-deficient point.

**Proof:** First, according to (Barata & Hussein, 2012, Theorem 6.1), the lower-level solution set of (8) can be represented by

$$\mathcal{S}(W_1) = \left\{ (X_{\mathrm{trn}} W_1)^\dagger Y_{\mathrm{trn}} + \left( I - (X_{\mathrm{trn}} W_1)^\dagger (X_{\mathrm{trn}} W_1) \right) b, \ \forall b \right\} = (X_{\mathrm{trn}} W_1)^\dagger Y_{\mathrm{trn}} + \mathrm{Ker}(X_{\mathrm{trn}} W_1). \tag{35}$$

Then $\forall W_1$, we aim to solve the following problem

$$\min_b \frac{1}{2} \left\| A - (X_{\mathrm{val}} W_1) \left( I - (X_{\mathrm{trn}} W_1)^\dagger (X_{\mathrm{trn}} W_1) \right) b \right\|^2 \tag{36}$$

where $A = Y_{\mathrm{val}} - X_{\mathrm{val}} W_1 (X_{\mathrm{trn}} W_1)^\dagger Y_{\mathrm{trn}}$ is fixed when $W_1$ is given. Applying (Barata & Hussein, 2012, Theorem 6.1) to (36) again, we know (36) achieves minimal when

$$b = \left[ (X_{\mathrm{val}} W_1) \left( I - (X_{\mathrm{trn}} W_1)^\dagger (X_{\mathrm{trn}} W_1) \right) \right]^\dagger A.$$

When $X_{\mathrm{trn}} = X_{\mathrm{val}} = Y_{\mathrm{trn}} = Y_{\mathrm{val}} = I$, we have

$$b = \left[ W_1 \left( I - W_1^\dagger W_1 \right) \right]^\dagger A = 0, \quad \text{and} \quad A = Y_{\mathrm{val}} - X_{\mathrm{val}} W_1 (X_{\mathrm{trn}} W_1)^\dagger Y_{\mathrm{trn}} = I - W_1 W_1^\dagger$$

Therefore, the bilevel objective has the form of $\mathsf{F}(W_1) = \frac{1}{2} \|A\|^2 = \frac{1}{2} \|I - W_1 W_1^\dagger\|^2$. Without loose of generality, we consider $W_1 \in \mathbb{R}^{2 \times 2}$ and the objective value at two point

$$F\left( \begin{pmatrix} 0 & 0 \\ 0 & 1 \end{pmatrix} \right) = \frac{1}{2}, \quad F\left( \begin{pmatrix} 2 & 0 \\ 0 & -1 \end{pmatrix} \right) = 0$$

However, the function value at the median point does not satisfy the convexity condition, i.e.

$$F\left( \frac{1}{2} \times \begin{pmatrix} 0 & 0 \\ 0 & 1 \end{pmatrix} + \frac{1}{2} \times \begin{pmatrix} 2 & 0 \\ 0 & -1 \end{pmatrix} \right) = F\left( \begin{pmatrix} 1 & 0 \\ 0 & 0 \end{pmatrix} \right) = \frac{1}{2} > \frac{1}{2} \times \frac{1}{2} + \frac{1}{2} \times 0 = \frac{1}{4}.$$

As a result, $\mathsf{F}(W_1)$ is not convex.

### F.3 PRELIMINARIES OF PENALTY REFORMULATION

Ideally, we want to solve the penalized problem (9) instead of the original bilevel representation learning problem (8). However, the following lemma shows that $\mathsf{L}_{\text{trn}}(W_1, \cdot)$ is not Lipschitz smooth and PL over $W_2$ uniformly for all $W_1 \in \mathbb{R}^{m \times h}$. Therefore, the equivalence of the penalized problem to the bilevel problem and the differentiability of lower-level value function $\mathsf{L}_{\text{trn}}^*(W_1)$ cannot be established directly from (Shen et al., 2023). In this section, we provide the complete proof of them.

**Lemma 17** (Non-uniform smoothness and PL). *The lower-level function $\mathsf{L}_{\text{trn}}(W_1, \cdot)$ is smooth with $\sigma_{\max}^2(X_{\text{trn}})\sigma_{\max}^2(W_1)$ and PL with $\sigma_*^2(X_{\text{trn}}W_1)$. Moreover, if $\sigma_{\min}(W_1) > 0$, then $\mathsf{L}_{\text{trn}}(W_1, \cdot)$ is PL with $\sigma_{\min}^2(X_{\text{trn}})\sigma_{\min}^2(W_1)$.*

**Proof:** First, we know that $h(\tilde{W}) = \frac{1}{2}\|Y_{\text{trn}} - \tilde{W}\|^2$ is 1-strongly convex and 1-Lipschitz smooth. Given matrix $W_1$, then according to (Karimi et al., 2016, Appendix B), $\mathsf{L}_{\text{trn}}(W_1, W_2) = h(X_{\text{trn}}W_1W_2)$ is in the form of strongly convex composite with a linear mapping so that it is $\sigma_*^2(X_{\text{trn}}W_1)$-PL over $W_2$. Also, $\mathsf{L}_{\text{trn}}(W_1, W_2) = h(X_{\text{trn}}W_1W_2)$ is $\sigma_{\max}^2(X_{\text{trn}}W_1)$-Lipschitz smooth over $W_2$. Therefore, the Lipschitz smoothness constant of $\mathsf{L}_{\text{trn}}(W_1, \cdot)$ can be upper bounded by

$$\sigma_{\max}^2(X_{\text{trn}}W_1) \overset{(a)}{\leq} \sigma_{\max}^2(X_{\text{trn}})\sigma_{\max}^2(W_1)$$

where (a) comes from (17). If $W_1$ and $X_{\text{trn}}$ are full row rank, the PL constant is lower bounded by

$$\sigma_*^2(X_{\text{trn}}W_1) = \sigma_{\min}^2(X_{\text{trn}}W_1) \overset{(b)}{\geq} \sigma_{\min}^2(X_{\text{trn}})\sigma_{\min}^2(W_1)$$

where (b) is derived from (18), $X_{\text{trn}} \neq 0$ and $W_1$ is full row rank which is inferred from $W_1$ is a fat matrix with $\sigma_{\min}(W_1) > 0$.

**Definition 4** ($\epsilon$- solution of penalized problem). *We say $(W_1^*, W_2^*)$ is an $\epsilon$- global solution to penalized problem (9) if for any $(W_1, W_2)$, it holds that*

$$\mathsf{L}_\gamma(W_1^*, W_2^*) \leq \mathsf{L}_\gamma(W_1, W_2) + \epsilon.$$

**Theorem 4** (Relations of global solutions). *Suppose Assumption 2 holds. Any $\epsilon_2$ solution to the penalized problem with $\gamma$ is an $(\epsilon_2, \epsilon_\gamma)$ solution to the bilevel problem (8) for some $\epsilon_\gamma = \mathcal{O}(\epsilon_2^2)$. Conversely, for any $\epsilon_1$, there exists $\gamma^* = \mathcal{O}(\epsilon_1^{-1})$ such that for any $\gamma > \gamma^*$, the global optimal solution of bilevel problem (8) must be an $\epsilon_1$ optimal solution of penalized problem with $\gamma$.*

The above theorem shows that to ensure the penalized problem (9) is an $(\epsilon, \epsilon)$ approximate solution to the bilevel problem (8), i.e. $\epsilon_\gamma = \mathcal{O}(\epsilon)$, one need to choose $\epsilon_2 = \mathcal{O}(\sqrt{\epsilon})$ and $\gamma = \mathcal{O}(\epsilon^{-0.5})$.

**Proof:** We will prove the relations through four steps.

**Step 1:** First, we prove that under Assumption 2, there exists an $(\epsilon_1, \epsilon_2)$ solution to the bilevel representation learning problem (8) with full rank $W_1$.

Choose any $(\epsilon_1, \epsilon_2)$ solution $(W_1^*, W_2^*)$ to the bilevel representation learning problem (8) that satisfies Assumption 2. If $W_1^*$ is not full rank, then we can find full rank $W_1$ and $W_2$ such that $W_1W_2 = W_1^*W_2^*$ by the following process. By singular value decomposition, we can decompose $W_1^* = U\Sigma V^\top$ with $\Sigma = \begin{bmatrix} \Sigma_1 & 0 \\ 0 & 0 \end{bmatrix} \in \mathbb{R}^{m \times h}$, and orthogonal matrix $U = [U_1 \ U_2] \in \mathbb{R}^{m \times m}$ and $V = [V_1 \ V_2] \in \mathbb{R}^{h \times h}$. Also, by assuming $\text{Rank}(A) = r$, we know that $U_1 \in \mathbb{R}^{m \times r}, V_1 \in \mathbb{R}^{h \times r}$ and $\Sigma_1 \in \mathbb{R}^{r \times r}$ are full rank submatrix. Therefore, $W_1^*$ can be decomposed by

$$W_1^* = [U_1 \ U_2] \begin{bmatrix} \Sigma_1 & 0 \\ 0 & 0 \end{bmatrix} \begin{bmatrix} V_1^\top \\ V_2^\top \end{bmatrix} = [U_1\Sigma_1 \ 0] \begin{bmatrix} V_1^\top \\ V_2^\top \end{bmatrix} = U_1\Sigma_1 V_1^\top$$

and $V_2$ is the orthogonal basis of $\text{Ker}(W_1^*)$. Furthermore, we can decompose $V_2 = [V_3 \ V_4] \in \mathbb{R}^{h \times (h-r)}$ with $V_3 \in \mathbb{R}^{h \times (m-r)}$. In this way, we can construct $W_1$ and $W_2$ as

$$W_1 = W_1^* + U_2V_2^\top = U_1\Sigma_1 V_1^\top + U_2V_3^\top \text{ and } W_2 = V_1V_1^\top W_2^*$$

where $W_1$ is of full row rank $m$. By Lemma 5, $\mathsf{L}^*_{\mathrm{trn}}(W_1) = 0$. On the other hand, as the training and validation losses only depend on the value of $W_1 W_2$ and $W_1 W_2 = W_1^* W_2^*$, we have

$$\mathsf{L}_{\mathrm{val}}(W_1, W_2) = \mathsf{L}_{\mathrm{val}}(W_1^*, W_2^*), \quad \mathsf{L}_{\mathrm{trn}}(W_1, W_2) = \mathsf{L}_{\mathrm{trn}}(W_1^*, W_2^*).$$

Since $(W_1^*, W_2^*)$ satisfies Assumption 2 and $\mathsf{L}^*_{\mathrm{trn}}(W_1) = 0$, we have

$$\mathsf{L}_{\mathrm{trn}}(W_1, W_2) - \mathsf{L}^*_{\mathrm{trn}}(W_1) = \mathsf{L}_{\mathrm{trn}}(W_1, W_2) = \mathsf{L}_{\mathrm{trn}}(W_1^*, W_2^*) \leq \epsilon_2.$$

With the fact that $\mathsf{L}_{\mathrm{val}}(W_1, W_2) = \mathsf{L}_{\mathrm{val}}(W_1^*, W_2^*)$, we know $(W_1, W_2)$ is also an $(\epsilon_1, \epsilon_2)$ solution of the bilevel representation learning problem and $W_1$ is full row rank by the construction.

**Step 2:** Next, we will prove that any $\epsilon_2$ solution to the penalized problem with $\gamma$ is an $(\epsilon_2, \epsilon_\gamma)$ solution to the bilevel problem for some $\epsilon_\gamma$. For any $\epsilon_2$ solution of the penalized problem $(W_1^\gamma, W_2^\gamma)$, we can find $W_2 \in \arg\min_{W_2' \in \mathcal{S}(W_1^\gamma)} \|W_2' - W_2^\gamma\|$ and

$$\sigma_{\max}(W_2) = \sigma_{\max}(\mathrm{Proj}_{\mathcal{S}(W_1^\gamma)}(W_2^\gamma)) \leq \sigma_{\max}(W_2^\gamma).$$

Then we can derive the Lipschitz continuity of $\mathsf{L}_{\mathrm{val}}(W_1^\gamma, \cdot)$ by

$$
\begin{aligned}
\mathsf{L}_{\mathrm{val}}(W_1^\gamma, W_2^\gamma) - \mathsf{L}_{\mathrm{val}}(W_1^\gamma, W_2) &= \frac{1}{2}\|Y_{\mathrm{val}} - X_{\mathrm{val}} W_1^\gamma W_2^\gamma\|^2 - \frac{1}{2}\|Y_{\mathrm{val}} - X_{\mathrm{val}} W_1^\gamma W_2\|^2 \\
&= \frac{1}{2}\langle 2Y_{\mathrm{val}} - X_{\mathrm{val}} W_1^\gamma (W_2^\gamma + W_2), -X_{\mathrm{val}} W_1^\gamma (W_2^\gamma - W_2)\rangle \\
&\leq L(\gamma)\|W_2^\gamma - W_2^*\|
\end{aligned}
$$

where $L(\gamma) = (\sigma_{\max}(Y_{\mathrm{val}}) + \sigma_{\max}(X_{\mathrm{val}})\sigma_{\max}(W_1^\gamma)\sigma_{\max}(W_2^\gamma))\sigma_{\max}(X_{\mathrm{val}})\sigma_{\max}(W_1^\gamma)$. So it follows

$$
\begin{aligned}
&\mathsf{L}_{\mathrm{val}}(W_1^\gamma, W_2^\gamma) + \gamma(\mathsf{L}_{\mathrm{trn}}(W_1^\gamma, W_2^\gamma) - \mathsf{L}_{\mathrm{trn}}(W_1^\gamma)) \\
&\leq \mathsf{L}_{\mathrm{val}}(W_1^\gamma, W_2) + \epsilon_2 \\
&\leq \mathsf{L}_{\mathrm{val}}(W_1^\gamma, W_2^\gamma) + L(\gamma)\|W_2^\gamma - W_2\| + \epsilon_2 \\
&\leq \mathsf{L}_{\mathrm{val}}(W_1^\gamma, W_2^\gamma) + \sqrt{\frac{2}{\sigma_*(X_{\mathrm{trn}} W_1^\gamma)}} L(\gamma)\sqrt{\mathsf{L}_{\mathrm{trn}}(W_1^\gamma, W_2^\gamma) - \mathsf{L}_{\mathrm{trn}}(W_1^\gamma)} + \epsilon_2
\end{aligned}
\tag{37}
$$

where the first inequality comes from the definition of $\epsilon_2$ solution of penalized problem and $\mathsf{L}_{\mathrm{trn}}(W_1^\gamma, W_2) - \mathsf{L}_{\mathrm{trn}}(W_1^\gamma) = 0$, and the last inequality is derived from the PL condition of $\mathsf{L}_{\mathrm{trn}}(W_1^\gamma)$ and Lemma 6. According to (37), we have either

$$\mathsf{L}_{\mathrm{trn}}(W_1^\gamma, W_2^\gamma) - \mathsf{L}_{\mathrm{trn}}(W_1^\gamma) \leq \frac{2\epsilon_2}{\gamma} \tag{38}$$

or $\epsilon_2 < \frac{\gamma}{2}(\mathsf{L}_{\mathrm{trn}}(W_1^\gamma, W_2^\gamma) - \mathsf{L}_{\mathrm{trn}}(W_1^\gamma))$ and thus

$$\frac{\gamma}{2}(\mathsf{L}_{\mathrm{trn}}(W_1^\gamma, W_2^\gamma) - \mathsf{L}_{\mathrm{trn}}(W_1^\gamma)) \leq \sqrt{\frac{2}{\sigma_*(X_{\mathrm{trn}} W_1^\gamma)}} L(\gamma)\sqrt{\mathsf{L}_{\mathrm{trn}}(W_1^\gamma, W_2^\gamma) - \mathsf{L}_{\mathrm{trn}}(W_1^\gamma)}$$

which yields

$$\mathsf{L}_{\mathrm{trn}}(W_1^\gamma, W_2^\gamma) - \mathsf{L}_{\mathrm{trn}}(W_1^\gamma) \leq \frac{8}{\sigma_*(X_{\mathrm{trn}} W_1^\gamma)}\left(\frac{L(\gamma)}{\gamma}\right)^2. \tag{39}$$

By choosing $\gamma = \mathcal{O}(\epsilon_2^{-1})$ and noting that $\{L(\gamma), \sigma_*(X_{\mathrm{trn}} W_1^\gamma)\} = \mathcal{O}(1)$, (38) and (39) indicate that

$$\epsilon_\gamma := \mathsf{L}_{\mathrm{trn}}(W_1^\gamma, W_2^\gamma) - \mathsf{L}^*_{\mathrm{trn}}(W_1^\gamma) \leq \max\left\{\frac{2\epsilon_2}{\gamma}, \frac{8}{\sigma_*(X_{\mathrm{trn}} W_1^\gamma)}\left(\frac{L(\gamma)}{\gamma}\right)^2\right\} = \mathcal{O}\left(\frac{\epsilon_2}{\gamma}\right) = \mathcal{O}\left(\epsilon_2^2\right).$$

Moreover, for any $(\tilde{W}_1, \tilde{W}_2)$ satisfying $\mathsf{L}_{\mathrm{trn}}(\tilde{W}_1, \tilde{W}_2) - \mathsf{L}^*_{\mathrm{trn}}(\tilde{W}_1) \leq \epsilon_\gamma$, it holds that

$$\mathsf{L}_{\mathrm{val}}(W_1^\gamma, W_2^\gamma) + \gamma(\mathsf{L}_{\mathrm{trn}}(W_1^\gamma, W_2^\gamma) - \mathsf{L}^*_{\mathrm{trn}}(W_1^\gamma)) \leq \mathsf{L}_{\mathrm{val}}(\tilde{W}_1, \tilde{W}_2) + \gamma(\mathsf{L}_{\mathrm{trn}}(\tilde{W}_1, \tilde{W}_2) - \mathsf{L}^*_{\mathrm{trn}}(\tilde{W}_1)) + \epsilon_2$$

and thus

$$\mathsf{L}_{\mathrm{val}}(W_1^\gamma, W_2^\gamma) - \mathsf{L}_{\mathrm{val}}(\tilde{W}_1, \tilde{W}_2) \leq \gamma(\mathsf{L}_{\mathrm{trn}}(\tilde{W}_1, \tilde{W}_2) - \mathsf{L}^*_{\mathrm{trn}}(\tilde{W}_1) - \epsilon_\gamma) + \epsilon_2 \leq \epsilon_2$$

which indicates $(W_1^\gamma, W_2^\gamma)$ is $(\epsilon_2, \epsilon_\gamma)$ optimal solution of the bilevel problem. This concludes that any $\epsilon_2$ solution to the penalized problem is an $(\epsilon_2, \epsilon_\gamma)$ solution to the bilevel problem.

**Step 3:** To prove the converse part, we first claim that for any $\epsilon_2$, there exists an $\epsilon_2$ solution to the penalized problem with full rank $W_1$, which is built on the previous two steps. Take the $\epsilon_2$ solution $(W_1^\gamma, W_2^\gamma)$ of the penalized problem in the second step, it is an $(\epsilon_2, \epsilon_\gamma)$ optimal solution of the bilevel problem. From the first step, we know there exists an $(\epsilon_2, \epsilon_\gamma)$ optimal solution of the bilevel problem $(W_1^\epsilon, W_2^\epsilon)$ with full rank $W_1^\epsilon$. Therefore, we have $\mathsf{L}_{\mathrm{val}}(W_1^\gamma, W_2^\gamma) = \mathsf{L}_{\mathrm{val}}(W_1^\epsilon, W_2^\epsilon)$ and the feasibility

$$\mathsf{L}_{\mathrm{trn}}(W_1^\epsilon, W_2^\epsilon) - \mathsf{L}_{\mathrm{trn}}^*(W_1^\epsilon) \leq \epsilon_\gamma = \mathsf{L}_{\mathrm{trn}}(W_1^\gamma, W_2^\gamma) - \mathsf{L}_{\mathrm{trn}}^*(W_1^\gamma).$$

Adding these two inequalities together, we have $\mathsf{L}_\gamma(W_1^\epsilon, W_2^\epsilon) \leq \mathsf{L}_\gamma(W_1^\gamma, W_2^\gamma)$, which suggests that $(W_1^\epsilon, W_2^\epsilon)$ is also an $\epsilon_2$ solution to the penalized problem. As $W_1^\epsilon$ is full rank, we prove the claim.

**Step 4:** We will then prove the converse part, i.e. for any $\epsilon_1$, there exists $\gamma^*$ such that any global solution of bilevel problem is an $\epsilon_1$ solution to the penalized problem for any $\gamma \geq \gamma^*$. Take any global solution of bilevel problem and denote it as $(\tilde{W}_1, \tilde{W}_2)$. According to the first step, we know that there exists a global solution to the bilevel problem $(W_1^*, W_2^*)$ with full rank $W_1^*$. Also there exists an $\epsilon_1$ solution to the penalized problem $(W_1^\gamma, W_2^\gamma)$ with full rank $W_1^\gamma$. Therefore, we can restrict the bilevel and penalized problem on the following constrained sets

$$\mathcal{W}_1 = \{W_1 : r_1 \leq \sigma_{\min}(W_1) \leq \sigma_{\max}(W_1) \leq R_1\}, \quad \text{and} \quad \mathcal{W}_2 = \{W_2 : \sigma_{\max}(W_2) \leq R_2\}$$

where $r_1 < \min\{\sigma_{\min}(W_1^*), \sigma_{\min}(W_1^\gamma)\}$, $R_i > \max\{\sigma_{\max}(W_i^*), \sigma_{\max}(W_i^\gamma)\}$, $i = 1, 2$. In this way, $\mathcal{W}_2$ is a convex closed set while $\mathcal{W}_1$ is a closed but nonconvex set, and

$$\min_{W_1 \in \mathcal{W}_1, W_2 \in \mathcal{W}_2} \mathsf{L}_\gamma(W_1, W_2) = \min_{W_1, W_2} \mathsf{L}_\gamma(W_1, W_2)$$

because the constrained sets include the minimizer of penalized problem. Also, $\mathsf{L}_{\mathrm{val}}(W_1, \cdot)$ is uniformly Lipschitz continuous on $\mathcal{W}_1$ and $\mathcal{W}_2$, and $\mathsf{L}_{\mathrm{trn}}(W_1, \cdot)$ is uniformly smooth and PL on $\mathcal{W}_1$ and $\mathcal{W}_2$. Adapted from (Shen et al., 2023, Theorem 1), there exists $\gamma^* = \mathcal{O}(\epsilon_1^{-1})$ s.t. for any $\gamma > \gamma^*$, we have [2]

$$\mathsf{L}_\gamma(W_1^*, W_2^*) \leq \min_{W_1 \in \mathcal{W}_1, W_2 \in \mathcal{W}_2} \mathsf{L}_\gamma(W_1, W_2) + \epsilon_1$$
$$= \min_{W_1, W_2} \mathsf{L}_\gamma(W_1, W_2) + \epsilon_1$$

Since $\mathsf{L}_\gamma(\tilde{W}_1, \tilde{W}_2) = \mathsf{L}_{\mathrm{val}}(\tilde{W}_1, \tilde{W}_2) = \mathsf{L}_{\mathrm{val}}(W_1^*, W_2^*) = \mathsf{L}_\gamma(W_1^*, W_2^*)$, we know $(\tilde{W}_1, \tilde{W}_2)$ is an $\epsilon_1$ optimal point of the penalized problem, which concludes the proof.

The following lemma shows that $\mathsf{L}^*(W_1)$ is not differentiable at the whole space, but it is differentiable for $W_1$ with lower and upper bounded singular value.

**Lemma 18** (Danskin type theorem). *Assume there exists $r, R \in \mathbb{R}_{>0}$ such that for all $W_1 \in \mathcal{W}_1$, $0 < r \leq \sigma_{\min}(W_1) \leq \sigma_{\max}(W_1) \leq R$. Then $\mathsf{L}_{\mathrm{trn}}^*(W_1)$ is differentiable on $\mathcal{W}_1$ with the gradient*

$$\nabla \mathsf{L}_{\mathrm{trn}}^*(W_1) = \nabla_{W_1} \mathsf{L}_{\mathrm{trn}}(W_1, W_2^*), \qquad \forall W_2^* \in \mathcal{S}(W_1) \text{ with finite norm.}$$

*Moreover, $\nabla \mathsf{L}_{\mathrm{trn}}^*(W_1) = 0$ on $\mathcal{W}_1$.*

**Proof:** First, $\forall W_1 \in \mathcal{W}_1$, without loss of generality, we can set $W_2^* = (X_{\mathrm{trn}} W_1)^\dagger Y_{\mathrm{trn}} \in \mathcal{S}(W_1)$ with finite $\ell_2$ norm

$$\sigma_{\max}(W_2^*) = \sigma_{\max}((X_{\mathrm{trn}} W_1)^\dagger Y_{\mathrm{trn}}) \overset{(a)}{\leq} \sigma_{\max}((X_{\mathrm{trn}} W_1)^\dagger) \sigma_{\max}(Y_{\mathrm{trn}})$$
$$= \frac{\sigma_{\max}(Y_{\mathrm{trn}})}{\sigma_{\min}(X_{\mathrm{trn}} W_1)} \leq \frac{\sigma_{\max}(Y_{\mathrm{trn}})}{\sigma_{\min}(X_{\mathrm{trn}}) \sigma_{\min}(W_1)} \leq \frac{\sigma_{\max}(Y_{\mathrm{trn}})}{\sigma_{\min}(X_{\mathrm{trn}}) r}$$

where (a) comes from $\sigma_{\max}(A^\dagger) = 1/\sigma_{\min}(A^\dagger)$ if $\sigma_{\min}(A^\dagger) > 0$. The benefit of choosing $W_2^*$ in this way is to ensure $\nabla_{W_2} \mathsf{L}_{\mathrm{trn}}(W_1, W_2^*)$ is Lipschitz continuous over $W_1$, which can be proved by

$$\|\nabla_{W_2} \mathsf{L}_{\mathrm{trn}}(W_1, W_2^*) - \nabla_{W_2} \mathsf{L}_{\mathrm{trn}}(W_1', W_2^*)\|_2$$
$$= \| - (X_{\mathrm{trn}} W_1)^\top (Y_{\mathrm{trn}} - X_{\mathrm{trn}} W_1 W_2^*) + (X_{\mathrm{trn}} W_1')^\top (Y_{\mathrm{trn}} - X_{\mathrm{trn}} W_1' W_2^*)\|_2$$

---

[2] (Shen et al., 2023, Theorem 1) only requires the convexity of $\mathcal{W}_2$

$$\leq \|(X_{\text{trn}}W_1)^\top X_{\text{trn}}(W_1 - W_1')W_2^*\|_2 + \|(W_1 - W_1')^\top X_{\text{trn}}^\top (Y_{\text{trn}} - X_{\text{trn}}W_1'W_2^*)\|_2$$

$$\leq \sigma_{\max}^2(X_{\text{trn}})\sigma_{\max}(W_1)\sigma_{\max}(W_2^*)\|W_1 - W_1'\|$$
$$+ \sigma_{\max}(X_{\text{trn}})[\sigma_{\max}(Y_{\text{trn}}) + \sigma_{\max}(X_{\text{trn}})\sigma_{\max}(W_1')\sigma_{\max}(W_2^*)]\|W_1 - W_1'\|$$

$$\leq \left[ \frac{\sigma_{\max}^2(X_{\text{trn}})\sigma_{\max}(Y_{\text{trn}})R}{\sigma_{\min}(X_{\text{trn}})r} + \sigma_{\max}(X_{\text{trn}})\sigma_{\max}(Y_{\text{trn}})\left(1 + \frac{\sigma_{\max}(X_{\text{trn}})R}{\sigma_{\min}(X_{\text{trn}})r}\right)\right]\|W_1 - W_1'\|$$

where $W_1, W_1' \in \mathcal{W}_1$.

On the other hand, for any $W_1 \in \mathcal{W}_1$, as long as it does not belong to the boundary, we can find $\bar{a} > 0$ such that for any $a \leq \bar{a}$, $W_1 + ad \in \mathcal{W}_1$ holds for any unit direction $d \in \mathbb{R}^{m \times h}$. Even for $W_1$ at the boundary of $\mathcal{W}_1$, one can slightly enlarge $\mathcal{W}_1$ by adjusting $r$ and $R$ correspondingly to ensure the singular value of any $W_1 \in \mathbb{B}(W_1, \bar{a})$ is uniformly lower and upper bounded. Then based on the Lipschitz continuity of $\nabla_{W_2}\mathsf{L}_{\text{trn}}(W_1, W_2^*)$ over $W_1$ and $W_2$ and the PL property of $\mathsf{L}_{\text{trn}}(W_1, \cdot)$, and according to (Nouiehed et al., 2019, Lemma A.3), there exist $L > 0$, for any $0 < a \leq \bar{a}$ and any unit direction $d \in \mathbb{R}^{m \times h}$, one can choose $W_2^*(a) \in S(W_1 + ad)$ such that $\|W_2^*(a) - W_2^*\| \leq La$. By the Taylor expansion,

$$\mathsf{L}_{\text{trn}}^*(W_1 + ad) - \mathsf{L}_{\text{trn}}^*(W_1) = \mathsf{L}_{\text{trn}}(W_1 + ad, W_2^*(a)) - \mathsf{L}_{\text{trn}}(W_1, W_2^*)$$
$$= a\nabla_{W_1}^\top \mathsf{L}_{\text{trn}}(W_1, W_2^*)d + \nabla_{W_2}^\top \mathsf{L}_{\text{trn}}(W_1, W_2^*)(W_2^*(a) - W_2^*) + \mathcal{O}(a^2)$$
$$= a\nabla_{W_1}^\top \mathsf{L}_{\text{trn}}(W_1, W_2^*)d + \mathcal{O}(a^2)$$

By the definition of directional derivative, we know

$$\mathsf{L}_{\text{trn}}^{*,'}(W_1; d) = \lim_{r \to 0^+} \frac{\mathsf{L}_{\text{trn}}^*(W_1 + ad) - \mathsf{L}_{\text{trn}}^*(W_1)}{a} = \nabla_{W_1}^\top \mathsf{L}_{\text{trn}}(W_1, W_2^*)d$$

Since the above relationship holds for any direction $d$, then we get $\nabla \mathsf{L}_{\text{trn}}^*(W_1) = \nabla_{W_1}\mathsf{L}_{\text{trn}}(W_1, W_2^*)$. In addition, the above discussion holds for any bounded $W_2^*$, which yields the first part of the conclusion. Moreover, as $\nabla_{W_1}\mathsf{L}_{\text{trn}}(W_1, W_2^*) = \nabla\ell_{\text{trn}}(W_1 W_2^*)W_2^{*,\top} = X_{\text{trn}}^\top \ell_{\text{trn}}(W_1 W_2^*)W_2^{*,\top}$ and $\ell_{\text{trn}}(W_1 W_2^*) = \min_{W_2} \mathsf{L}_{\text{trn}}(W_1, W_2) = 0$ according to Lemma 5, we get $\nabla \mathsf{L}_{\text{trn}}^*(W_1) = \nabla_{W_1}\mathsf{L}_{\text{trn}}(W_1, W_2^*) = \nabla_{W_1}\mathsf{L}_{\text{trn}}(W_1, W_2^*) = 0$.

**Lemma 19.** *Under Assumption 2, for any $\epsilon > 0$, there exists an $\epsilon$-solution to the penalized problem $(W_1^\epsilon, W_2^\epsilon)$ with $\mathsf{L}_{\text{trn}}^*(W_1^\epsilon) = 0$.*

Lemma 19 is pivotal in ensuring that the penalized problem is well-posed. Otherwise, we encounter the situation where the optimal weight $(W_1^*, W_2^*)$ learned from the penalized method

$$\mathsf{L}_{\text{trn}}^*(W_1^*) \neq \min_{W_1, W_2} \mathsf{L}_{\text{trn}}(W_1, W_2) = 0.$$

This implies that the learned bottom layer weight fails to fit well on the training dataset – a scenario we want to avoid. Without making assumption on the penalized problem, the well-poseness of it is derived from the singularity assumption of the bilevel problem and their approximated equivalence.

**Proof:** Lemma 19 is a side product of Theorem 4. According to **Step 2-1** in the proof of Theorem 4, we know that for any $\epsilon$, there exists an $\epsilon$-solution to the penalized problem with full rank $W_1$. Together with Lemma 5, we arrive at the conclusion.

### F.4 PROOF OF THEOREM 1

We restate Theorem 1 and the descent lemma in the following theorem.

**Theorem 5** (Local joint PL and smoothness). *Suppose Assumption 2 holds, assume $\sigma_{\min}^2(W_1^k) > 0$, $\sigma_{\min}^2(W_2^k) > 0$ and denote $X_\gamma = [X_{\text{val}}; \sqrt{\gamma}X_{\text{trn}}]$. Then the descent lemma holds with $L_k$ defined in (51) and the local joint PL inequality holds with $\mu_k = (\sigma_{\min}^2(W_1^k) + \sigma_{\min}^2(W_2^k))\sigma_*^2(X_\gamma)$, i.e.*

$$\|\nabla \mathsf{L}_\gamma(W_1^k, W_2^k)\|^2 \geq 2\mu_k(\mathsf{L}_\gamma(W_1^k, W_2^k) - \mathsf{L}_\gamma^*)$$

$$\mathsf{L}_\gamma(Z^{k+1}) \leq \mathsf{L}_\gamma(Z^k) - \left(\frac{\alpha}{2} - \alpha^2 L_k\right)\|\nabla \mathsf{L}_\gamma(Z^k)\|^2 + \left(\frac{\alpha}{2} + \alpha^2 L_k\right)\|\delta_k\|^2.$$

*where $Z = (W_1, W_2)$, $Z^k = (W_1^k, W_2^k)$, $\alpha$ and $\mathsf{L}_\gamma^* = \min_Z \mathsf{L}_\gamma(Z)$. Moreover, one has*

$$\mathsf{L}_\gamma(Z^{k+1}) - \mathsf{L}_\gamma^* \leq \left(1 - \alpha\mu_k + 2\alpha^2 L_k \mu_k\right)\left(\mathsf{L}_\gamma(Z^k) - \mathsf{L}_\gamma^*\right) + \left(\frac{\alpha}{2} + \alpha^2 L_k\right)\|\delta_k\|^2.$$

**Proof:** We first prove the local PL inequality by four steps.

**Step 1-1: Local PL inequality for $\mathsf{L}_{\mathrm{trn}}(W_1, W_2)$ and $\mathsf{L}_{\mathrm{val}}(W_1, W_2)$.** First, we know that $h(\tilde{W}) = \frac{1}{2}\|Y_{\mathrm{trn}} - \tilde{W}\|^2$ is 1-strongly convex and 1-Lipschitz smooth. Given matrix $W$, then according to (Karimi et al., 2016, Appendix B), $\ell_{\mathrm{trn}}(W) = h(X_{\mathrm{trn}}W)$ is in the form of strongly convex composite with a linear mapping so that it is $\sigma_*^2(X_{\mathrm{trn}})$-PL over $W$. As $X_{\mathrm{trn}}$ is full rank, we have $\sigma_*^2(X_{\mathrm{trn}}) = \sigma_{\min}^2(X_{\mathrm{trn}})$ and $\ell_{\mathrm{trn}}(W) = h(X_{\mathrm{trn}}W)$ is $\sigma_{\min}^2(X_{\mathrm{trn}})$-PL over $W$. On the other hand, we also have

$$\min_W \ell_{\mathrm{trn}}(W) = \min_{W_1, W_2} \mathsf{L}_{\mathrm{trn}}(W_1, W_2) \tag{40}$$

because

$$\min_W \ell_{\mathrm{trn}}(W) \leq \ell_{\mathrm{trn}}(W_1^* W_2^*) = \min_{W_1, W_2} \mathsf{L}_{\mathrm{trn}}(W_1, W_2) \tag{41}$$

where $W_1^*, W_2^* = \arg\min_{W_1, W_2} \mathsf{L}_{\mathrm{trn}}(W_1, W_2)$. The reverse direction of (41) holds true because for any $W^* \in \arg\min_W \ell_{\mathrm{trn}}(W)$, it can be decomposed to

$$W^* = \begin{bmatrix} W^* & 0_{n \times (h-m)} \end{bmatrix} \begin{bmatrix} I_{m \times m} \\ 0_{(h-m) \times m} \end{bmatrix}.$$

Based on the above facts and denoting $W = W_1 W_2$, we can prove the local PL property of $\mathsf{L}_{\mathrm{trn}}$ over $(W_1, W_2)$ by

$$
\begin{aligned}
\|\nabla \mathsf{L}_{\mathrm{trn}}(W_1, W_2)\|^2 &= \|\nabla \ell_{\mathrm{trn}}(W) W_2^\top\|^2 + \|W_1^\top \nabla \ell_{\mathrm{trn}}(W)\|^2 \\
&= \|W_2 \nabla \ell_{\mathrm{trn}}(W)^\top\|^2 + \|W_1^\top \nabla \ell_{\mathrm{trn}}(W)\|^2 \\
&\overset{(a)}{\geq} \sigma_{\min}^2(W_2)\|\nabla \ell_{\mathrm{trn}}(W)\|^2 + \sigma_{\min}^2(W_1)\|\nabla \ell_{\mathrm{trn}}(W)\|^2 \\
&= (\sigma_{\min}^2(W_1) + \sigma_{\min}^2(W_2))\|\nabla \ell_{\mathrm{trn}}(W)\|^2 \\
&\overset{(b)}{\geq} 2(\sigma_{\min}^2(W_1) + \sigma_{\min}^2(W_2))\sigma_{\min}^2(X_{\mathrm{trn}})(\ell_{\mathrm{trn}}(W) - \min_W \ell_{\mathrm{trn}}(W)) \\
&\overset{(c)}{\geq} 2(\sigma_{\min}^2(W_1) + \sigma_{\min}^2(W_2))\sigma_{\min}^2(X_{\mathrm{trn}})(\mathsf{L}_{\mathrm{trn}}(W_1, W_2) - \min_{W_1, W_2} \mathsf{L}_{\mathrm{trn}}(W_1, W_2))
\end{aligned}
\tag{42}
$$

where (a) comes from Lemma 4, (b) is derived from $\ell_{\mathrm{trn}}(W)$ is $\sigma_{\min}^2(X_{\mathrm{trn}})$-PL, and (c) is because $\min_W \ell_{\mathrm{trn}}(W) = \min_{W_1, W_2} \mathsf{L}_{\mathrm{trn}}(W_1, W_2)$. Similarly, for the validation loss, it holds that

$$\|\nabla \mathsf{L}_{\mathrm{val}}(W_1^k, W_2^k)\|^2 \geq 2(\sigma_{\min}^2(W_1) + \sigma_{\min}^2(W_2))\sigma_{\min}^2(X_{\mathrm{val}})(\mathsf{L}_{\mathrm{val}}(W_1, W_2) - \min_{W_1, W_2} \mathsf{L}_{\mathrm{val}}(W_1, W_2)).$$

**Step 1-2: Local PL inequality for $\widetilde{\mathsf{L}}_\gamma(W_1, W_2) =: \mathsf{L}_{\mathrm{val}}(W_1, W_2) + \gamma \mathsf{L}_{\mathrm{trn}}(W_1, W_2)$.**

By noticing the fact that $(\ell_{\mathrm{val}} + \gamma \ell_{\mathrm{trn}})(W) = \frac{1}{2}\|Y_\gamma - X_\gamma W\|^2$ with $Y_\gamma = [Y_{\mathrm{val}}; \sqrt{\gamma} Y_{\mathrm{trn}}]$ and $X_\gamma = [X_{\mathrm{val}}; \sqrt{\gamma} X_{\mathrm{trn}}]$, we have

$$
\begin{aligned}
\|\nabla \widetilde{\mathsf{L}}_\gamma(W_1, W_2)\|^2 &= \|\nabla(\ell_{\mathrm{val}} + \gamma \ell_{\mathrm{trn}})(W) W_2^\top\|^2 + \|W_1^\top \nabla(\ell_{\mathrm{val}} + \gamma \ell_{\mathrm{trn}})(W)\|^2 \\
&\overset{(a)}{\geq} 2(\sigma_{\min}^2(W_1) + \sigma_{\min}^2(W_2))\sigma_*^2(X_\gamma)(\widetilde{\mathsf{L}}_\gamma(W_1, W_2) - \min_{W_1, W_2} \widetilde{\mathsf{L}}_\gamma(W_1, W_2))
\end{aligned}
\tag{43}
$$

where (a) is derived similarly from the derivation of (42).

**Step 1-3: We then prove the relation of $\mathsf{L}_\gamma(W_1, W_2)$ and $\widetilde{\mathsf{L}}_\gamma(W_1, W_2)$ that**

$$\min_{W_1, W_2} \mathsf{L}_\gamma(W_1, W_2) = \min_{W_1, W_2} \widetilde{\mathsf{L}}_\gamma(W_1, W_2). \tag{44}$$

Since $\mathsf{L}_{\mathrm{trn}}(W_1, W_2) \geq 0$, we know $\mathsf{L}_{\mathrm{trn}}^*(W_1) = \min_{W_2} \mathsf{L}_{\mathrm{trn}}(W_1, W_2) \geq 0$, and thus

$$\mathsf{L}_\gamma(W_1, W_2) = \mathsf{L}_{\mathrm{val}}(W_1, W_2) + \gamma \mathsf{L}_{\mathrm{trn}}(W_1, W_2) - \gamma \mathsf{L}_{\mathrm{trn}}^*(W_1) \leq \mathsf{L}_{\mathrm{val}}(W_1, W_2) + \gamma \mathsf{L}_{\mathrm{trn}}(W_1, W_2).$$

Taking the minimization over both sides yields

$$\min_{W_1,W_2} \mathsf{L}_\gamma(W_1, W_2) \le \min_{W_1,W_2} (\mathsf{L}_{\mathrm{val}}(W_1, W_2) + \gamma \mathsf{L}_{\mathrm{trn}}(W_1, W_2)) = \min_{W_1,W_2} \widetilde{\mathsf{L}}_\gamma(W_1, W_2). \tag{45}$$

By Lemma 19, for any $\epsilon > 0$, $\exists(W_1^\epsilon, W_2^\epsilon)$ is an $\epsilon$-solution of $\mathsf{L}_\gamma(W_1, W_2)$ with $\mathsf{L}_{\mathrm{trn}}^*(W_1^*) = 0$, so

$$\epsilon + \min_{W_1,W_2} \mathsf{L}_\gamma(W_1, W_2) \ge \mathsf{L}_\gamma(W_1^\epsilon, W_2^\epsilon) = \mathsf{L}_{\mathrm{val}}(W_1^\epsilon, W_2^\epsilon) + \gamma \mathsf{L}_{\mathrm{trn}}(W_1^\epsilon, W_2^\epsilon)$$

$$\ge \min_{W_1,W_2} (\mathsf{L}_{\mathrm{val}}(W_1, W_2) + \gamma \mathsf{L}_{\mathrm{trn}}(W_1, W_2)) = \min_{W_1,W_2} \widetilde{\mathsf{L}}_\gamma(W_1, W_2)$$

holds for any $\epsilon$. Letting $\epsilon \to 0$, we get

$$\min_{W_1,W_2} \mathsf{L}_\gamma(W_1, W_2) \ge \min_{W_1,W_2} \widetilde{\mathsf{L}}_\gamma(W_1, W_2)$$

Together with (45), (44) must hold true.

**Step 1-4: Local PL property of $\mathsf{L}_\gamma(W_1, W_2)$.**

$$\|\nabla \mathsf{L}_\gamma(W_1, W_2)\|^2 = \|\nabla \ell_{\mathrm{val}}(W)W_2^\top + \gamma \nabla \ell_{\mathrm{trn}}(W)W_2^\top - \gamma \nabla \mathsf{L}_{\mathrm{trn}}^*(W_1)\|^2$$

$$+ \|W_1^\top \nabla \ell_{\mathrm{val}}(W) + \gamma W_1^\top \nabla \ell_{\mathrm{trn}}(W)\|^2$$

$$\overset{(a)}{=} \|\nabla \ell_{\mathrm{val}}(W)W_2^\top + \gamma \nabla \ell_{\mathrm{trn}}(W)W_2^\top\|^2 + \|W_1^\top \nabla \ell_{\mathrm{val}}(W) + \gamma W_1^\top \nabla \ell_{\mathrm{trn}}(W)\|^2$$

$$= \|\nabla(\ell_{\mathrm{val}} + \gamma \ell_{\mathrm{trn}})(W)W_2^\top\|^2 + \|W_1^\top \nabla(\ell_{\mathrm{val}} + \gamma \ell_{\mathrm{trn}})(W)\|^2$$

$$\overset{(b)}{\ge} 2(\sigma_{\min}^2(W_1) + \sigma_{\min}^2(W_2))\sigma_*^2(X_\gamma)\left(\widetilde{\mathsf{L}}_\gamma(W_1, W_2) - \min_{W_1,W_2} \widetilde{\mathsf{L}}_\gamma(W_1, W_2)\right)$$

$$\overset{(c)}{=} 2(\sigma_{\min}^2(W_1) + \sigma_{\min}^2(W_2))\sigma_*^2(X_\gamma)(\mathsf{L}_\gamma(W_1, W_2) - \min_{W_1,W_2} \mathsf{L}_\gamma(W_1, W_2))$$

$$= 2\mu_k(\mathsf{L}_\gamma(W_1, W_2) - \min_{W_1,W_2} \mathsf{L}_\gamma(W_1, W_2))$$

where (a) comes from Lemma 18 and (b) is derived from (43), and (c) holds because of (44) and $\mathsf{L}_{\mathrm{trn}}^*(W_1) = 0$ when $\sigma_{\min}(W_1) > 0$.

Next, we will prove the descent lemma. Denoting $Z^k = (W_1^k, W_2^k)$, the update of PBGD can be formulated as

$$Z^{k+1} = Z^k - \alpha(\nabla \mathsf{L}_\gamma(Z^k) + \delta_k) \tag{46}$$

where $\delta_k = \gamma(\nabla \mathsf{L}_{\mathrm{trn}}^*(W_1^k) - \nabla_{W_1} \mathsf{L}_{\mathrm{trn}}(W_1^k, W_3^{k+1}))$. Let us denote

$$H(\kappa) = \nabla^2 \widetilde{\mathsf{L}}_\gamma\left((1-\kappa)W_1^k + \kappa W_1^{k+1}, (1-\kappa)W_2^k + \kappa W_2^{k+1}\right)$$

$$= \nabla^2 \widetilde{\mathsf{L}}_\gamma\left(W_1^k - \alpha\kappa(\nabla_{W_1} \mathsf{L}_\gamma(Z^k) + \delta_k), W_2^k - \alpha\kappa \nabla_{W_2} \mathsf{L}_\gamma(Z^k)\right)$$

$$= \nabla^2 \widetilde{\mathsf{L}}_\gamma\left(W_1^k - \alpha\kappa(\nabla_{W_1} \widetilde{\mathsf{L}}_\gamma(Z^k) + \delta_k), W_2^k - \alpha\kappa \nabla_{W_2} \widetilde{\mathsf{L}}_\gamma(Z^k)\right)$$

$$= \nabla^2 \widetilde{\mathsf{L}}_\gamma\left(Z^k - \alpha\kappa(\nabla \widetilde{\mathsf{L}}_\gamma(Z^k) + \delta_k)\right) \tag{47}$$

then by second order Taylor expansion, we have

$$\mathsf{L}_\gamma(Z^{k+1}) \le \widetilde{\mathsf{L}}_\gamma(Z^{k+1})$$

$$= \widetilde{\mathsf{L}}_\gamma(Z^k) + \langle \nabla \widetilde{\mathsf{L}}_\gamma(Z^k), Z^{k+1} - Z^k \rangle + \int_0^1 (1-\kappa) \langle Z_{t+1} - Z_t, H(\kappa)(Z^{k+1} - Z^k) \rangle \, d\kappa$$

$$\overset{(a)}{=} \mathsf{L}_\gamma(Z^k) - \alpha\langle \nabla \mathsf{L}_\gamma(Z^k), \nabla \mathsf{L}_\gamma(Z^k) + \delta_k \rangle + \int_0^1 (1-\kappa) \langle Z_{t+1} - Z_t, H(\kappa)(Z^{k+1} - Z^k) \rangle \, d\kappa$$

$$\overset{(b)}{\le} \mathsf{L}_\gamma(Z^k) - \frac{\alpha\|\nabla \mathsf{L}_\gamma(Z^k)\|^2}{2} + \frac{\alpha\|\delta_k\|^2}{2} + \alpha^2 \int_0^1 (1-\kappa) \langle \nabla \mathsf{L}_\gamma(Z^k) + \delta_k, H(\kappa)(\nabla \mathsf{L}_\gamma(Z^k) + \delta_k) \rangle \, d\kappa$$

$$\overset{(c)}{=} \mathsf{L}_\gamma(Z^k) - \frac{\alpha\|\nabla\mathsf{L}_\gamma(Z^k)\|^2}{2} + \frac{\alpha\|\delta_k\|^2}{2} + \alpha^2\|\nabla\mathsf{L}_\gamma(Z^k) + \delta_k\|^2 \int_0^1 (1-\kappa)\langle g_k, H(\kappa)g_k\rangle \, d\kappa$$

$$\overset{(d)}{\leq} \mathsf{L}_\gamma(Z^k) - \frac{\alpha\|\nabla\mathsf{L}_\gamma(Z^k)\|^2}{2} + \frac{\alpha\|\delta_k\|^2}{2} + 2\alpha^2\left(\|\nabla\mathsf{L}_\gamma(Z^k)\|^2 + \|\delta_k\|^2\right) \int_0^1 (1-\kappa)L_k \, d\kappa$$

$$= \mathsf{L}_\gamma(Z^k) - \left(\frac{\alpha}{2} - \alpha^2 L_k\right)\|\nabla\mathsf{L}_\gamma(Z^k)\|^2 + \left(\frac{\alpha}{2} + \alpha^2 L_k\right)\|\delta_k\|^2 \tag{48}$$

where (a) is derived from (46), $\sigma_{\min}(W_1^k) > 0$, Lemma 5 and Lemma 18, (b) holds because

$$\langle a, b\rangle = \frac{\|a\|^2}{2} + \frac{\|b\|^2}{2} - \frac{\|a-b\|^2}{2} \geq \frac{\|a\|^2}{2} - \frac{\|a-b\|^2}{2}$$

$g_k$ in (c) is defined as $\frac{\nabla\mathsf{L}_\gamma(Z^k)+\delta_k}{\|\nabla\mathsf{L}_\gamma(Z^k)+\delta_k\|}$, and (d) is earned by Lemma 22.

Finally, according to (48) and the local PL property of $\mathsf{L}_\gamma(W_1, W_2)$, we have

$$\mathsf{L}_\gamma(Z^{k+1}) \leq \mathsf{L}_\gamma(Z^k) - \left(\frac{\alpha}{2} - \alpha^2 L_k\right)\|\nabla\mathsf{L}_\gamma(Z^k)\|^2 + \left(\frac{\alpha}{2} + \alpha^2 L_k\right)\|\delta_k\|^2$$

$$\leq \mathsf{L}_\gamma(Z^k) - \left(\alpha - 2\alpha^2 L_k\right)\mu_k(\mathsf{L}_\gamma(Z^k) - \mathsf{L}_\gamma^*) + \left(\frac{\alpha}{2} + \alpha^2 L_k\right)\|\delta_k\|^2.$$

Subtracting both sides by $\mathsf{L}_\gamma^*$ yields

$$\mathsf{L}_\gamma(Z^{k+1}) - \mathsf{L}_\gamma^* \leq \left(1 - \alpha\mu_k + 2\alpha^2 L_k\mu_k\right)\left(\mathsf{L}_\gamma(Z^k) - \mathsf{L}_\gamma^*\right) + \left(\frac{\alpha}{2} + \alpha^2 L_k\right)\|\delta_k\|^2.$$

We give the characteristic of $L_k$, the bound of $\langle g_k, H(\kappa)g_k\rangle$, by showing $\langle g_k, H(0)g_k\rangle$ and $|\langle g_k, (H(\kappa) - H(0))g_k\rangle|$ are bounded subsequently. The upper bound of $\langle g_k, H(0)g_k\rangle$ is adapted from (Xu et al., 2024, Lemma D.1) as it is independent on the update direction, as long as $g_k$ is normalized.

**Lemma 20** ((Xu et al., 2024, Lemma D.1)). *Let* $g_k = \frac{\nabla\mathsf{L}_\gamma(Z^k)+\delta_k}{\|\nabla\mathsf{L}_\gamma(Z^k)+\delta_k\|}$ *and* $H(\kappa)$ *defined in* (47), *then it holds that*

$$\langle g_k, H(0)g_k\rangle$$

$$= \frac{1}{\alpha^2\|\nabla\mathsf{L}_\gamma(W_1^k, W_2^k) + \delta_k\|^2} \cdot \frac{d^2}{ds^2}M(s)\Big|_{s=0}$$

$$\leq \frac{1}{\alpha^2\|\nabla\mathsf{L}_\gamma(W_1^k, W_2^k) + \delta_k\|^2} \left(\langle\nabla\ell_\gamma(A(s)), \frac{d^2}{ds^2}A(s)\rangle\Big|_{s=0} + \sigma_{\max}^2(X_\gamma)\|\frac{d}{ds}A(s+\kappa)\|^2\Big|_{s=0}\right)$$

$$\leq \sigma_{\max}^2(X_\gamma)(\sigma_{\max}^2(W_1^k) + \sigma_{\max}^2(W_2^k)) + \sqrt{2\sigma_{\max}^2(X_\gamma)(\mathsf{L}_\gamma(W_1^k, W_2^k) - \mathsf{L}_\gamma^*)}. \tag{49}$$

To show the boundedness of $|\langle g_k, (H(\kappa) - H(0))g_k\rangle|$, we define the loss at the intermediate point and the product of $W_1^k - s\alpha(\nabla_{W_1}\mathsf{L}_\gamma(W_1^k, W_2^k) + \delta_k)$ and $W_2^k - s\alpha\nabla_{W_2}\mathsf{L}_\gamma(W_1^k, W_2^k)$ as follows.

$$M(s) = \mathsf{L}_\gamma\left(W_1^k - s\alpha\left(\nabla_{W_1}\mathsf{L}_\gamma(W_1^k, W_2^k) + \delta_k\right), W_2^k - s\alpha\nabla_{W_2}\mathsf{L}_\gamma(W_1^k, W_2^k)\right)$$

$$A(s) = W^k - s\alpha\left(\left(\nabla_{W_1}\mathsf{L}_\gamma(W_1^k, W_2^k) + \delta_k\right)W_2^k + W_1^k\nabla_{W_2}\mathsf{L}_\gamma(W_1^k, W_2^k)\right)$$

$$+ s^2\alpha^2\left(\nabla_{W_1}\mathsf{L}_\gamma(W_1^k, W_2^k) + \delta_k\right)\nabla_{W_2}\mathsf{L}_\gamma(W_1^k, W_2^k)$$

where $W^k = W_1^k W_2^k$ and $\ell_\gamma = \ell_{\text{val}} + \gamma\ell_{\text{trn}}$. In this way, we have

$$M(0) = \mathsf{L}_\gamma(W_1^k, W_2^k), \quad M(1) = \mathsf{L}_\gamma(W_1^{k+1}, W_2^{k+1}), \quad \text{and} \quad M(s) = \ell_\gamma(A(s)).$$

Then establishing the bound of $\|H(\kappa) - H(0)\|$ depends on the amount of $\|A(\kappa) - A(0)\|$ because $H(\kappa) = \nabla^2\widetilde{\mathsf{L}}_\gamma(M(\kappa)) = \nabla^2\widetilde{\mathsf{L}}_\gamma(\ell_\gamma(A(\kappa)))$.

**Lemma 21.** *For any* $\kappa \in [0, 1)$, *if* $\|\delta_k\| \leq \delta$ *and* $\sigma_{\min}(W_1^k) > 0, \sigma_{\min}(W_2^k) > 0$, *then it holds that*

$$\|A(\kappa) - A(0)\| \leq \alpha\left(\sigma_{\max}^2(W_1^k) + \sigma_{\max}^2(W_2^k)\right)\sqrt{2\sigma_{\max}^2(X_\gamma)(\mathsf{L}_\gamma(W_1^k, W_2^k) - \mathsf{L}_\gamma^*)}$$

$$+ \alpha\delta\sigma_{\max}(W_2^k) + 2\alpha^2\sigma_{\max}^2(X_\gamma)\sigma_{\max}(W^k)(\mathsf{L}_\gamma(W_1^k, W_2^k) - \mathsf{L}_\gamma^*)$$

$$+ \alpha^2\delta\sigma_{\max}(W_1^k)\sqrt{2\sigma_{\max}^2(X_\gamma)(\mathsf{L}_\gamma(W_1^k, W_2^k) - \mathsf{L}_\gamma^*)}. \tag{50}$$

**Proof:** According to the definition, we have

$$\|A(\kappa) - A(0)\| = \left\| -\kappa\alpha \left( \left( \nabla_{W_1} \mathsf{L}_\gamma(W_1^k, W_2^k) + \delta_k \right) W_2^k + W_1^k \nabla_{W_2} \mathsf{L}_\gamma(W_1^k, W_2^k) \right) \right.$$
$$\left. + \kappa^2 \alpha^2 \left( \nabla_{W_1} \mathsf{L}_\gamma(W_1^k, W_2^k) + \delta_k \right) \nabla_{W_2} \mathsf{L}_\gamma(W_1^k, W_2^k) \right\|$$
$$\leq \kappa\alpha \left\| \left( \nabla_{W_1} \mathsf{L}_\gamma(W_1^k, W_2^k) + \delta_k \right) W_2^k + W_1^k \nabla_{W_2} \mathsf{L}_\gamma(W_1^k, W_2^k) \right\|$$
$$+ \kappa^2 \alpha^2 \left\| \left( \nabla_{W_1} \mathsf{L}_\gamma(W_1^k, W_2^k) + \delta_k \right) \nabla_{W_2} \mathsf{L}_\gamma(W_1^k, W_2^k) \right\|$$
$$\overset{(22)\&(23)}{\leq} \kappa\alpha \left( \sigma_{\max}^2(W_1^k) + \sigma_{\max}^2(W_2^k) \right) \sqrt{2\sigma_{\max}^2(X_\gamma)(\mathsf{L}_\gamma(W_1^k, W_2^k) - \mathsf{L}_\gamma^*)}$$
$$+ \kappa\alpha\delta\sigma_{\max}(W_2^k) + 2\kappa^2\alpha^2\sigma_{\max}^2(X_\gamma)\sigma_{\max}(W^k)(\mathsf{L}_\gamma(W_1^k, W_2^k) - \mathsf{L}_\gamma^*)$$
$$+ \kappa^2\alpha^2\delta\sigma_{\max}(W_1^k)\sqrt{2\sigma_{\max}^2(X_\gamma)(\mathsf{L}_\gamma(W_1^k, W_2^k) - \mathsf{L}_\gamma^*)}.$$

Letting $\kappa < 1$, we get the conclusion.

**Lemma 22.** *For any $\kappa \in [0, 1)$, let $\delta := \|\delta_k\|$, then it holds that*

$$\langle g_k, H(\kappa)g_k \rangle \leq \sigma_{\max}^2(X_\gamma)(\sigma_{\max}^2(W_1^k) + \sigma_{\max}^2(W_2^k)) + 3\alpha\delta\sigma_{\max}^2(X_\gamma)\sigma_{\max}(W_2^k)$$
$$+ \left( 1 + 3\alpha\sigma_{\max}^2(X_\gamma)(\sigma_{\max}^2(W_1^k) + \sigma_{\max}^2(W_2^k)) \right) \sqrt{2\sigma_{\max}^2(X_\gamma)(\mathsf{L}_\gamma(W_1^k, W_2^k) - \mathsf{L}_\gamma^*)}$$
$$+ 3\alpha^2\sigma_{\max}^2(X_\gamma)\delta\sigma_{\max}(W_1^k)\sqrt{2\sigma_{\max}^2(X_\gamma)(\mathsf{L}_\gamma(W_1^k, W_2^k) - \mathsf{L}_\gamma^*)}$$
$$+ 6\alpha^2\sigma_{\max}^4(X_\gamma)\sigma_{\max}(W^k)(\mathsf{L}_\gamma(W_1^k, W_2^k) - \mathsf{L}_\gamma^*) =: L_k \qquad (51)$$

**Proof:** First, we observe that $\langle g_k, H(\kappa)g_k \rangle$ is the second-order directional derivative of $\mathsf{L}_\gamma$ with respect to the update direction, i.e.

$$\langle g_k, H(\kappa)g_k \rangle = \frac{1}{\alpha^2\|\nabla\mathsf{L}_\gamma(W_1^k, W_2^k) + \delta_k\|^2} \cdot \frac{d^2}{ds^2}M(s + \kappa)\Big|_{s=0} \qquad (52)$$

For the directional derivative, we have

$$\frac{d^2}{ds^2}M(s + \kappa)\Big|_{s=0}$$
$$= \frac{d^2}{ds^2}\ell_\gamma(A(s + \kappa))\Big|_{s=0}$$
$$= \frac{d}{ds}\langle \nabla\ell_\gamma(A(s + \kappa)), \frac{d}{ds}A(s + \kappa)\rangle\Big|_{s=0}$$
$$= \langle \nabla\ell_\gamma(A(s + \kappa)), \frac{d^2}{ds^2}A(s + \kappa)\rangle\Big|_{s=0} + \langle \frac{d}{ds}A(s + \kappa), \nabla^2\ell_\gamma(A(s + \kappa))\frac{d}{ds}A(s + \kappa)\rangle\Big|_{s=0}$$
$$\overset{(a)}{\leq} \langle \nabla\ell_\gamma(A(s + \kappa)), \frac{d^2}{ds^2}A(s + \kappa)\rangle\Big|_{s=0} + \sigma_{\max}^2(X_\gamma)\|\frac{d}{ds}A(s + \kappa)\|^2\Big|_{s=0} \qquad (53)$$

where (a) uses the fact that $\ell_\gamma$ is $\sigma_{\max}^2(X_\gamma)$ smooth. Then we bound $\langle \nabla\ell_\gamma(A(s + \kappa)), \frac{d^2}{ds^2}A(s + \kappa)\rangle\Big|_{s=0}$ and $\|\frac{d}{ds}A(s + \kappa)\|^2\Big|_{s=0}$ as follows.

$$\langle \nabla\ell_\gamma(A(s + \kappa)), \frac{d^2}{ds^2}A(s + \kappa)\rangle\Big|_{s=0}$$
$$= 2\langle \nabla\ell_\gamma(A(\kappa)), \alpha^2\left( \nabla_{W_1}\mathsf{L}_\gamma(W_1^k, W_2^k) + \delta_k \right) \nabla_{W_2}\mathsf{L}_\gamma(W_1^k, W_2^k)\rangle$$
$$= 2\langle \nabla\ell_\gamma(A(\kappa)) - \nabla\ell_\gamma(A(0)), \alpha^2\left( \nabla_{W_1}\mathsf{L}_\gamma(W_1^k, W_2^k) + \delta_k \right) \nabla_{W_2}\mathsf{L}_\gamma(W_1^k, W_2^k)\rangle$$
$$+ 2\langle \nabla\ell_\gamma(A(0)), \alpha^2\left( \nabla_{W_1}\mathsf{L}_\gamma(W_1^k, W_2^k) + \delta_k \right) \nabla_{W_2}\mathsf{L}_\gamma(W_1^k, W_2^k)\rangle$$
$$\leq 2\alpha^2\sigma_{\max}^2(X_\gamma)\|A(\kappa) - A(0)\|\|\left( \nabla_{W_1}\mathsf{L}_\gamma(W_1^k, W_2^k) + \delta_k \right)\nabla_{W_2}\mathsf{L}_\gamma(W_1^k, W_2^k)\|$$
$$+ \langle \nabla\ell_\gamma(A(s)), \frac{d^2}{ds^2}A(s)\rangle\Big|_{s=0} \qquad (54)$$

On the other hand,

$$\|\frac{d}{ds}A(s + \kappa)\|^2\Big|_{s=0}$$

$$\begin{aligned}
&= \big\| \alpha \left( \nabla_{W_1} \mathsf{L}_\gamma(W_1^k, W_2^k) + \delta_k \right) W_2^k + \alpha W_1^k \nabla_{W_2} \mathsf{L}_\gamma(W_1^k, W_2^k) \\
&\quad -2\kappa\alpha^2 \left( \nabla_{W_1} \mathsf{L}_\gamma(W_1^k, W_2^k) + \delta_k \right) \nabla_{W_2} \mathsf{L}_\gamma(W_1^k, W_2^k) \big\|^2 \\
&= \alpha^2 \left\| \left( \nabla_{W_1} \mathsf{L}_\gamma(W_1^k, W_2^k) + \delta_k \right) W_2^k + W_1^k \nabla_{W_2} \mathsf{L}_\gamma(W_1^k, W_2^k) \right\|^2 \\
&\quad + 4\kappa^2\alpha^4 \left\| \left( \nabla_{W_1} \mathsf{L}_\gamma(W_1^k, W_2^k) + \delta_k \right) \nabla_{W_2} \mathsf{L}_\gamma(W_1^k, W_2^k) \right\|^2 \\
&\quad - 4\kappa\alpha^3 \left\langle \left( \nabla_{W_1} \mathsf{L}_\gamma(W_1^k, W_2^k) + \delta_k \right) \nabla_{W_2} \mathsf{L}_\gamma(W_1^k, W_2^k), \right. \\
&\qquad \left. \left( \nabla_{W_1} \mathsf{L}_\gamma(W_1^k, W_2^k) + \delta_k \right) W_2^k + W_1^k \nabla_{W_2} \mathsf{L}_\gamma(W_1^k, W_2^k) \right\rangle \\
&\leq \left\| \frac{d}{ds} A(s) \right\|^2 \Big|_{s=0} + 4\kappa^2\alpha^4 \left\| \left( \nabla_{W_1} \mathsf{L}_\gamma(W_1^k, W_2^k) + \delta_k \right) \nabla_{W_2} \mathsf{L}_\gamma(W_1^k, W_2^k) \right\|^2 \\
&\quad - 4\kappa\alpha^3 \left\langle \left( \nabla_{W_1} \mathsf{L}_\gamma(W_1^k, W_2^k) + \delta_k \right) \nabla_{W_2} \mathsf{L}_\gamma(W_1^k, W_2^k), \right. \\
&\qquad \left. \left( \nabla_{W_1} \mathsf{L}_\gamma(W_1^k, W_2^k) + \delta_k \right) W_2^k + W_1^k \nabla_{W_2} \mathsf{L}_\gamma(W_1^k, W_2^k) \right\rangle
\end{aligned} \tag{55}$$

Plugging (54) and (55) into (53), we get

$$\begin{aligned}
&\frac{d^2}{ds^2} M(s+\kappa) \Big|_{s=0} \\
&\leq 2\alpha^2 \sigma_{\max}^2(X_\gamma) \| A(\kappa) - A(0) \| \| \left( \nabla_{W_1} \mathsf{L}_\gamma(W_1^k, W_2^k) + \delta_k \right) \nabla_{W_2} \mathsf{L}_\gamma(W_1^k, W_2^k) \| \\
&\quad + \left\langle \nabla \ell_\gamma(A(s)), \frac{d^2}{ds^2} A(s) \right\rangle \Big|_{s=0} + \sigma_{\max}^2(X_\gamma) \left\| \frac{d}{ds} A(s) \right\|^2 \Big|_{s=0} \\
&\quad + 4\kappa^2\alpha^4 \sigma_{\max}^2(X_\gamma) \left\| \left( \nabla_{W_1} \mathsf{L}_\gamma(W_1^k, W_2^k) + \delta_k \right) \nabla_{W_2} \mathsf{L}_\gamma(W_1^k, W_2^k) \right\|^2 \\
&\quad - 4\kappa\alpha^3 \sigma_{\max}^2(X_\gamma) \left\langle \left( \nabla_{W_1} \mathsf{L}_\gamma(W_1^k, W_2^k) + \delta_k \right) \nabla_{W_2} \mathsf{L}_\gamma(W_1^k, W_2^k), \right. \\
&\qquad \left. \left( \nabla_{W_1} \mathsf{L}_\gamma(W_1^k, W_2^k) + \delta_k \right) W_2^k + W_1^k \nabla_{W_2} \mathsf{L}_\gamma(W_1^k, W_2^k) \right\rangle \\
&\leq \alpha^2 \sigma_{\max}^2(X_\gamma) \| A(\kappa) - A(0) \| \| \nabla \mathsf{L}_\gamma(W_1^k, W_2^k) + \delta_k \|^2 \\
&\quad + \left[ \sigma_{\max}^2(X_\gamma)(\sigma_{\max}^2(W_1^k) + \sigma_{\max}^2(W_2^k)) + \sqrt{2\sigma_{\max}^2(X_\gamma)(\widetilde{\mathsf{L}}_\gamma(W_1^k, W_2^k) - \mathsf{L}_\gamma^*)} \right] \alpha^2 \| \nabla \mathsf{L}_\gamma(W_1^k, W_2^k) + \delta_k \|^2 \\
&\quad + 4\kappa^2\alpha^4 \sigma_{\max}^4(X_\gamma) \sigma_{\max}(W^k)(\mathsf{L}_\gamma(W_1^k, W_2^k) - \mathsf{L}_\gamma^*) \| \nabla \mathsf{L}_\gamma(W_1^k, W_2^k) + \delta_k \|^2 \\
&\quad + 2\kappa^2\alpha^4 \sigma_{\max}^2(X_\gamma) \delta \sigma_{\max}(W_1^k) \sqrt{2\sigma_{\max}^2(X_\gamma)(\mathsf{L}_\gamma(W_1^k, W_2^k) - \mathsf{L}_\gamma^*)} \| \nabla \mathsf{L}_\gamma(W_1^k, W_2^k) + \delta_k \|^2 \\
&\quad + 2\kappa\alpha^3 \sigma_{\max}^2(X_\gamma) \left( \sigma_{\max}^2(W_1^k) + \sigma_{\max}^2(W_2^k) \right) \sqrt{2\sigma_{\max}^2(X_\gamma)(\mathsf{L}_\gamma(W_1^k, W_2^k) - \mathsf{L}_\gamma^*)} \| \nabla \mathsf{L}_\gamma(W_1^k, W_2^k) + \delta_k \|^2 \\
&\quad + 2\kappa\alpha^3 \sigma_{\max}^2(X_\gamma) \delta \sigma_{\max}(W_2^k) \| \nabla \mathsf{L}_\gamma(W_1^k, W_2^k) + \delta_k \|^2
\end{aligned}$$

where the last inequality follows from Lemma 9 and (49). Plugging the above bound and the bound of $\| A(\kappa) - A(0) \|$ in (50) into (52) and note that $\kappa \leq 1$, we obtain that

$$\begin{aligned}
\langle g_k, H(\kappa) g_k \rangle &= \frac{1}{\alpha^2 \| \nabla \mathsf{L}_\gamma(W_1^k, W_2^k) + \delta_k \|^2} \cdot \frac{d^2}{ds^2} M(s+\kappa) \Big|_{s=0} \\
&\leq \alpha \sigma_{\max}^2(X_\gamma) \left( \sigma_{\max}^2(W_1^k) + \sigma_{\max}^2(W_2^k) \right) \sqrt{2\sigma_{\max}^2(X_\gamma)(\mathsf{L}_\gamma(W_1^k, W_2^k) - \mathsf{L}_\gamma^*)} \\
&\quad + \alpha\delta \sigma_{\max}^2(X_\gamma) \sigma_{\max}(W_2^k) + 2\alpha^2 \sigma_{\max}^4(X_\gamma) \sigma_{\max}(W^k)(\mathsf{L}_\gamma(W_1^k, W_2^k) - \mathsf{L}_\gamma^*) \\
&\quad + \alpha^2 \delta \sigma_{\max}^2(X_\gamma) \sigma_{\max}(W_1^k) \sqrt{2\sigma_{\max}^2(X_\gamma)(\mathsf{L}_\gamma(W_1^k, W_2^k) - \mathsf{L}_\gamma^*)} \\
&\quad + \sigma_{\max}^2(X_\gamma)(\sigma_{\max}^2(W_1^k) + \sigma_{\max}^2(W_2^k)) + \sqrt{2\sigma_{\max}^2(X_\gamma)(\mathsf{L}_\gamma(W_1^k, W_2^k) - \mathsf{L}_\gamma^*)} \\
&\quad + 4\alpha^2 \sigma_{\max}^4(X_\gamma) \sigma_{\max}(W^k)(\mathsf{L}_\gamma(W_1^k, W_2^k) - \mathsf{L}_\gamma^*) \\
&\quad + 2\alpha^2 \sigma_{\max}^2(X_\gamma) \delta \sigma_{\max}(W_1^k) \sqrt{2\sigma_{\max}^2(X_\gamma)(\mathsf{L}_\gamma(W_1^k, W_2^k) - \mathsf{L}_\gamma^*)} \\
&\quad + 2\alpha \sigma_{\max}^2(X_\gamma) \left( \sigma_{\max}^2(W_1^k) + \sigma_{\max}^2(W_2^k) \right) \sqrt{2\sigma_{\max}^2(X_\gamma)(\mathsf{L}_\gamma(W_1^k, W_2^k) - \mathsf{L}_\gamma^*)} \\
&\quad + 2\alpha \sigma_{\max}^2(X_\gamma) \delta \sigma_{\max}(W_2^k) \\
&= \sigma_{\max}^2(X_\gamma)(\sigma_{\max}^2(W_1^k) + \sigma_{\max}^2(W_2^k)) + 3\alpha\delta \sigma_{\max}^2(X_\gamma) \sigma_{\max}(W_2^k)
\end{aligned}$$

$$+ \left(1 + 3\alpha\sigma_{\max}^2(X_\gamma)(\sigma_{\max}^2(W_1^k) + \sigma_{\max}^2(W_2^k))\right) \sqrt{2\sigma_{\max}^2(X_\gamma)(\mathsf{L}_\gamma(W_1^k, W_2^k) - \mathsf{L}_\gamma^*)}$$

$$+ 3\alpha^2\sigma_{\max}^2(X_\gamma)\delta\sigma_{\max}(W_1^k)\sqrt{2\sigma_{\max}^2(X_\gamma)(\mathsf{L}_\gamma(W_1^k, W_2^k) - \mathsf{L}_\gamma^*)}$$

$$+ 6\alpha^2\sigma_{\max}^4(X_\gamma)\sigma_{\max}(W^k)(\mathsf{L}_\gamma(W_1^k, W_2^k) - \mathsf{L}_\gamma^*)$$

which completes the proof.

### F.5 PROOF OF THEOREM 2

We restate Theorem 2 in a formal way as follows.

**Theorem 6** (Almost linear convergence rate). *Suppose that Assumption 2 holds, letting $\alpha_1$ be the smallest positive solution of the following equation of $\alpha$*

$$5\alpha\sigma_{\max}^2(X_\gamma)(\mathsf{L}_\gamma(W_1^0, W_2^0) - \mathsf{L}_\gamma^*) = (1 - \exp(-\sqrt{\alpha}))(c_1 + c_2(2 - \exp(\sqrt{\alpha})))\sigma_*^2(X_\gamma). \quad (56)$$

*If the above equation does not have a positive solution, set $\alpha_1 = \infty$. Similarly, let $\alpha_2$ be the positive solution of $\alpha L = 1$ with $L$ defined in (64). Moreover, let $\alpha_3 < (\log(2 + \frac{c_1}{c_2}))^2$. Then for any $0 < \alpha < \min\{\alpha_1, \alpha_2, \alpha_3\}$ and $0 < \beta \le \alpha_2$, there exists $T = \mathcal{O}(\log\gamma + \log\epsilon^{-1} + K)$ such that $\delta \le \min\left\{\sqrt{\frac{\alpha\mu^2(\mathsf{L}_\gamma(W_1^0, W_2^0) - \mathsf{L}_\gamma^*)}{3}}, \mathcal{O}(\epsilon^{1/4})\right\}$ and*

$$\left(2\alpha\delta\sqrt{c_2^1\exp(\sqrt{\alpha})} + 2\alpha^2\delta^2 + \frac{3\alpha^2\delta^2 c_2\exp(\sqrt{\alpha})\sigma_{\max}^2(X_\gamma)}{\mu}\right) K$$

$$\le \frac{\alpha c_2\exp(\sqrt{\alpha})\sigma_{\max}^2(X_\gamma)}{\mu}\left(\mathsf{L}_\gamma(W_1^0, W_2^0) - \mathsf{L}_\gamma^*\right). \quad (57)$$

*Therefore, when $\gamma = \mathcal{O}(\epsilon^{-0.5})$, $\mu = \mathcal{O}(\gamma), L = \mathcal{O}(\gamma)$, $\alpha = \mathcal{O}(\gamma^{-1})$, we have $\alpha\mu = \mathcal{O}(1)$ and to achieve the $(\epsilon, \epsilon)$ stationary point, i.e. $\mathsf{L}_{\mathrm{val}}(W_1^k, W_2^k) - \min_{W_1, W_2} \mathsf{L}_{\mathrm{val}}(W_1, W_2) \le \epsilon$ and $\mathsf{L}_{\mathrm{trn}}(W_1^k, W_2^k) - \min_{W_2} \mathsf{L}_{\mathrm{trn}}(W_1, W_2) \le \epsilon$, one need $KT = \mathcal{O}(\log(\epsilon^{-1})^2)$ iterations.*

**Proof:** Denote $W^k = W_1^k W_2^k$ and $D_k = (W_1^k)^\top W_1^k - (W_2^k)^\top W_2^k$ as the imbalanced matrix. We will prove this theorem by induction. Define the following properties as

- $\mathrm{P}_1(k)$: $\mathsf{L}_\gamma(W_1^{k+1}, W_2^{k+1}) - \mathsf{L}_\gamma^* \le \left(1 - \frac{\alpha\mu}{2}\right)\left(\mathsf{L}_\gamma(W_1^k, W_2^k) - \mathsf{L}_\gamma^*\right) + \frac{3\alpha\delta^2}{4}$
- $\mathrm{P}_2(k)$: $w_1 \le \sigma_{\min}(W^{k+1}) \le \sigma_{\max}(W^{k+1}) \le w_2$
- $\mathrm{P}_3(k)$: $\|D_{k+1} - D_0\| \le \frac{5\alpha c_2\exp(\sqrt{\alpha})\sigma_{\max}^2(X_\gamma)}{(c_1 + 2c_2(1 - \exp(\sqrt{\alpha})))\sigma_*^2(X_\gamma)}\left(\mathsf{L}_\gamma(W_1^0, W_2^0) - \mathsf{L}_\gamma^*\right)$
- $\mathrm{P}_4(k)$: $c_1^i + 2c_2^i(1 - \exp(\sqrt{\alpha})) \le \sigma_{\min}^2(W_i^{k+1}) \le \sigma_{\max}^2(W_i^{k+1}) \le c_2^i\exp(\sqrt{\alpha}), \ \ i = 1, 2$

where $c_1 = c_1^1 + c_1^2, c_2 = c_2^1 + c_2^2$. Suppose that the above properties hold at iteration $0, \cdots, k-1$, we aim to prove that $\mathrm{P}_1(k), \mathrm{P}_2(k), \mathrm{P}_3(k), \mathrm{P}_4(k)$ hold recursively.

**Step 1:** $\mathrm{P}_1(k)$ **holds.** If we can prove the following uniform bounds of the local PL, smoothness constants and the lower-level error

$$\mu_k \ge \mu, \ \ L_k \le L, \ \ \|\delta_k\| \le \delta \quad (58)$$

then it holds that

$$1 - \alpha\mu_k + 2\alpha^2 L_k\mu_k = 1 - \mu_k(\alpha - 2\alpha^2 L_k) \qquad \text{and} \qquad \left(\frac{\alpha}{2} + \alpha^2 L_k\right)\|\delta_k\|^2 \le \left(\frac{\alpha}{2} + \alpha^2 L\right)\delta^2$$

$$\overset{(a)}{\le} 1 - \mu(\alpha - 2\alpha^2 L_k) \qquad\qquad\qquad \overset{(b)}{\le} \frac{3\alpha\delta^2}{4}$$

$$\overset{(c)}{\le} 1 - \frac{\alpha\mu}{2} \quad (59)$$

where (a) comes from $\alpha \le \frac{1}{L} \le \frac{1}{L_k}$, (b) and (c) are due to $\alpha \le \frac{1}{4L}$ and $L_k \le L$. Then according to Theorem 1, we get the conclusion.

Next, we start to prove (58). According to $P_4(k-1)$, the lower bound of $\mu_k$ is earned by

$$\mu_k = (\sigma^2_{\min}(W_1^k) + \sigma^2_{\min}(W_2^k))\sigma^2_*(X_\gamma) \geq \left(c_1 + c_2(2 - \exp(\sqrt{\alpha}))\right)\sigma^2_*(X_\gamma) =: \mu \quad (60)$$

where $c_1 = c_1^1 + c_1^2$ and $c_2 = c_2^1 + c_2^2$. We then prove the upper bound of $\|\delta_k\|$. By definition,

$$
\begin{aligned}
\|\delta_k\| &= \gamma^2 \|\nabla_{W_1}\mathsf{L}_{\mathrm{trn}}(W_1^k, W_3^{k+1}) - \nabla_{W_1}\mathsf{L}_{\mathrm{trn}}(W_1^k)\| \\
&\leq \gamma^2 \sigma^2_{\max}(X_{\mathrm{trn}})\sigma^2_{\max}(W_1^k)\,\mathrm{dist}(W_3^{k+1}, \mathcal{S}(W_1^k)) \\
&\leq 2\gamma^2 \sigma^4_{\max}(X_{\mathrm{trn}})\sigma^4_{\max}(W_1^k)\left(1 - \frac{\sigma^2_{\min}(X_{\mathrm{trn}})\sigma^2_{\min}(W_1^k)}{\sigma^2_{\max}(X_{\mathrm{trn}})\sigma^2_{\max}(W_1^k)}\right)^T \left(\mathsf{L}_{\mathrm{trn}}(W_1^k, W_3^0) - \mathsf{L}^*_{\mathrm{trn}}(W_1^k)\right) \\
&\leq 2\gamma^2 \sigma^4_{\max}(X_{\mathrm{trn}})\sigma^4_{\max}(W_1^k)\left(1 - \frac{\sigma^2_{\min}(X_{\mathrm{trn}})\sigma^2_{\min}(W_1^k)}{\sigma^2_{\max}(X_{\mathrm{trn}})\sigma^2_{\max}(W_1^k)}\right)^T \mathsf{L}_{\mathrm{trn}}(W_1^k, W_3^0)
\end{aligned}
$$

where the first inequality is due to the Lipschitz continuity of $\mathsf{L}_{\mathrm{trn}}(W_1^k, \cdot)$, the second inequality comes from Lemma 7 and Lemma 17. According to $P_4(k-1)$, $\sigma_{\max}(W_1^k), \sigma_{\min}(W_1^k)$ is upper and lower bounded, so the upper bound of $\|\delta_k\| \leq \delta$ exists and it exponentially decreases with $T$.

Finally, we prove the upper bound of $L_k$. As $\mathsf{L}_\gamma(W_1^k, W_2^k) = \widetilde{\mathsf{L}}_\gamma(W_1^k, W_2^k)$ when $\sigma_{\min}(W_i^k) > 0$, according to the definition, it holds that

$$
\begin{aligned}
L_k \overset{(51)}{=} \ & \sigma^2_{\max}(X_\gamma)(\sigma^2_{\max}(W_1^k) + \sigma^2_{\max}(W_2^k)) + 3\alpha\delta\sigma^2_{\max}(X_\gamma)\sigma_{\max}(W_2^k) \\
& + \left(1 + 3\alpha\sigma^2_{\max}(X_\gamma)(\sigma^2_{\max}(W_1^k) + \sigma^2_{\max}(W_2^k))\right)\sqrt{2\sigma^2_{\max}(X_\gamma)(\mathsf{L}_\gamma(W_1^k, W_2^k) - \mathsf{L}^*_\gamma)} \\
& + 3\alpha^2\sigma^2_{\max}(X_\gamma)\delta\sigma_{\max}(W_1^k)\sqrt{2\sigma^2_{\max}(X_\gamma)(\mathsf{L}_\gamma(W_1^k, W_2^k) - \mathsf{L}^*_\gamma)} \\
& + 6\alpha^2\sigma^4_{\max}(X_\gamma)\sigma_{\max}(W^k)(\mathsf{L}_\gamma(W_1^k, W_2^k) - \mathsf{L}^*_\gamma). \quad (61)
\end{aligned}
$$

Therefore, to prove the upper bound of $L_k$, we need the following upper bounds

$$
\sigma^2_{\max}(W_1^k) + \sigma^2_{\max}(W_2^k) \overset{P_4(k-1)}{\leq} c_2\exp(\sqrt{\alpha})
$$

$$
\begin{aligned}
\mathsf{L}_\gamma(W_1^k, W_2^k) - \mathsf{L}^*_\gamma &\overset{P_1(k-1)}{\leq} (1 - \frac{\alpha\mu}{2})^k(\mathsf{L}_\gamma(W_1^0, W_2^0) - \mathsf{L}^*_\gamma) + \frac{3\alpha\delta^2}{4}\sum_{\kappa=0}^{k-1}(1 - \frac{\alpha\mu}{2})^\kappa \\
&\leq \left(1 - \frac{\alpha\mu}{2}\right)^k(\mathsf{L}_\gamma(W_1^0, W_2^0) - \mathsf{L}^*_\gamma) + \frac{3\delta^2}{2\mu}\mathbb{1}_{k\geq 1} \quad (62) \\
&\overset{(a)}{\leq} \mathsf{L}_\gamma(W_1^0, W_2^0) - \mathsf{L}^*_\gamma \quad (63)
\end{aligned}
$$

$$
\sigma^2_{\max}(W_i^k) \overset{P_4(k-1)}{\leq} c_2^i\exp(\sqrt{\alpha}), \quad \sigma_{\max}(W^k) \overset{P_2(k-1)}{\leq} w_2, \quad \|\delta_k\| \leq \delta.
$$

where (a) comes from $\delta \leq \sqrt{\frac{\alpha\mu^2(\mathsf{L}_\gamma(W_1^0, W_2^0) - \mathsf{L}^*_\gamma)}{3}}$. Plugging the above upper bounds to (61), we get

$$
\begin{aligned}
L_k \leq \ & \sigma^2_{\max}(X_\gamma)c_2\exp(\sqrt{\alpha}) + 3\alpha\delta\sigma^2_{\max}(X_\gamma)\sqrt{c_2\exp(\sqrt{\alpha})} \\
& + \left(1 + 3\alpha\sigma^2_{\max}(X_\gamma)c_2\exp(\sqrt{\alpha})\right)\sqrt{2\sigma^2_{\max}(X_\gamma)(\mathsf{L}_\gamma(W_1^k, W_2^k) - \mathsf{L}^*_\gamma)} \\
& + 3\alpha^2\sigma^2_{\max}(X_\gamma)\delta\sqrt{c_2\exp(\sqrt{\alpha})}\sqrt{2\sigma^2_{\max}(X_\gamma)(\mathsf{L}_\gamma(W_1^k, W_2^k) - \mathsf{L}^*_\gamma)} \\
& + 6\alpha^2\sigma^4_{\max}(X_\gamma)w_2(\mathsf{L}_\gamma(W_1^k, W_2^k) - \mathsf{L}^*_\gamma) =: L. \quad (64)
\end{aligned}
$$

After obtaining the bounds for $\mu_k, \|\delta_k\|, L_k$, $P_1(k)$ holds because of Theorem 1 and (59).

**Step 2:** $P_2(k)$ **holds.** Since $(\ell_{\mathrm{val}} + \gamma\ell_{\mathrm{trn}})(W)$ is $\sigma^2_{\max}(X_\gamma)$-Lipschitz smooth, we have

$$(\ell_{\mathrm{val}} + \gamma\ell_{\mathrm{trn}})(W^{k+1}) \leq (\ell_{\mathrm{val}} + \gamma\ell_{\mathrm{trn}})(W) + \langle\nabla(\ell_{\mathrm{val}} + \gamma\ell_{\mathrm{trn}})(W), W^{k+1} - W\rangle + \frac{\sigma^2_{\max}(X_\gamma)}{2}\|W^{k+1} - W\|^2.$$

Setting $W = W^* \in \arg\min_W (\ell_{\text{val}} + \gamma\ell_{\text{trn}})(W)$ yields

$$(\ell_{\text{val}} + \gamma\ell_{\text{trn}})(W^{k+1}) \le \min_W (\ell_{\text{val}} + \gamma\ell_{\text{trn}})(W) + \frac{\sigma^2_{\max}(X_\gamma)}{2}\|W^{k+1} - W^*\|^2.$$

and thus

$$\begin{aligned}
\frac{\sigma^2_*}{2}(X_\gamma)\|W^{k+1} - W^*\|^2 &\le (\ell_{\text{val}} + \gamma\ell_{\text{trn}})(W^{k+1}) - \min_W (\ell_{\text{val}} + \gamma\ell_{\text{trn}})(W) \\
&= \widetilde{\mathsf{L}}_\gamma(W_1^{k+1}, W_2^{k+1}) - \min_{W_1, W_2} \widetilde{\mathsf{L}}_\gamma(W_1, W_2) \\
&= \mathsf{L}_\gamma(W_1^{k+1}, W_2^{k+1}) - \mathsf{L}^*_\gamma.
\end{aligned}$$

Also, $(\ell_{\text{val}} + \gamma\ell_{\text{trn}})(W)$ is $\sigma^2_*(X_\gamma)$-PL, so the quadratic growth condition with holds by Lemma 6

$$2\sigma^2_*(X_\gamma)\|W^{k+1} - W^*\|^2 \le (\ell_{\text{val}} + \gamma\ell_{\text{trn}})(W^{k+1}) - \min_W (\ell_{\text{val}} + \gamma\ell_{\text{trn}})(W) = \mathsf{L}_\gamma(W_1^{k+1}, W_2^{k+1}) - \mathsf{L}^*_\gamma$$

where $W^* = \arg\min_{W \in \arg\min_W (\ell_{\text{val}}+\gamma\ell_{\text{trn}})(W)} \|W^{k+1} - W\|^2$. As a result, we have

$$\begin{aligned}
\sigma_{\max}(W^{k+1}) &= \sigma_{\max}\left(W^{k+1} - W^* + W^*\right) \\
&\le \sigma_{\max}\left(W^*\right) + \|W^{k+1} - W^*\|_2 \\
&\le \sigma_{\max}\left(W^*\right) + \|W^{k+1} - W^*\| \\
&\le \sigma_{\max}\left(W^*\right) + \sqrt{\frac{1}{2\sigma^2_*(X_\gamma)}(\mathsf{L}_\gamma(W_1^{k+1}, W_2^{k+1}) - \mathsf{L}^*_\gamma)} \\
&\overset{(63)}{\le} \sigma_{\max}\left(W^*\right) + \sqrt{\frac{1}{2\sigma^2_*(X_\gamma)}(\mathsf{L}_\gamma(W_1^0, W_2^0) - \mathsf{L}^*_\gamma)} =: w_2
\end{aligned}$$

where the first inequality is derived from the Weyl's inequality. Similarly, the lower bound of singular value is achieved by the Weyl's inequality

$$\begin{aligned}
\sigma_{\min}(W^{k+1}) &= \sigma_{\min}\left(W^{k+1} - W^* + W^*\right) \\
&\ge \sigma_{\min}\left(W^*\right) - \|W^{k+1} - W^*\|_2 \\
&\ge \sigma_{\min}\left(W^*\right) - \|W^{k+1} - W^*\| \\
&\ge \sigma_{\min}\left(W^*\right) - \sqrt{\frac{1}{2\sigma^2_*(X_\gamma)}(\mathsf{L}_\gamma(W_1^0, W_2^0) - \mathsf{L}^*_\gamma)}.
\end{aligned}$$

As the singular value is always nonnegative, we can define a lower bound

$$w_1 := \left[\sigma_{\min}\left(W^*\right) - \sqrt{\frac{1}{2\sigma^2_*(X_\gamma)}(\mathsf{L}_\gamma(W_1^0, W_2^0) - \mathsf{L}^*_\gamma)}\right]_+ \tag{65}$$

which is strict positive when initializing $W_1^0, W_2^0$ close to the optimal.

**Step 3: $\mathrm{P}_3(k)$ holds.** Denoting $W^k = W_1^k W_2^k, \ell_\gamma = \ell_{\text{val}} + \gamma\ell_{\text{trn}}$ and utilizing the PBGD update

$$W_1^{k+1} = W_1^k - \alpha(\nabla\ell_\gamma(W^k)(W_2^k)^\top + \delta_k) \quad \text{and} \quad W_2^{k+1} = W_2^k - \alpha(W_1^k)^\top\nabla\ell_\gamma(W^k) \tag{66}$$

we can expand the difference of imbalance matrix as follows

$$\begin{aligned}
D_{k+1} - D_k &= W_1^{k+1,\top}W_1^{k+1} - W_2^{k+1,\top}W_2^{k+1} - (W_1^k)^\top W_1^k + (W_2^k)^\top W_2^k \\
&= \left(W_1^k - \alpha(\nabla\ell_\gamma(W^k)(W_2^k)^\top + \delta_k)\right)^\top \left(W_1^k - \alpha(\nabla\ell_\gamma(W^k)(W_2^k)^\top + \delta_k)\right) \\
&\quad - \left(W_2^k - \alpha(W_1^k)^\top\nabla\ell_\gamma(W^k)\right)^\top \left(W_2^k - \alpha(W_1^k)^\top\nabla\ell_\gamma(W^k)\right) - (W_1^k)^\top W_1^k + (W_2^k)^\top W_2^k \\
&= -2\alpha(W_1^k)^\top(\nabla\ell_\gamma(W^k)(W_2^k)^\top + \delta_k) + \alpha^2(\nabla\ell_\gamma(W^k)(W_2^k)^\top + \delta_k)^\top(\nabla\ell_\gamma(W^k)(W_2^k)^\top + \delta_k) \\
&\quad + 2\alpha\left((W_1^k)^\top\nabla\ell_\gamma(W^k)\right)^\top W_2^k - \alpha^2\left((W_1^k)^\top\nabla\ell_\gamma(W^k)\right)^\top \left((W_1^k)^\top\nabla\ell_\gamma(W^k)\right)
\end{aligned}$$

$$= -2\alpha (W_1^k)^\top \delta_k + \alpha^2 (\nabla \ell_\gamma(W^k)(W_2^k)^\top + \delta_k)^\top (\nabla \ell_\gamma(W^k)(W_2^k)^\top + \delta_k)$$
$$- \alpha^2 \left( (W_1^k)^\top \nabla \ell_\gamma(W^k) \right)^\top \left( (W_1^k)^\top \nabla \ell_\gamma(W^k) \right)$$

Taking the norm of both sides yields

$$\|D_{k+1} - D_k\| \le 2\alpha\delta\sigma_{\max}(W_1^k) + \alpha^2 \|\nabla\ell_\gamma(W^k)(W_2^k)^\top + \delta_k\|^2 + \alpha^2 \|(W_1^k)^\top \nabla\ell_\gamma(W^k)\|^2$$
$$\le 2\alpha\delta\sigma_{\max}(W_1^k) + 2\alpha^2(\sigma_{\max}^2(W_2^k)\|\nabla\ell_\gamma(W^k)\|^2 + \delta^2) + \alpha^2 \sigma_{\max}^2(W_1^k)\|\nabla\ell_\gamma(W^k)\|^2$$
$$= 2\alpha\delta\sigma_{\max}(W_1^k) + 2\alpha^2\delta^2 + \alpha^2(2\sigma_{\max}^2(W_2^k) + \sigma_{\max}^2(W_1^k))\|\nabla\ell_\gamma(W^k)\|^2$$
$$\le 2\alpha\delta\sqrt{c_2^1 \exp(\sqrt{\alpha})} + 2\alpha^2\delta^2 + 2\alpha^2 c_2 \exp(\sqrt{\alpha})\sigma_{\max}^2(X_\gamma)(\mathsf{L}_\gamma(W_1^k, W_2^k) - \mathsf{L}_\gamma^*)$$
$$\overset{(62)}{\le} 2\alpha\delta\sqrt{c_2^1 \exp(\sqrt{\alpha})} + 2\alpha^2\delta^2$$
$$+ 2\alpha^2 c_2 \exp(\sqrt{\alpha})\sigma_{\max}^2(X_\gamma)\left(\left(1 - \frac{\alpha\mu}{2}\right)^k (\mathsf{L}_\gamma(W_1^0, W_2^0) - \mathsf{L}_\gamma^*) + \frac{3\delta^2}{2\mu}\right) \quad (67)$$

where the third inequality is due to $\mathrm{P}_4(k-1)$ and $\ell_\gamma$ is $\sigma_{\max}^2(X_\gamma)$ Lipschitz smooth.

As a result, it follows

$$\|D_{k+1} - D_0\| \le \sum_{\kappa=0}^{k} \|D_{\kappa+1} - D_\kappa\|$$
$$\overset{(67)}{\le} \sum_{\kappa=0}^{k} 2\alpha\delta\sqrt{c_2^1 \exp(\sqrt{\alpha})} + 2\alpha^2\delta^2$$
$$+ \sum_{\kappa=0}^{k} 2\alpha^2 c_2 \exp(\sqrt{\alpha})\sigma_{\max}^2(X_\gamma)\left(\left(1 - \frac{\alpha\mu}{2}\right)^k (\mathsf{L}_\gamma(W_1^0, W_2^0) - \mathsf{L}_\gamma^*) + \frac{3\delta^2}{2\mu}\right)$$
$$\le \left(2\alpha\delta\sqrt{c_2^1 \exp(\sqrt{\alpha})} + 2\alpha^2\delta^2 + \frac{3\alpha^2\delta^2 c_2 \exp(\sqrt{\alpha})\sigma_{\max}^2(X_\gamma)}{\mu}\right) K$$
$$+ \frac{4\alpha c_2 \exp(\sqrt{\alpha})\sigma_{\max}^2(X_\gamma)}{\mu} (\mathsf{L}_\gamma(W_1^0, W_2^0) - \mathsf{L}_\gamma^*)$$
$$\le \frac{5\alpha c_2 \exp(\sqrt{\alpha})\sigma_{\max}^2(X_\gamma)}{(c_1 + c_2(2 - \exp(\sqrt{\alpha}))) \sigma_*^2(X_\gamma)} (\mathsf{L}_\gamma(W_1^0, W_2^0) - \mathsf{L}_\gamma^*)$$

where the last inequality holds by (57) and (60).

**Step 4:** $\mathrm{P}_4(k)$ **holds.** According to (Xu et al., 2023, Appendix C), the initial weights can be bounded by

$$c_1^1 \le \sigma_{\min}(W_1^0) \le \sigma_{\max}(W_1^0) \le c_2^1$$
$$c_1^2 \le \sigma_{\min}(W_2^0) \le \sigma_{\max}(W_2^0) \le c_2^2$$

and the singular values of weight matrix can be bounded by the imbalance matrix

$$b_l^1 = c_1^1 - 2\|D_{k+1} - D_0\| \le \sigma_{\min}(W_1^{k+1}) \le \sigma_{\max}(W_1^{k+1}) \le c_2^1 + \|D_{k+1} - D_0\| = b_u^1$$
$$b_l^2 = c_1^2 - 2\|D_{k+1} - D_0\| \le \sigma_{\min}(W_2^{k+1}) \le \sigma_{\max}(W_2^{k+1}) \le c_2^2 + \|D_{k+1} - D_0\| = b_u^2.$$

Moreover, we have

$$b_u^i = c_2^i + \|D_{k+1} - D_0\|$$
$$\le c_2^i + \frac{5\alpha c_2 \exp(\sqrt{\alpha})\sigma_{\max}^2(X_\gamma)}{(c_1 + 2c_2(1 - \exp(\sqrt{\alpha}))) \sigma_*^2(X_\gamma)} (\mathsf{L}_\gamma(W_1^0, W_2^0) - \mathsf{L}_\gamma^*)$$
$$\le c_2^i + (1 - \exp(-\sqrt{\alpha})) (c_1 + c_2(2 - \exp(\sqrt{\alpha}))) \sigma_*^2(X_\gamma) \times \frac{c_2 \exp(\sqrt{\alpha})}{(c_1 + c_2(2 - \exp(\sqrt{\alpha}))) \sigma_*^2(X_\gamma)}$$
$$\le \exp(\sqrt{\alpha})c_2^i$$

where the first inequality holds due to $\mathbb{P}_3(k)$, and the second inequality holds because of the condition (56). Since $b_l^i + 2b_u^i = c_1^i + 2c_2^1$ and $b_u^i \leq \exp(\sqrt{\alpha})c_2^1$, $\mathrm{P}_4(k)$ holds because

$$c_1^i + 2c_2^i(1 - \exp(\sqrt{\alpha})) \leq b_l^1 \leq \sigma_{\min}(W_i^{k+1}) \leq \sigma_{\max}(W_i^{k+1}) \leq b_u^i \leq \exp(\sqrt{\alpha})c_2^i.$$

Therefore, the iterates of PBGD satisfy

$$\mathsf{L}_\gamma(W_1^k, W_2^k) - \mathsf{L}_\gamma^* \leq \left(1 - \frac{\alpha\mu}{2}\right)^k (\mathsf{L}_\gamma(W_1^0, W_2^0) - \mathsf{L}_\gamma^*) + \mathcal{O}\left(\frac{\epsilon^{0.5}}{\mu}\right).$$

Together with Theorem 1 and Theorem 4, we arrive at the conclusion.

## G   PROOF FOR DATA HYPER-CLEANING

We provide the omitted proof of lemmas and theorems for data hyper-cleaning.

### G.1   INDEPENDENCE OF THE LOWER-LEVEL SOLUTION SET $\mathcal{S}(u)$

**Lemma 23.** *If $X_{\mathrm{trn}}X_{\mathrm{trn}}^\dagger$ is a diagonal matrix, then for any $u$, $\mathcal{S}(u)$ that is independent of $u$ and thus*

$$\ell_{\mathrm{trn}}^*(u) = \frac{1}{2}\left\|\sqrt{\psi_N(u)}\left(I - X_{\mathrm{trn}}X_{\mathrm{trn}}^\dagger\right)Y_{\mathrm{trn}}\right\|^2 = \frac{1}{2}\sum_{i=1}^N \psi(u_i)\|y_i\|^2\mathbb{1}\left([X_{\mathrm{trn}}X_{\mathrm{trn}}^\dagger]_{ii} \neq 1\right). \quad (68)$$

**Proof:**  According to (Barata & Hussein, 2012, Theorem 6.1), one of the lower-level solutions $W^*$ of (10) should satisfy

$$\begin{aligned}
W^* &= (\sqrt{\psi_N(u)}X_{\mathrm{trn}})^\dagger \sqrt{\psi_N(u)}Y_{\mathrm{trn}} \\
&\stackrel{(a)}{=} X_{\mathrm{trn}}^\dagger \sqrt{\psi_N(u)}^{-1}\sqrt{\psi_N(u)}Y_{\mathrm{trn}} \\
&= (X_{\mathrm{trn}})^\dagger Y_{\mathrm{trn}}
\end{aligned} \quad (69)$$

where (a) holds because $\sqrt{\psi_N(u)}$ is invertiable for any $u \in \mathcal{U}$ and Lemma 10. Therefore,

$$\mathsf{L}_{\mathrm{trn}}^*(u) = \frac{1}{2}\left\|\sqrt{\psi_N(u)}\left(I - X_{\mathrm{trn}}X_{\mathrm{trn}}^\dagger\right)Y_{\mathrm{trn}}\right\|^2 = \frac{1}{2}\sum_{i=1}^N \psi(u_i)\|y_i\|^2\mathbb{1}([X_{\mathrm{trn}}X_{\mathrm{trn}}^\dagger]_{ii} \neq 1)$$

where $[X_{\mathrm{trn}}X_{\mathrm{trn}}^\dagger]_{ii}$ denotes the element of matrix $[X_{\mathrm{trn}}X_{\mathrm{trn}}^\dagger]$ at the position $(i,i)$.

**Remark 3.** *Lemma 23 suggests that the $(\epsilon, 0)$ solution to the bilevel problem is meaningless since the static lower-level solution set makes the bilevel objective in (10) independent of $u$. This is also aligned with the observation that the model achieved by data reweighting is independent of data weight when the training dataset is linear independent (Zhai et al., 2023, Theorem 1). However, the $\epsilon$-lower-level solution relies on $u$ (the rationale is detailed below). This could also explain the reason why solving bilevel problem by nested approach to the $(\epsilon, 0)$ stationary point has accuracy drop over the $(\epsilon, \epsilon)$ stationary point attained by penalized method in the data hyper-cleaning task (Xiao et al., 2023b).*

**Explanation of Remark 3.** The gradient of lower-level objective can be computed as

$$\begin{aligned}
\nabla_W \ell_{\mathrm{trn}}(u, W) &= -\frac{1}{2}(\sqrt{\psi_N(u)}X_{\mathrm{trn}})^\top \sqrt{\psi_N(u)}(Y_{\mathrm{trn}} - X_{\mathrm{trn}}W) \\
&= -\frac{1}{2}X_{\mathrm{trn}}^\top \psi_N(u)(Y_{\mathrm{trn}} - X_{\mathrm{trn}}W) \\
&= \frac{1}{2}\sum_{i=1}^N \psi(u_i)x_i(x_i^\top W - y_i^\top).
\end{aligned} \quad (70)$$

When the weight of $i$-th sample $\psi(u_i)$ is close to 0, $i$-th sample minimally influences the lower-level optimization. Ideally, the optimal response weight which fits the dataset excluding the $i$-th sample, will also perform approximately well on the weighted objective.

To see it, let $u^i = (\bar{u}, \cdots, \bar{u}, -\bar{u}, \bar{u}, \cdots, \bar{u})$ with $-\bar{u}$ at the $i$-th position and $\bar{u}$ elsewhere. We can then choose the minimal norm solution $W(u^i) \in \arg\min_W \frac{1}{2} \sum_{j \neq i} \|x_j^\top W - y_j^\top\|^2$ such that $\sum_{j \neq i} x_j(x_j^\top W - y_j^\top) = 0$. According to Lemma 23, $W(u^i) \in \arg\min_W \frac{1}{2} \sum_{j \neq i} \|x_j^\top W - y_j^\top\|$ is also solution to $\frac{1}{2} \sum_{j \neq i} \psi(u_j) \|x_j^\top W - y_j^\top\|^2$ so that $\sum_{j \neq i} \psi(u_j) x_j(x_j^\top W - y_j^\top) = 0$. By (70),

$$\nabla_W \ell_{\mathrm{trn}}(u^i, W(u^i)) = \frac{\psi(-\bar{u})}{2} x_i(x_i^\top W(u^i) - y_i^\top)$$

holds for any $i$. Taking the norm yields

$$\|\nabla_W \ell_{\mathrm{trn}}(u^i, W(u^i))\| = \frac{\psi(-\bar{u})}{2} \|x_i(x_i^\top W(u^i) - y_i^\top)\|$$

which is very small because $\psi(-\bar{u})$ is close to 0 and $\|x_i(x_i^\top W(u^i) - y_i^\top)\|$ is bounded. According to the PL property of the training loss, we know the function value gap $\ell_{\mathrm{trn}}(u^i, W(u^i)) - \ell_{\mathrm{trn}}^*(u^i)$ is also small enough, suggesting that $W(u^i)$ is an $\epsilon$-solution of lower-level for some $\epsilon$.

**Lemma 24.** *If $[X_{\mathrm{trn}}; X_{\mathrm{val}}][X_{\mathrm{trn}}; X_{\mathrm{val}}]^\dagger$ is a diagonal matrix, then for any $u$ and $\gamma$, $\arg\min_W \ell_\gamma(u, W)$ is independent of $u$.*

**Proof:** The proof follows that of Lemma 23 by replacing $X_{\mathrm{trn}}$ with $[X_{\mathrm{trn}}; X_{\mathrm{val}}]$.

### G.2 PROOF OF LEMMA 2: LOCAL BLOCKWISE PL AND SMOOTHNESS

We first restate Lemma 2 in a formal way as follows.

**Lemma 25** (Local blockwise PL and smoothness of $\ell_\gamma(u, W)$)**.** *If $X_{\mathrm{trn}} X_{\mathrm{trn}}^\dagger$ is a diagonal matrix, then for any $u \in \mathcal{U}$ and any $W$, the penalized objective $\ell_\gamma(u, W)$ is $L_w^\gamma$-smooth and $\mu_w^\gamma$-PL over $W$, where the constants are defined as*

$$\mu_w^\gamma := \sigma_*^2 \left( X_{\mathrm{val}}^\top X_{\mathrm{val}} + \gamma(1 - \psi(\bar{u})) X_{\mathrm{trn}}^\top X_{\mathrm{trn}} \right), \quad L_w^\gamma := \sigma_{\max}^2 \left( X_{\mathrm{val}}^\top X_{\mathrm{val}} + \gamma \psi(\bar{u}) X_{\mathrm{trn}}^\top X_{\mathrm{trn}} \right).$$

*Similarly, the lower-level objective $\ell_{\mathrm{trn}}(u, W)$ is $L_w$-smooth and $\mu_w$-PL over $W$, with the constants*

$$\mu_w := \sigma_*^2 \left( (1 - \psi(\bar{u})) X_{\mathrm{trn}}^\top X_{\mathrm{trn}} \right), \quad L_w := \sigma_{\max}^2 \left( (1 - \psi(\bar{u})) X_{\mathrm{trn}}^\top X_{\mathrm{trn}} \right).$$

*Moreover, $\ell_\gamma(u, W)$ is $\gamma \ell_{\mathrm{trn}}(W)$ smooth and $\frac{\gamma c(W) \psi(\bar{u})(1 - \psi(\bar{u}))^2}{4}$-PL over $u \in \mathcal{U}$, where we define*

$$c(W) = \min_i \left\{ \|y_i^\top - x_i^\top W\|^2 - \|y_i\|^2 \mathbb{1}([X_{\mathrm{trn}} X_{\mathrm{trn}}^\dagger]_i \neq 1) \right\}_{>0}$$

*as the lower bound of the **positive mismatch** in the training loss. If there is no **positive mismatch**, we can set $c(W)$ to any positive number.*

**Proof:** We first show that Sigmoid function $\psi(u_i)$ is $\psi(\bar{u})(1 - \psi(\bar{u}))^2$ PL and 1 smooth over $u_i$, where $u_i$ is the $i$-th element in $u$. According to the definition, the gradient of the Sigmoid function can be computed as $\nabla \psi(u_i) = \psi(u_i)(1 - \psi(u_i))$, and its Hessian has the form of

$$
\begin{aligned}
\nabla^2 \psi(u_i) &= \psi(u_i)(1 - \psi(u_i))^2 + \psi(u_i)(-\psi(u_i)(1 - \psi(u_i))) \\
&= \psi(u_i)(1 - \psi(u_i))^2 - \psi(u_i)^2(1 - \psi(u_i)) \\
&= \psi(u_i)(1 - \psi(u_i))(1 - 2\psi(u_i)) \leq 1
\end{aligned}
\tag{71}
$$

which validates the 1 smoothness. On the other hand,

$$
\begin{aligned}
\|\nabla \psi(u_i)\|^2 &= \psi(u_i)^2(1 - \psi(u_i))^2 \\
&\geq \{ \min_{-\bar{u} \leq u_i \leq \bar{u}} \psi(u_i)(1 - \psi(u_i))^2 \} \psi(u_i) \\
&\geq \min\{\psi(-\bar{u})(1 - \psi(-\bar{u}))^2, \psi(\bar{u})(1 - \psi(\bar{u}))^2\} \psi(u_i) \\
&\geq \min\{\psi(\bar{u})^2(1 - \psi(\bar{u})), \psi(\bar{u})(1 - \psi(\bar{u}))^2\} \psi(u_i) \\
&\geq \psi(\bar{u})(1 - \psi(\bar{u}))^2 \psi(u_i) \\
&\geq \psi(\bar{u})(1 - \psi(\bar{u}))^2 (\psi(u_i) - \min_{u_i} \psi(u_i))
\end{aligned}
\tag{72}
$$

where the third inequality holds since $\psi(-\bar{u}) = 1 - \psi(\bar{u})$, the fourth inequality comes from $\bar{u} > 0$ so that $\psi(\bar{u}) > 1 - \psi(\bar{u})$, and the last inequality follows from $\min_{u_i} \psi(u_i) = 0$. This proves that $\psi(u_i)$ is $\psi(\bar{u})(1 - \psi(\bar{u}))^2$ PL when $u_i \in [-\bar{u}, \bar{u}]$.

Recall that the penalized objective can be written as

$$\ell_\gamma(u, W) = \ell_{\mathrm{val}}(W) + \frac{\gamma}{2} \sum_{i=1}^{N} \psi(u_i) \left[ \|y_i^\top - x_i^\top W\|^2 - \|y_i\|^2 \mathbb{1}([X_{\mathrm{trn}} X_{\mathrm{trn}}^\dagger]_i \neq 1) \right].$$

Therefore, the gradient of $\mathsf{L}_\gamma(u, W)$ satisfies

$$\|\nabla_u \ell_\gamma(u, W)\|^2 = \sum_{i=1}^{N} \|\nabla_{u_i} \ell_\gamma(u, W)\|^2$$

$$= \sum_{i=1}^{N} \left\| \frac{\gamma}{2} \nabla \psi(u_i) \left[ \|y_i^\top - x_i^\top W\|^2 - \|y_i\|^2 \mathbb{1}([X_{\mathrm{trn}} X_{\mathrm{trn}}^\dagger]_i \neq 1) \right] \right\|^2$$

$$= \frac{\gamma^2}{4} \sum_{i=1}^{N} \nabla \psi(u_i)^2 \left[ \|y_i^\top - x_i^\top W\|^2 - \|y_i\|^2 \mathbb{1}([X_{\mathrm{trn}} X_{\mathrm{trn}}^\dagger]_i \neq 1) \right]^2$$

$$\overset{(72)}{\geq} \frac{\gamma^2}{4} \sum_{i=1}^{N} \psi(\bar{u})(1 - \psi(\bar{u}))^2 \psi(u_i) \left[ \|y_i^\top - x_i^\top W\|^2 - \|y_i\|^2 \mathbb{1}([X_{\mathrm{trn}} X_{\mathrm{trn}}^\dagger]_i \neq 1) \right]^2$$

$$\geq \frac{\gamma^2 c(W) \psi(\bar{u})(1 - \psi(\bar{u}))^2}{4} \sum_{i=1}^{N} \psi(u_i) \left[ \|y_i^\top - x_i^\top W\|^2 - \|y_i\|^2 \mathbb{1}([X_{\mathrm{trn}} X_{\mathrm{trn}}^\dagger]_i \neq 1) \right]$$

$$= \frac{\gamma c(W) \psi(\bar{u})(1 - \psi(\bar{u}))^2}{2} (\ell_\gamma(u, W) - \mathsf{L}_{\mathrm{val}}(W))$$

$$\geq \frac{\gamma c(W) \psi(\bar{u})(1 - \psi(\bar{u}))^2}{2} (\ell_\gamma(u, W) - \min_u \ell_\gamma(u, W)) \tag{73}$$

where the last inequality holds from $\min_u \mathsf{L}_\gamma(u, W) = \mathsf{L}_{\mathrm{val}}(W)$. This shows $\ell_\gamma(u, W)$ is $\frac{\gamma c(W) \psi(\bar{u})(1 - \psi(\bar{u}))^2}{4}$ PL over $u$. The Hessian of $\ell_\gamma(u, W)$ can be calculated by

$$\|\nabla_u^2 \ell_\gamma(u, W)\| = \sum_{i=1}^{N} \left\| \frac{\gamma}{2} \nabla^2 \psi(u_i) \left[ \|y_i^\top - x_i^\top W\|^2 - \|y_i\|^2 \mathbb{1}([X_{\mathrm{trn}} X_{\mathrm{trn}}^\dagger]_i \neq 1) \right] \right\|$$

$$\leq \frac{\gamma}{2} \sum_{i=1}^{N} \|\nabla^2 \psi(u_i)\| \|y_i^\top - x_i^\top W\|^2$$

$$\leq \frac{\gamma}{2} \sum_{i=1}^{N} \|y_i^\top - x_i^\top W\|^2 = \gamma \ell_{\mathrm{trn}}(W) \tag{74}$$

which indicates $\ell_\gamma(u, W)$ is $\gamma \ell_{\mathrm{trn}}(W)$ smooth over $u$.

For the property over $W$, we can use the matrix form of the objective and define

$$\widetilde{\ell}_u(W) := \frac{1}{2} \|Y_{\mathrm{val}} - X_{\mathrm{val}} W\|^2 + \frac{\gamma}{2} \left\| \sqrt{\psi_N(u)} \left( Y_{\mathrm{trn}} - X_{\mathrm{trn}} W \right) \right\|^2 = \frac{1}{2} \|Y_\gamma(u) - X_\gamma(u) W\|^2$$

where $X_\gamma(u) = [X_{\mathrm{val}}; \sqrt{\gamma \psi_N(u)} X_{\mathrm{trn}}]$ and $Y_\gamma(u) = [Y_{\mathrm{val}}; \sqrt{\gamma \psi_N(u)} Y_{\mathrm{trn}}]$.

Then according to (Karimi et al., 2016, Appendix B), $\widetilde{\ell}_u(W)$ is $\sigma_{\max}^2(X_\gamma(u))$ smooth and $\sigma_*^2(X_\gamma(u))$ PL. Since

$$\ell_\gamma(u, W) = \widetilde{\ell}_u(W) - h(u) \quad \text{with} \quad h(u) = -\frac{\gamma}{2} \sum_{i=1}^{N} \psi(u_i) \|y_i\|^2 \mathbb{1}([X_{\mathrm{trn}} X_{\mathrm{trn}}^\dagger]_i \neq 1)$$

where $h(u)$ is independent of $W$, we have $\ell_\gamma(u, W)$ is $\sigma_{\max}^2(X_\gamma(u))$ smooth over $W$ and

$$\|\nabla_W \ell_\gamma(u, W)\|^2 = \|\nabla \widetilde{\ell}_u(W)\|^2 \geq 2\sigma_*^2(X_\gamma(u)) \left( \widetilde{\ell}_u(W) - \min_W \widetilde{\ell}_u(W) \right)$$

$$\stackrel{(a)}{=} 2\sigma_*^2(X_\gamma(u)) \left( \ell_\gamma(u, W) - \min_W \ell_\gamma(u, W) \right) \qquad (75)$$

indicating that $\ell_\gamma(u, W)$ is $\sigma_*^2(X_\gamma(u))$ PL over $W$. On the other hand, we have

$$X_\gamma(u)^\top X_\gamma(u) = [X_{\text{val}}^\top \ X_{\text{trn}}^\top \sqrt{\gamma \psi_N(u)}] \begin{bmatrix} X_{\text{val}} \\ \sqrt{\gamma \psi_N(u)} X_{\text{trn}} \end{bmatrix}$$
$$= X_{\text{val}}^\top X_{\text{val}} + \gamma X_{\text{trn}}^\top \psi_N(u) X_{\text{trn}}.$$

As $\sigma(\cdot)$ is strictly increasing and $\psi(-u) = 1 - \psi(u)$, we have $(1 - \psi(\bar{u}))I \preccurlyeq \psi_N(u) \preccurlyeq \psi(\bar{u})I$ and

$$X_{\text{val}}^\top X_{\text{val}} + \gamma(1 - \psi(\bar{u}))X_{\text{trn}}^\top X_{\text{trn}} \preccurlyeq X_\gamma(u)^\top X_\gamma(u) \preccurlyeq X_{\text{val}}^\top X_{\text{val}} + \gamma\psi(\bar{u})X_{\text{trn}}^\top X_{\text{trn}}.$$

Moreover, the definition of $X_\gamma(u)$ suggests that it is of constant rank when $u \in \mathcal{U}$. In this way, the singular value of the data matrix will be bounded by

$$\sigma_*^2(X_\gamma(u)) \geq \sigma_* \left( X_{\text{val}}^\top X_{\text{val}} + \gamma(1 - \psi(\bar{u}))X_{\text{trn}}^\top X_{\text{trn}} \right) =: \mu_w^\gamma \qquad (76)$$
$$\sigma_{\max}^2(X_\gamma(u)) \leq \sigma_{\max} \left( X_{\text{val}}^\top X_{\text{val}} + \gamma\psi(\bar{u})X_{\text{trn}}^\top X_{\text{trn}} \right) =: L_w^\gamma \qquad (77)$$

which means $\ell_\gamma(u, W)$ is uniformly smooth over $W$ with $L_w$ and is uniformly PL over $W$ with $\mu_w$. Similarly, the lower-level objective $\ell_{\text{trn}}(u, W)$ is uniformly smooth over $W$ with $L_w$ and PL over $W$ with $\mu_w$ which are defined as

$$\mu_w := (1 - \psi(\bar{u}))\sigma_* \left( X_{\text{trn}}^\top X_{\text{trn}} \right), \quad L_w := \sigma_{\max} \left( X_{\text{trn}}^\top X_{\text{trn}} \right).$$

### G.3 LIPSCHITZ CONTINUITY OF SOLUTION TO THE PENALIZED PROBLEM

**Lemma 26.** *There exists $W_\gamma^*(u) \in \arg\min \ell_\gamma(u, W)$ such that $W_\gamma^*(u)$ is $L_{wu}^*$ Lipschitz continuous over $u$ where $L_{wu}^* = \mathcal{O}(1)$. Moreover, $\|W_\gamma^*(u)\| \leq L_w^* = \mathcal{O}(1)$.*

**Proof:** This result can be deduced from a general result under the PL condition and the smoothness of $\ell_\gamma(u, \cdot)$, as demonstrated by (Nouiehed et al., 2019, Lemma A.3). Given that $\ell_\gamma(u, \cdot)$ is a squared loss composite with a linear mapping —- a specific case of a PL function – we aim to separately derive the Lipschitz continuity of the solution set for clarity and simplicity.

Let $W_\gamma^*(u) = X_\gamma(u)^\dagger Y_\gamma(u)$ be the minimal norm solution of $\min_W \ell_\gamma(u, W)$, where $X_\gamma(u) = [X_{\text{val}}; \sqrt{\gamma \psi_N(u)} X_{\text{trn}}]$ and $Y_\gamma(u) = [Y_{\text{val}}; \sqrt{\gamma \psi_N(u)} Y_{\text{trn}}]$. According to (76) and (77), we have

$$\|X_\gamma(u)^\dagger\|_2 \leq \frac{1}{\sigma_*(X_\gamma(u))} \leq \frac{1}{\sqrt{\mu_w^\gamma}} = \mathcal{O}(\gamma^{-\frac{1}{2}})$$

$$\|X_\gamma(u^1)^\dagger - X_\gamma(u^2)^\dagger\| \stackrel{(a)}{\leq} \sqrt{2} \max \left\{ \|X_\gamma(u^1)\|_2^2, \|X_\gamma(u^2)\|_2^2 \right\} \|X_\gamma(u^1) - X_\gamma(u^2)\|$$
$$\leq \frac{\sqrt{2}}{\sigma_*(X_\gamma(\bar{u}))^2} \|X_\gamma(u^1) - X_\gamma(u^2)\|$$
$$\stackrel{(b)}{\leq} \frac{\sqrt{2}}{\mu_w^\gamma} \sqrt{\gamma} \|(\sqrt{\psi_N(u^1)} - \sqrt{\psi_N(u^2)})X_{\text{trn}}\|$$
$$\leq \frac{\sqrt{2}}{\mu_w^\gamma} \sqrt{\gamma} \|\sqrt{\psi_N(u^1)} - \sqrt{\psi_N(u^2)}\|_2 \|X_{\text{trn}}\|$$
$$\stackrel{(c)}{\leq} \frac{\sqrt{2}}{2\mu_w^\gamma \sqrt{1 - \psi(\bar{u})}} \sqrt{\gamma} \|X_{\text{trn}}\| \|\psi_N(u^1) - \psi_N(u^2)\|_2$$
$$\stackrel{(d)}{\leq} \frac{\sqrt{2}}{2\mu_w^\gamma \sqrt{1 - \psi(\bar{u})}} \sqrt{\gamma} \|X_{\text{trn}}\| \|u^1 - u^2\| = \mathcal{O}(\gamma^{-\frac{1}{2}}) \|u^1 - u^2\|$$

where (a) results from (Stewart, 1977, Theorem 3.3), (b) is derived from the definition of $X_\gamma(u)$, and (c) is because $\sqrt{a} - \sqrt{b} = \frac{a-b}{\sqrt{a}+\sqrt{b}}$ and the bound of $\sqrt{\psi(u_i^1)} + \sqrt{\psi(u_i^1)} \geq 2\sqrt{1 - \psi(\bar{u})}, \forall i$ and (d) comes from the 1-Lischitz continuity of the Sigmoid function $\psi(\cdot)$.

Similarly, we can derive the bound for $Y_\gamma(u)$ as

$$\|Y_\gamma(u)\|_2 = \sigma_{\max}(Y_\gamma(u)) \leq \sqrt{\sigma_{\max}(Y_{\mathrm{val}}^\top Y_{\mathrm{val}} + \gamma\psi(\bar{u})Y_{\mathrm{trn}}^\top Y_{\mathrm{trn}})} = \mathcal{O}(\gamma^{\frac{1}{2}})$$

$$\|Y_\gamma(u^1) - Y_\gamma(u^2)\| \overset{(a)}{=} \sqrt{\gamma}\|(\sqrt{\psi_N(u^1)} - \sqrt{\psi_N(u^2)})Y_{\mathrm{trn}}\|$$

$$\overset{(b)}{\leq} \frac{\sqrt{\gamma}\|Y_{\mathrm{trn}}\|}{2\sqrt{1-\psi(\bar{u})}}\|u^1 - u^2\| = \mathcal{O}(\gamma^{\frac{1}{2}})\|u^1 - u^2\|.$$

where (a) comes from the definition and (b) is derived from the Lipschitz continuity of Sigmoid function, $\|AB\| \leq \|A\|_2\|B\|$ and $\sqrt{a} - \sqrt{b} = \frac{a-b}{\sqrt{a}+\sqrt{b}}$. As a result, for any $u^1$ and $u^2$, it holds that

$$\|W_\gamma^*(u^1) - W_\gamma^*(u^2)\| \leq \|X_\gamma(u^1)^\dagger\|_2\|Y_\gamma(u^1) - Y_\gamma(u^2)\| + \|Y_\gamma(u^2)\|_2\|X_\gamma(u^1)^\dagger - X_\gamma(u^2)^\dagger\|$$

$$\leq \left(\frac{\sqrt{\gamma}\|Y_{\mathrm{trn}}\|}{2\sqrt{\mu_w^\gamma(1-\psi(\bar{u}))}} + \frac{\sqrt{2\gamma\sigma_{\max}(Y_{\mathrm{val}}^\top Y_{\mathrm{val}} + \gamma\psi(\bar{u})Y_{\mathrm{trn}}^\top Y_{\mathrm{trn}})}}{2\mu_w^\gamma\sqrt{1-\psi(\bar{u})}}\|X_{\mathrm{trn}}\|\right)\|u^1 - u^2\|.$$

Let us denote

$$L_{wu}^* := \frac{\sqrt{\gamma}\|Y_{\mathrm{trn}}\|}{2\sqrt{\mu_w^\gamma(1-\psi(\bar{u}))}} + \frac{\sqrt{2\gamma\sigma_{\max}(Y_{\mathrm{val}}^\top Y_{\mathrm{val}} + \gamma\psi(\bar{u})Y_{\mathrm{trn}}^\top Y_{\mathrm{trn}})}}{2\mu_w^\gamma\sqrt{1-\psi(\bar{u})}}\|X_{\mathrm{trn}}\| = \mathcal{O}(1) \qquad (78)$$

then $W_\gamma^*(u)$ is $L_{wu}^*$ Lipschitz continuous on $u$.

Besides, the boundedness of $W_\gamma^*(u)$ can be earned by

$$\|W_\gamma^*(u)\| \leq \|X_\gamma(u)^\dagger\|_2\|Y_\gamma(u)\| \leq \max\{n, N+N'\}\|X_\gamma(u)^\dagger\|_2\|Y_\gamma(u)\|_2$$

$$\leq \frac{\max\{n, N+N'\}\sqrt{\sigma_{\max}(Y_{\mathrm{val}}^\top Y_{\mathrm{val}} + \gamma\psi(\bar{u})Y_{\mathrm{trn}}^\top Y_{\mathrm{trn}})}}{\sqrt{\mu_w^\gamma}} =: L_w^* = \mathcal{O}(1).$$

### G.4 GLOBAL SOLUTION RELATIONS IN DATA HYPER-CLEANING

Based on the smoothness and PL of $\ell_{\mathrm{trn}}(u, W)$ over $W$, we are expected to analyze the approximate behavior of the penalized problem (11) to the bilevel hyper-cleaning problem (10). Since upper-level problem is not Lipschitz continuous globally, the result in (Shen et al., 2023) can not be applied directly. In light of Remark 3, we only focus on whether $\epsilon_2$ solution of the penalized problem can recover some approximate solution of the bilevel problem in the following lemma.

**Theorem 7.** *For any $\gamma$ and any $\epsilon_2$, suppose that there exists an $\epsilon_2$ solution $(u, W)$ to the $\gamma$-penalized problem* (11) *which has bounded norm $\|W\| \leq R$ and $R$ is independent of $u$ and $\gamma$. For such $\epsilon_2$ solution $(u, W)$, there exists $\gamma^* = \mathcal{O}(\epsilon_1^{-1})$ such that for any $\gamma > \gamma^*$, $(u, W)$ is also an $(\epsilon_2, \epsilon_\gamma)$ solution to the bilevel problem* (10) *for some $\epsilon_\gamma \leq \frac{\epsilon_1 + \epsilon_2}{\gamma - \gamma^*}$.*

Similar to the representation learning, to ensure the penalized problem (11) is an $\epsilon$-approximate solution to the bilevel data hyper-cleaning problem (10), i.e. $\epsilon_\gamma = \mathcal{O}(\epsilon)$, one need to choose $\epsilon_1 = \mathcal{O}(\sqrt{\epsilon})$ and set the penalty parameter $\gamma = \mathcal{O}(\epsilon^{-0.5})$. Therefore, if we can verify the iterates generates by PBGD achieves $\epsilon$-solution of the penalized problem and are bounded with radius independent of $\gamma$ and $u$, then PBGD can converge to some approximate solution of bilevel problem.

**Proof:** Given $\gamma$, we first prove the Lipschitz continuity of the upper-level objective (validation loss) at the lower-level solution set, i.e. for any $\epsilon_2$ solution $(u, W)$ of $\gamma$-penalized problem with $\|W\| \leq R$, let $W_u = \arg\min_W \ell_{\mathrm{trn}}(u, W)$, then we have $\|W_u\| = \|\mathrm{Proj}_{\mathcal{S}(u)}(W)\| \leq \|W\| \leq R$ and

$$\ell_{\mathrm{val}}(W) - \ell_{\mathrm{val}}(W_u) = \frac{1}{2}\|Y_{\mathrm{val}} - X_{\mathrm{val}}W\|^2 - \frac{1}{2}\|Y_{\mathrm{val}} - X_{\mathrm{val}}W_u\|^2$$

$$= \frac{1}{2}\langle 2Y_{\mathrm{val}} - X_{\mathrm{val}}(W + W_u), X_{\mathrm{val}}(W - W_u)\rangle$$

$$\geq -\left(\|Y_{\mathrm{val}}\| + \|X_{\mathrm{val}}\|\|W\|\right)\|X_{\mathrm{val}}\|\|W - W_u\|$$

$$\geq - \left( \|Y_{\text{val}}\| + \|X_{\text{val}}\|R \right) \|X_{\text{val}}\| \|W - W_u\|.$$

Then by choosing $\gamma^* = \frac{(\|Y_{\text{val}}\| + \|X_{\text{val}}\|R)^2 \|X_{\text{val}}\|^2}{2\mu_w \epsilon_1}$, it follows that

$$
\begin{aligned}
& \ell_{\text{val}}(W) + \gamma^*(\ell_{\text{trn}}(u, W) - \ell_{\text{trn}}^*(u)) - \ell_{\text{val}}(W_u) \\
& \geq - \left( \|Y_{\text{val}}\| + \|X_{\text{val}}\|R \right) \|X_{\text{val}}\| \|W - W_u\| + \gamma^*(\ell_{\text{trn}}(u, W) - \ell_{\text{trn}}^*(u)) \\
& \geq - \left( \|Y_{\text{val}}\| + \|X_{\text{val}}\|R \right) \|X_{\text{val}}\| \|W - W_u\| + \frac{\mu_w \gamma^*}{2} \|W - W_u\|^2 \\
& \geq \min_{z \in \mathbb{R}_+} - \left( \|Y_{\text{val}}\| + \|X_{\text{val}}\|R \right) \|X_{\text{val}}\| z + \frac{\mu_w \gamma^*}{2} z^2 \\
& = - \frac{\left( \|Y_{\text{val}}\| + \|X_{\text{val}}\|R \right)^2 \|X_{\text{val}}\|^2}{2\mu_w \gamma^*} = -\epsilon_1.
\end{aligned}
\tag{79}
$$

According to the definition of $\epsilon_2$ stationary point of $\gamma$ penalized problem, we have

$$
\begin{aligned}
\ell_{\text{val}}(W) + \gamma(\ell_{\text{trn}}(u, W) - \ell_{\text{trn}}^*(u)) & \leq \ell_{\text{val}}(W_u) + \gamma(\ell_{\text{trn}}(u, W_u) - \ell_{\text{trn}}^*(u)) + \epsilon_2 \\
& \leq \ell_{\text{val}}(W_u) + \epsilon_2 \\
& \overset{(79)}{\leq} \ell_{\text{val}}(W) + \gamma^*(\ell_{\text{trn}}(u, W) - \ell_{\text{trn}}^*(u)) + \epsilon_1 + \epsilon_2.
\end{aligned}
$$

Substracting $\ell_{\text{val}}(W)$ from both sides and rearraging yields

$$\epsilon_\gamma := \ell_{\text{trn}}(u, W) - \ell_{\text{trn}}^*(u) \leq \frac{\epsilon_1 + \epsilon_2}{\gamma - \gamma^*}$$

when $\gamma > \gamma^*$. Then for any $(u', W')$ satisfying $\ell_{\text{trn}}(u', W') - \ell_{\text{trn}}^*(u') \leq \epsilon_\gamma$, it holds that

$$\ell_{\text{val}}(W) + \gamma(\ell_{\text{trn}}(u, W) - \ell_{\text{trn}}^*(u)) \leq \ell_{\text{val}}(W') + \gamma(\ell_{\text{trn}}(u', W') - \ell_{\text{trn}}^*(u')) + \epsilon_2.$$

Rearraging terms yields

$$\ell_{\text{val}}(W) \leq \ell_{\text{val}}(W') + \gamma(\ell_{\text{trn}}(u', W') - \ell_{\text{trn}}^*(u') - \epsilon_\gamma) + \epsilon_2 \leq \ell_{\text{val}}(W') + \epsilon_2$$

which means $(u, W)$ is an $(\epsilon_2, \epsilon_\gamma)$ solution to bilevel problem.

### G.5 PARAMETERIZED DATA AND LABEL MATRIX FAMILY ARE ACUTE

**Lemma 27.** *For $u \in \mathcal{U}$, data matrix family $\{X_\gamma(u), u \in \mathcal{U}\}$ is acute, and $\{Y_\gamma(u), u \in \mathcal{U}\}$ is acute. Therefore, $\mathrm{Ran}(X_\gamma(u))$ remains the same for $u \in \mathcal{U}$ and so does $\mathrm{Ran}(Y_\gamma(u))$.*

**Proof:** For any $u$, we can write $X_\gamma(u)$ and $Y_\gamma(u)$ as

$$X_\gamma(u) = \begin{bmatrix} I & \\ & \sqrt{\gamma \sigma_N(u)} \end{bmatrix} \begin{bmatrix} X_{\text{val}} \\ X_{\text{trn}} \end{bmatrix}, \quad Y_\gamma(u) = \begin{bmatrix} I & \\ & \sqrt{\gamma \sigma_N(u)} \end{bmatrix} \begin{bmatrix} Y_{\text{val}} \\ Y_{\text{trn}} \end{bmatrix}.$$

According to Lemma 12, it is obvious that $\mathrm{rank}(X_\gamma(u)) = \mathrm{rank}([X_{\text{val}}; X_{\text{trn}}])$ and $\mathrm{rank}(Y_\gamma(u)) = \mathrm{rank}([Y_{\text{val}}; Y_{\text{trn}}])$, which verifies the constant rank condition in Lemma 11. To show the acute property, we need to further prove that for any $u^1$ and $u^2$, if we denote $A = X_\gamma(u^1)$ and $B = X_\gamma(u^2)$, then $\mathrm{rank}(A) = \mathrm{rank}(P_A B R_A)$ with $P_A = AA^\dagger$ and $R_A = A^\dagger A$.

To do so, we first notice that, there exists a diagonal matrix $\Lambda$ such that $B = \Lambda A$ where

$$\Lambda = \begin{bmatrix} I & \\ & \sqrt{\sigma_N(u^2)/\sigma_N(u^1)} \end{bmatrix}.$$

Then we can write

$$\mathrm{rank}(P_A B R_A) = \mathrm{rank}(AA^\dagger \Lambda AA^\dagger A) \overset{(a)}{=} \mathrm{rank}(AA^\dagger \Lambda A) \tag{80}$$

where (a) is because $AA^\dagger A = A$. Furthermore, by singular value decomposition, we can decompose $A = U \Sigma V^\top$ with $\Sigma = \begin{bmatrix} \Sigma_1 & 0 \\ 0 & 0 \end{bmatrix} \in \mathbb{R}^{(N+N') \times m}$, and orthogonal matrix $U = [U_1 \ U_2] \in$

$\mathbb{R}^{(N+N')\times(N+N')}$ and $V = [V_1\ V_2] \in \mathbb{R}^{m\times m}$. Also, by denoting $\text{rank}(A) = r$, we know that $U_1 \in \mathbb{R}^{(N+N')\times r}, V_1 \in \mathbb{R}^{m\times r}$ and $\Sigma_1 \in \mathbb{R}^{r\times r}$ are full rank. Thus, $A$ can be decomposed by

$$A = [U_1\ U_2] \begin{bmatrix} \Sigma_1 & 0 \\ 0 & 0 \end{bmatrix} \begin{bmatrix} V_1^\top \\ V_2^\top \end{bmatrix} = [U_1\Sigma_1\ \ 0] \begin{bmatrix} V_1^\top \\ V_2^\top \end{bmatrix} = U_1\Sigma_1 V_1^\top.$$

Besides, based on linear algebra knowledge, $A^\dagger = V_1\Sigma_1^{-1}U_1^\top$. In this way, we can further write

$$\begin{aligned}
\text{rank}(P_A B R_A) &\overset{(80)}{=} \text{rank}(AA^\dagger \Lambda A) = \text{rank}(U_1 U_1^\top \Lambda U_1 \Sigma_1 V_1^\top) \\
&\overset{(a)}{=} \text{rank}(U_1^\top \Lambda U_1) \overset{(b)}{=} \text{rank}((\sqrt{\Lambda}U_1)^\top \sqrt{\Lambda}U_1) \\
&\overset{(c)}{=} \text{rank}(\sqrt{\Lambda}U_1) \overset{(d)}{=} \text{rank}(U_1) = r = \text{rank}(A)
\end{aligned} \tag{81}$$

where (a) is derived from Lemma 12 with full column rank $U_1$ and full row rank $V_1^\top, \Sigma_1$, (b) is because $\Lambda$ is a nonsingular diagonal matrix, (c) and (d) come from Lemma 12. Combining (80) and (81) with Lemma 11, we know $A = X_\gamma(u^1)$ and $B = X_\gamma(u^2)$ are acute. Due to the arbitrary choices of $u^1, u^2 \in \mathcal{U}$, the family $\{X_\gamma(u), u \in \mathcal{U}\}$ is acute, which also holds for $\{Y_\gamma(u), u \in \mathcal{U}\}$. By the acute matrices in Proposition 1, the range space remains the same, which completes the proof.

### G.6 Monotonicity of loss functions

**Lemma 28.** *Given $u^1, u^2 \in \mathcal{U}$ and assuming $u^1 \leq u^2$ in the sense that $u_i^1 \leq u_i^2$ for any $i$, then*

$$\psi(u_i^1)\|y_i^\top - x_i^\top W_\gamma^*(u^1)\|^2 \leq \psi(u_i^2)\|y_i^\top - x_i^\top W_\gamma^*(u^2)\|^2$$
$$\|y_i'^\top - x_i'^\top W_\gamma^*(u^1)\|^2 \leq \|y_i'^\top - x_i'^\top W_\gamma^*(u^2)\|^2.$$

*where $\{x_i, y_i\}$ are training data samples and $\{x_i', y_i'\}$ are validation data samples. Moreover,*

$$\|y_i\|^2 \mathbb{1}([X_{\text{trn}}X_{\text{trn}}^\dagger]_i \neq 1) \leq \|y_i^\top - x_i^\top W_\gamma^*(u)\|^2.$$

**Proof:** For a matrix $A$, we denote $A_i$ as the $i$-th row of $A$. For any $u \in \mathcal{U}$, we have

$$\begin{aligned}
\|y_i^\top - x_i^\top W_\gamma^*(u)\|^2 &= \|y_i^\top - x_i^\top W_\gamma^*(u)\|^2 \\
&= \|y_i^\top - x_i^\top X_\gamma(u)^\dagger Y_\gamma(u)\|^2 \\
&= \|(Y_{\text{trn}} - X_{\text{trn}}X_\gamma^\dagger(u)Y_\gamma(u))_i\|^2 \\
&= \gamma^{-1}\psi(u_i)^{-1}\|(Y_\gamma(u) - X_\gamma(u)X_\gamma^\dagger(u)Y_\gamma(u))_{(N'+i)}\|^2 \\
&= \gamma^{-1}\psi(u_i)^{-1}\|((I - X_\gamma(u)X_\gamma^\dagger(u))Y_\gamma(u))_{(N'+i)}\|^2 \\
&= \gamma^{-1}\psi(u_i)^{-1}\|\text{Proj}_{\text{Ran}(X_\gamma(u))^\perp}(Y_\gamma(u))_{(N'+i)}\|^2
\end{aligned} \tag{82}$$

Then for $u^1 \leq u^2$, it holds that

$$\begin{aligned}
\|y_i^\top - x_i^\top W_\gamma^*(u^1)\|^2 &= \gamma^{-1}\psi(u_i^1)^{-1}\|\text{Proj}_{\text{Ran}(X_\gamma(u^1))^\perp}(Y_\gamma(u^1))_{(N'+i)}\|^2 \\
&\overset{(a)}{=} \gamma^{-1}\psi(u_i^1)^{-1}\|\text{Proj}_{\text{Ran}(X_\gamma(u^2))^\perp}(Y_\gamma(u^1))_{(N'+i)}\|^2 \\
&= \gamma^{-1}\psi(u_i^1)^{-1}\|((I - X_\gamma(u^2)X_\gamma^\dagger(u^2))Y_\gamma(u^1))_{(N'+i)}\|^2 \\
&\overset{(b)}{\leq} \gamma^{-1}\psi(u_i^1)^{-1}\|((I - X_\gamma(u^2)X_\gamma^\dagger(u^2))Y_\gamma(u^2))_{(N'+i)}\|^2 \\
&= \gamma^{-1}\psi(u_i^2)^{-1}\|((I - X_\gamma(u^2)X_\gamma^\dagger(u^2))Y_\gamma(u^2))_{(N'+i)}\|^2 \times \frac{\psi(u_i^2)}{\psi(u_i^1)} \\
&= \frac{\psi(u_i^2)}{\psi(u_i^1)}\|y_i^\top - x_i^\top W_\gamma^*(u^2)\|^2
\end{aligned}$$

where (a) is because $\text{Ran}(X_\gamma(u^1)) = \text{Ran}(X_\gamma(u^2))$ from Lemma 27, (b) is because each element in $Y_\gamma(u^1)$ is no greater than $Y_\gamma(u^2)$. For the validation loss, it can be proved similarly, while the only difference is the fact that

$$\|y_i'^\top - x_i'^\top W_\gamma^*(u)\|^2 = \|(Y_\gamma(u) - X_\gamma(u)X_\gamma^\dagger(u)Y_\gamma(u))_i\|^2$$

without magnitude of $\gamma^{-1}\psi(u_i)^{-1}$ because $Y_\gamma(u)$ and $X_\gamma(u)$ do not have such magnitude on validation data points.

Finally, as $\ell_{\text{trn}}(u, W) - \ell_{\text{trn}}^*(u) \geq 0$ holds for any $u \in \mathcal{U}$ and $W$, we have

$$\frac{\gamma}{2}\sum_{i=1}^{N}\psi(u_i)\left[\|y_i^\top - x_i^\top W\|^2 - \|y_i\|^2 \mathbb{1}([X_{\text{trn}}X_{\text{trn}}^\dagger]_i \neq 1)\right] \geq 0$$

holds for any $u \in \mathcal{U}$. Then letting $u$ be the vector which equals to $\bar{u}$ at $i$-th position and equals to $-\bar{u}$ elsewhere and taking $\bar{u} \to \infty$, we know

$$\|y_i^\top - x_i^\top W\|^2 - \|y_i\|^2 \mathbb{1}([X_{\text{trn}}X_{\text{trn}}^\dagger]_i \neq 1) \geq 0$$

holds for any $i$ and $W$. Taking $W = W_\gamma^*(u)$ yields the conclusion.

**Lemma 29.** *If $[X_{\text{trn}}; X_{\text{val}}][X_{\text{trn}}; X_{\text{val}}]^\dagger$ is diagonal matrix, $\ell_\gamma^*(u)$ is uniformly PL over $u \in \mathcal{U}$ with $\mu_u$ and is uniformly smooth over $u \in \mathcal{U}$ with $L_u$, and the constants are defined as*

$$\mu_u := \frac{\gamma\psi(\bar{u})(1-\psi(\bar{u}))^2 \min_i \|y_i\|^2}{4} = \mathcal{O}(\gamma), \quad L_u := \frac{\gamma}{2}\sum_{i=1}^{N}\|y_i\|^2 = \mathcal{O}(\gamma). \qquad (83)$$

*Moreover, the gradient of $\ell_\gamma^*(u)$ is also bounded by $L_u$, i.e. $\|\nabla_u \ell_\gamma^*(u)\| \leq L_u$.*

**Proof:** Since $\ell_\gamma(u, \cdot)$ is smooth and PL, according to (Nouiehed et al., 2019, Lemma A.5), we know $\ell_\gamma^*(u) = \min_W \ell_\gamma(u, W)$ is smooth with the gradient

$$\nabla\ell_\gamma^*(u) = \nabla_u \ell_\gamma(u, W), \quad \forall W \in \arg\min_W \ell_\gamma(u, W) = \arg\min_W \tilde{\ell}_\gamma(u, W).$$

We write the minimal-norm optimal solution of penalized problem as $W_\gamma^*(u) \in \arg\min_W \ell_\gamma(u, W)$. Plugging $W_\gamma^*(u)$ into (73), we know

$$\|\nabla\ell_\gamma^*(u)\|^2 = \|\nabla_u \ell_\gamma(u, W_\gamma^*(u))\|^2 \geq \frac{\gamma c(W_\gamma^*(u))\psi(\bar{u})(1-\psi(\bar{u}))^2}{2}(\ell_\gamma(u, W_\gamma^*(u)) - \min_u \ell_\gamma(u, W_\gamma^*(u)))$$

$$= \frac{\gamma c(W_\gamma^*(u))\psi(\bar{u})(1-\psi(\bar{u}))^2}{2}(\ell_\gamma^*(u) - \min_u \ell_\gamma^*(u))$$

which suggests that $\nabla\ell_\gamma^*(u)$ is $\frac{\gamma c(W_\gamma^*(u))\psi(\bar{u})(1-\psi(\bar{u}))^2}{4}$ PL over $u \in \mathcal{U}$. Besides, letting $u_0 = [\bar{u}, \cdots, \bar{u}]$, we have

$$c(W_\gamma^*(u)) = \min_i \left\{\|y_i^\top - x_i^\top W_\gamma^*(u)\|^2 - \|y_i\|^2 \mathbb{1}([X_{\text{trn}}X_{\text{trn}}^\dagger]_i \neq 1)\right\}_{>0}$$

$$= \min_i \left\{\gamma^{-1}\psi(u_i)^{-1}\|((I - X_\gamma(u)X_\gamma^\dagger(u))Y_\gamma(u))_{(N'+i)}\|^2 - \|y_i\|^2 \mathbb{1}([X_{\text{trn}}X_{\text{trn}}^\dagger]_i \neq 1)\right\}_{>0}$$

$$\overset{(a)}{=} \min_i \left\{\gamma^{-1}\psi(u_i)^{-1}\|((I - X_\gamma(u)X_\gamma^\dagger(u))Y_\gamma(u))_{(N'+i)}\|^2\right\}_{>0}$$

$$= \min_i \left\{\gamma^{-1}\psi(u_i)^{-1}\|\operatorname{Proj}_{\operatorname{Ran}(X_\gamma(u))^\perp}(Y_\gamma(u))_{(N'+i)}\|^2\right\}_{>0}$$

$$\overset{(b)}{\geq} \frac{\psi(-\bar{u})}{\psi(\bar{u})}\min_i \left\{\|y_i^\top - x_i^\top W_\gamma^*(-u_0)\|^2\right\}_{>0} = \mathcal{O}(1) \qquad (84)$$

where (a) comes from

$$\gamma^{-1}\psi(u_i)^{-1}\|((I - X_\gamma(u)X_\gamma^\dagger(u))Y_\gamma(u))_{(N'+i)}\|^2 - \|y_i\|^2 \mathbb{1}([X_{\text{trn}}X_{\text{trn}}^\dagger]_i \neq 1) > 0$$

if and only if $\gamma^{-1}\psi(u_i)^{-1}\|((I - X_\gamma(u)X_\gamma^\dagger(u))Y_\gamma(u))_{(N'+i)}\|^2 > 0$ and $[X_{\text{trn}}X_{\text{trn}}^\dagger]_i = 1$, (b) is due to Lemma 28. This means $\ell_\gamma^*(u)$ is uniformly PL over $u \in \mathcal{U}$ with constant

$$\mu_u := \frac{\gamma\psi(\bar{u})(1-\psi(\bar{u}))^2 \min_i \left\{\|y_i^\top - x_i^\top W_\gamma^*(-u_0)\|^2\right\}_{>0}}{4} = \mathcal{O}(\gamma). \qquad (85)$$

Moreover, $\nabla_u^2 \ell_\gamma(u', W)$ at $W = W_\gamma^*(u)$ can be upper bounded by

$$
\begin{aligned}
\|\nabla_u^2 \ell_\gamma(u', W_\gamma^*(u))\| &= \sum_{i=1}^{N} \left\| \frac{\gamma}{2} \nabla^2 \psi(u_i') \left[ \|y_i^\top - x_i^\top W_\gamma^*(u)\|^2 - \|y_i\|^2 \mathbb{1}([X_{\mathrm{trn}} X_{\mathrm{trn}}^\dagger]_i \neq 1) \right] \right\| \\
&\leq \frac{\gamma}{2} \sum_{i=1}^{N} \|\nabla^2 \psi(u_i')\| \left[ \|y_i^\top - x_i^\top W_\gamma^*(u)\|^2 - \|y_i\|^2 \mathbb{1}([X_{\mathrm{trn}} X_{\mathrm{trn}}^\dagger]_i \neq 1) \right] \\
&\leq \frac{\gamma}{2} \sum_{i=1}^{N} \left[ \|y_i^\top - x_i^\top W_\gamma^*(u)\|^2 - \|y_i\|^2 \mathbb{1}([X_{\mathrm{trn}} X_{\mathrm{trn}}^\dagger]_i \neq 1) \right] \\
&\leq \frac{\gamma}{2} \sum_{i=1}^{N} \left[ \|y_i^\top - x_i^\top W_\gamma^*(u)\|^2 \right] \\
&\overset{(a)}{\leq} \frac{\gamma}{2} \sum_{i=1}^{N} \|y_i^\top - x_i^\top W_\gamma^*(u_0)\|^2
\end{aligned}
\tag{86}
$$

where (a) comes from similar derivation as of (84). Similarly, for any $W = aW_\gamma(u^1) + (1-a)W_\gamma(u^2)$ where $a \in [0,1]$, $\nabla_{uW}^2 \ell_\gamma(u', W)$ can be bounded by

$$
\begin{aligned}
\|\nabla_{uW}^2 \ell_\gamma(u', W)\| &= \sum_{i=1}^{N} \left\| \gamma \nabla \psi(u_i') x_i (y_i^\top - x_i^\top W) \right\| \\
&\leq \gamma \sum_{i=1}^{N} \|x_i (y_i^\top - x_i^\top W)\| \\
&= \gamma \sum_{i=1}^{N} \|x_i (y_i^\top - x_i^\top (aW_\gamma(u^1) + (1-a)W_\gamma(u^2)))\| \\
&\leq a\gamma \sum_{i=1}^{N} \|x_i (y_i^\top - x_i^\top W_\gamma(u^1))\| + (1-a)\gamma \sum_{i=1}^{N} \|x_i (y_i^\top - x_i^\top W_\gamma(u^2))\| \\
&\leq \gamma \sum_{i=1}^{N} \|x_i\| \|y_i^\top - x_i^\top W_\gamma^*(u_0)\| = \mathcal{O}(\gamma).
\end{aligned}
\tag{87}
$$

Together (86) and (87) indicate that $\ell_\gamma^*(u)$ is smooth because for any $u^1, u^2 \in \mathbb{R}^N$, it holds that

$$
\begin{aligned}
&\|\nabla_u \ell_\gamma^*(u^1) - \nabla_u \ell_\gamma^*(u^2)\| \\
&= \|\nabla_u \ell_\gamma(u^1, W_\gamma^*(u^1)) - \nabla_u \ell_\gamma(u^2, W_\gamma^*(u^2))\| \\
&\leq \|\nabla_u \ell_\gamma(u^1, W_\gamma^*(u^1)) - \nabla_u \ell_\gamma(u^2, W_\gamma^*(u^1))\| + \|\nabla_u \ell_\gamma(u^2, W_\gamma^*(u^1)) - \nabla_u \ell_\gamma(u^2, W_\gamma^*(u^2))\| \\
&\overset{(a)}{\leq} \|\nabla_u^2 \ell_\gamma(u', W_\gamma^*(u^1))\| \|u^2 - u^1\| + \|\nabla_{uw}^2 \ell_\gamma(u^2, W')\| \|W_\gamma^*(u^1) - W_\gamma^*(u^2)\| \\
&\overset{(b)}{\leq} \frac{\gamma}{2} \sum_{i=1}^{N} \|y_i^\top - x_i^\top W_\gamma^*(u_0)\|^2 \|u^1 - u^2\| + \gamma \sum_{i=1}^{N} \|x_i\| \|y_i^\top - x_i^\top W_\gamma^*(u_0)\| L_{wu}^* \|u^1 - u^2\|
\end{aligned}
$$

where (a) comes from the mean value theorem with $u' = au^1 + (1-a)u^2$ and $W' = bW_\gamma^*(u^1) + (1-b)W_\gamma^*(u^2)$ for some $a, b \in [0,1]$, and (b) holds from (86) and (87). Therefore, we can define the smoothness constant of $\ell_\gamma^*$ as

$$
L_u := \frac{\gamma}{2} \sum_{i=1}^{N} \|y_i^\top - x_i^\top W_\gamma^*(u_0)\|^2 + \gamma \sum_{i=1}^{N} \|x_i\| \|y_i^\top - x_i^\top W_\gamma^*(u_0)\| L_{wu}^* = \mathcal{O}(\gamma).
\tag{88}
$$

Besides the bounded Hessian, the gradient of $\ell_\gamma(u, W_\gamma^*(u))$ is also bounded by $L_u$ because

$$
\|\nabla_u \ell_\gamma^*(u)\| = \|\nabla_u \ell_\gamma(u, W_\gamma^*(u))\|
$$

$$= \sum_{i=1}^{N} \left\| \frac{\gamma}{2} \nabla \psi(u_i) \left[ \|y_i^\top - x_i^\top W_\gamma^*(u)\|^2 - \|y_i\|^2 \mathbb{1}([X_{\text{trn}} X_{\text{trn}}^\dagger]_i \neq 1) \right] \right\|$$

$$\leq \frac{\gamma}{2} \sum_{i=1}^{N} \|\nabla \psi(u_i)\| \left[ \|y_i^\top - x_i^\top W_\gamma^*(u)\|^2 - \|y_i\|^2 \mathbb{1}([X_{\text{trn}} X_{\text{trn}}^\dagger]_i \neq 1) \right]$$

$$\leq \frac{\gamma}{2} \sum_{i=1}^{N} \left[ \|y_i^\top - x_i^\top W_\gamma^*(u)\|^2 \right] \overset{(a)}{\leq} \frac{\gamma}{2} \sum_{i=1}^{N} \|y_i^\top - x_i^\top W_\gamma^*(u_0)\|^2 \leq L_u \qquad (89)$$

where (a) can be obtained by (86).

### G.7 GRADIENT ESTIMATION ERROR

**Lemma 30.** *The gradient estimator has the bounded error*

$$\left\| \nabla \ell_\gamma^*(u) - \gamma \left( \nabla_u \ell_{\text{trn}}(u^k, W^{k+1}) - \nabla_u \ell_{\text{trn}}(u^k, Z^{k+1}) \right) \right\|$$

$$\leq \gamma \left( C(Z^0) \sqrt{\frac{2\ell_{\text{trn}}(Z^0)}{\mu_w}} + C(W^0) \sqrt{\frac{2\ell_\gamma(W^0)}{\mu_w^\gamma}} \right) (1 - \beta \mu_w)^{T/2}. \qquad (90)$$

*where $\beta \mu_w = \min\{\beta_1 \mu_w, \beta_2 \mu_w^\gamma\}$.*

**Proof:** Let $Z' = \text{Proj}_{\mathcal{S}(u^k)}(Z^{k+1})$, we know $\|Z'\| \leq \|Z^{k+1}\|$ and thus,

$$\|\nabla \ell_{\text{trn}}^*(u^k) - \nabla_u \ell_{\text{trn}}(u^k, Z^{k+1})\| \overset{(a)}{=} \|\nabla_u \ell_{\text{trn}}(u^k, Z') - \nabla_u \ell_{\text{trn}}(u^k, Z^{k+1})\|$$

$$= \left\| \frac{1}{2} \sum_{i=1}^{N} \nabla \psi(u_i) (\|y_i^\top - x_i^\top Z'\|^2 - \|y_i^\top - x_i^\top Z^{k+1}\|^2) \right\|$$

$$\leq \frac{1}{2} \sum_{i=1}^{N} \|2y_i - x_i^\top (Z' + Z^{k+1})\| \|x_i^\top (Z' - Z^{k+1})\|$$

$$\leq \sum_{i=1}^{N} (\|y_i\| + \|x_i\| \|Z^{k+1}\|) \|x_i\| \|Z' - Z^{k+1}\|$$

$$= \sum_{i=1}^{N} (\|y_i\| + \|x_i\| \|Z^{k+1}\|) \|x_i\| d(Z^{k+1}, \mathcal{S}(u^k)) \qquad (91)$$

where (a) results from the Danskin type theorem (Nouiehed et al., 2019, Lemma A.5) that $\nabla \ell_{\text{trn}}^*(u^k) = \nabla_u \ell_{\text{trn}}(u^k, Z), \forall Z \in \mathcal{S}(u^k)$. Then according to Lemmas 6 and 7, at each iteration $k$, we have

$$\|Z^{k+1}\| \leq \|Z^{k+1} - Z^0\| + \|Z^0\| \leq \|Z^0\| + \sqrt{\frac{8\ell_{\text{trn}}(u^k, Z^0)}{\mu_w}} \overset{(a)}{\leq} \|Z^0\| + \sqrt{\frac{8\ell_{\text{trn}}(Z^0)}{\mu_w}} \qquad (92)$$

and

$$d(Z^{k+1}, \mathcal{S}(u^k))^2 = d(Z^{k,T}, \mathcal{S}(u^k))^2 \leq \frac{2}{\mu_w} \left( \ell_{\text{trn}}(u^k, Z^{k,T}) - \min_Z \ell_{\text{trn}}(u^k, Z) \right)$$

$$\leq \frac{2(1 - \beta \mu_w)^T}{\mu_w} \left( \ell_{\text{trn}}(u^k, Z^{k,0}) - \min_Z \ell_{\text{trn}}(u^k, Z) \right)$$

$$= \frac{2(1 - \beta \mu_w)^T}{\mu_w} \left( \ell_{\text{trn}}(u^k, Z^0) - \min_Z \ell_{\text{trn}}(u^k, Z) \right)$$

$$\overset{(b)}{\leq} \frac{2(1 - \beta \mu_w)^T \ell_{\text{trn}}(Z^0)}{\mu_w} \qquad (93)$$

where (a) and (b) hold because $\min_Z \ell_{\text{trn}}(u^k, Z) \geq 0$ and $\psi(u) \leq 1$. Plugging (92), (93) into (91) and defining $C(Z^0) = \sum_{i=1}^N (\|y_i\| + \|x_i\|(\|Z^0\| + \sqrt{\frac{8\ell_{\text{trn}}(Z^0)}{\mu_w}}))\|x_i\|$, we obtain that

$$\|\nabla \ell_{\text{trn}}^*(u^k) - \nabla \ell_{\text{trn}}(u^k, Z^{k+1})\| \leq C(Z^0)\sqrt{\frac{2\ell_{\text{trn}}(Z^0)}{\mu_w}}(1 - \beta_1 \mu_w)^{T/2}. \tag{94}$$

Similarly, if we define $\tilde{\ell}_\gamma(u, W) = \ell_{\text{val}}(W) + \gamma \ell_{\text{trn}}(u, W)$ and $\tilde{\ell}_\gamma^*(u) := \min_W \tilde{\ell}_\gamma(u, W)$, as $\tilde{\ell}_\gamma(u, \cdot)$ is also smooth and PL, the gradient estimator of $\tilde{\ell}_\gamma^*$ can be also bounded by

$$\|\nabla \tilde{\ell}_\gamma^*(u^k) - \nabla_u \tilde{\ell}_\gamma(u^k, W^{k+1})\| \leq \gamma C(W^0)\sqrt{\frac{2\ell_\gamma(W^0)}{\mu_w^\gamma}}(1 - \beta_2 \mu_w^\gamma)^{T/2}. \tag{95}$$

We then define $\ell_\gamma^*(u) := \min_W \ell_\gamma(u, W)$ and since $\ell_\gamma(u, W) = \tilde{\ell}_\gamma(u, W) - \gamma \ell_{\text{trn}}^*(u)$, we have

$$\ell_\gamma^*(u) = \min_W \ell_\gamma(u, W) = \min_W \tilde{\ell}_\gamma(u, W) - \gamma \ell_{\text{trn}}^*(u) = \tilde{\ell}_\gamma^*(u) - \gamma \ell_{\text{trn}}^*(u)$$

and thus $\ell_\gamma^*(u)$ is differentiable and $\nabla \ell_\gamma^*(u) = \nabla \tilde{\ell}_\gamma^*(u) - \gamma \nabla \ell_{\text{trn}}^*(u)$.

Therefore, the gradient estimator of the penalized objective $\ell_\gamma^*$ can be bounded by

$$\begin{aligned}
&\left\|\nabla \ell_\gamma^*(u) - \gamma \left(\nabla_u \ell_{\text{trn}}(u^k, W^{k+1}) - \nabla_u \ell_{\text{trn}}(u^k, Z^{k+1})\right)\right\| \\
&\leq \|\nabla \tilde{\ell}_\gamma^*(u^k) - \nabla_u \tilde{\ell}_\gamma(u^k, W^{k+1})\| + \gamma \|\nabla \ell_{\text{trn}}^*(u^k) - \nabla \ell_{\text{trn}}(u^k, Z^{k+1})\| \\
&\leq \gamma C(Z^0)\sqrt{\frac{2\ell_{\text{trn}}(Z^0)}{\mu_w}}(1 - \beta_1 \mu_w)^{T/2} + \gamma C(W^0)\sqrt{\frac{2\ell_\gamma(W^0)}{\mu_w^\gamma}}(1 - \beta_2 \mu_w^\gamma)^{T/2} \\
&\leq \gamma \left(C(Z^0)\sqrt{\frac{2\ell_{\text{trn}}(Z^0)}{\mu_w}} + C(W^0)\sqrt{\frac{2\ell_\gamma(W^0)}{\mu_w^\gamma}}\right)(1 - \beta \mu_w)^{T/2}
\end{aligned}$$

where $\beta \mu_w = \min\{\beta_1 \mu_w, \beta_2 \mu_w^\gamma\}$.

## G.8 Proof of Theorem 3

**Proof:** We first prove the error bound condition of $\ell_\gamma^*(u)$ over the constraint $u \in \mathcal{U}$. We denote

$$c_i(u) := \|y_i^\top - x_i^\top W_\gamma^*(u)\|^2 - \|y_i\|^2 \mathbb{1}([X_{\text{trn}} X_{\text{trn}}^\dagger]_i \neq 1) \tag{96}$$

so that $\ell_\gamma^*(u) := \ell_{\text{val}}(W_\gamma^*(u)) + \frac{\gamma}{2} \sum_{i=1}^N \psi(u_i) c_i(u)$. Since $\ell_\gamma(u, \cdot)$ is uniformly PL, so the gradient of $\ell_\gamma^*(u)$ can be calculated by the Danskin type theorem as

$$\begin{aligned}
\nabla_{u_i} \ell_\gamma^*(u) &= \frac{\gamma}{2} \nabla \psi(u_i) \left[\|y_i^\top - x_i^\top W_\gamma^*(u)\|^2 - \|y_i\|^2 \mathbb{1}([X_{\text{trn}} X_{\text{trn}}^\dagger]_i \neq 1)\right] \\
&= \frac{\gamma}{2} \psi(u_i)(1 - \psi(u_i)) \left[\|y_i^\top - x_i^\top W_\gamma^*(u)\|^2 - \|y_i\|^2 \mathbb{1}([X_{\text{trn}} X_{\text{trn}}^\dagger]_i \neq 1)\right] \\
&= \frac{\gamma}{2} \psi(u_i)(1 - \psi(u_i)) c_i(u) \geq 0
\end{aligned}$$

which means $\ell_\gamma^*(u)$ is non-decreasing and (projected) gradient flow will never fluctuate, i.e. $u^{k+1} \leq u^k$ holds for any $k$. In this way, the projection operator is effective at most at the end point. If $\ell_\gamma^*(u)$ attains minimum for some $u \in \text{int}\,\mathcal{U}$, then $c_i(u) = 0$. In this case, iterates generated by projected gradient descent converge to $\min_{u \in \mathcal{U}} \ell_\gamma^*(u) = \min_u \ell_\gamma^*(u)$ because the projection operator is ineffective along the trajectory.

If $\ell_\gamma^*(u)$ attains minimum on the boundary $u \in \mathcal{U}$, we will then prove the sequence generated by projected gradient descent will still converge to $\min_{u \in \mathcal{U}} \ell_\gamma^*(u)$ by contradiction. It is clear that $\lim_{k \to \infty} \ell_\gamma^*(u^k) < \min_{u \in \mathcal{U}} \ell_\gamma^*(u)$ can not hold because $u^k \in \mathcal{U}$. Therefore, without loose of generality, we assume that $\lim_{k \to \infty} \ell_\gamma^*(u^k) > \min_{u \in \mathcal{U}} \ell_\gamma^*(u)$. Then according to the non-decreasing property of $\ell_\gamma^*(u)$, we know $\lim_{k \to \infty} u^k > -\bar{u}$ so that the projection operator is ineffective at any iteration

$K$. By smoothness and denoting $\widetilde{\nabla}\ell^*_\gamma(u^k) := \gamma\left(\nabla_u \ell_{\text{trn}}(u^k, W^{k+1}) - \nabla_u \ell_{\text{trn}}(u^k, Z^{k+1})\right), \ell^*_\gamma = \min \ell^*_\gamma(u)$, for any $k \le K$,

$$
\begin{aligned}
\ell^*_\gamma(u^{k+1}) &\le \ell^*_\gamma(u^k) + \langle \nabla \ell^*_\gamma(u^k), -\alpha \widetilde{\nabla}\ell^*_\gamma(u^k)\rangle + \frac{\alpha^2 L_u}{2}\|\widetilde{\nabla}\ell^*_\gamma(u^k)\|^2 \\
&\le \ell^*_\gamma(u^k) - \alpha\|\nabla \ell^*_\gamma(u^k)\|^2 + \frac{\alpha^2 L_u}{2}\|\nabla \ell^*_\gamma(u^k)\|^2 + \alpha\langle \nabla \ell^*_\gamma(u^k), -\widetilde{\nabla}\ell^*_\gamma(u^k) + \nabla \ell^*_\gamma(u^k)\rangle \\
&\quad + \frac{\alpha^2 L_u}{2}\langle \widetilde{\nabla}\ell^*_\gamma(u^k) + \nabla \ell^*_\gamma(u^k), \widetilde{\nabla}\ell^*_\gamma(u^k) - \nabla \ell^*_\gamma(u^k)\rangle \\
&\overset{(a)}{\le} \ell^*_\gamma(u^k) - \alpha\|\nabla \ell^*_\gamma(u^k)\|^2 + \frac{\alpha^2 L_u}{2}\|\nabla \ell^*_\gamma(u^k)\|^2 + \alpha L_u(\bar{u}+2)\|\nabla \ell^*_\gamma(u^k) - \widetilde{\nabla}\ell^*_\gamma(u^k)\| \\
&\overset{(b)}{\le} \ell^*_\gamma(u^k) - \alpha\mu_u(\ell^*_\gamma(u^k) - \ell^*_\gamma) + \alpha L_u(\bar{u}+2)\|\nabla \ell^*_\gamma(u^k) - \widetilde{\nabla}\ell^*_\gamma(u^k)\| \\
&\overset{(c)}{\le} \ell^*_\gamma(u^k) - \alpha\mu_u(\ell^*_\gamma(u^k) - \ell^*_\gamma) \\
&\quad + \alpha L_u(\bar{u}+2)\gamma\left(C(W^0)\sqrt{\frac{2\ell_\gamma(W^0)}{\mu^\gamma_w}} + C(Z^0)\sqrt{\frac{2\ell_{\text{trn}}(Z^0)}{\mu_w}}\right)(1-\beta\mu_w)^{T/2} \quad (97)
\end{aligned}
$$

where (a) comes from the Cauchy-Swartz inequality, $\|\nabla \ell^*_\gamma(u)\| \le L_u$ in Lemma 29, $\alpha L_u \le 1$ and

$$
\alpha\|\widetilde{\nabla}\ell^*_\gamma(u^k)\| = \|u^k - \text{Proj}_{\mathcal{U}}\left(u^k - \alpha\widetilde{\nabla}\ell^*_\gamma(u^k)\right)\| \le 2\bar{u}
$$

(b) results from blockwise PL condition, and (c) is derived from (90). Subtracting $\ell^*_\gamma$ from the both sides of (97), we get

$$
\begin{aligned}
\ell^*_\gamma(u^{k+1}) - \ell^*_\gamma &\le (1-\alpha\mu_u)(\ell^*_\gamma(u^k) - \ell^*_\gamma) \\
&\quad + \alpha L_u(\bar{u}+2)\gamma\left(C(W^0)\sqrt{\frac{2\ell_\gamma(W^0)}{\mu^\gamma_w}} + C(Z^0)\sqrt{\frac{2\ell_{\text{trn}}(Z^0)}{\mu_w}}\right)(1-\beta\mu_w)^{T/2}.
\end{aligned}
$$
(98)

Telescoping (98) from $k=1$ to $K$ yields

$$
\begin{aligned}
\ell^*_\gamma(u^K) - \ell^*_\gamma &\le (1-\alpha\mu_u)^K(\ell^*_\gamma(u^k) - \ell^*_\gamma) \\
&\quad + \frac{L_u(1+2\bar{u})\gamma\left(C(W^0)\sqrt{\frac{2\ell_\gamma(W^0)}{\mu^\gamma_w}} + C(Z^0)\sqrt{\frac{2\ell_{\text{trn}}(Z^0)}{\mu_w}}\right)}{\mu_l}(1-\beta\mu_w)^{T/2}. \quad (99)
\end{aligned}
$$

Taking $K \to \infty$, we know $\lim_{k\to\infty}\ell^*_\gamma(u^k) = \min \ell^*_\gamma(u) \le \min_{u\in\mathcal{U}} \ell^*_\gamma(u)$ which yields a contradiction to $\lim_{k\to\infty}\ell^*_\gamma(u^k) > \min_{u\in\mathcal{U}} \ell^*_\gamma(u)$. In conclusion, choosing $\gamma = \mathcal{O}(\epsilon^{0.5})$, and to achieve the $\epsilon$-stationary point of the penalized objective, we can set $T = \mathcal{O}(\log \epsilon^{-1})$ and $K = \mathcal{O}(\log \epsilon^{-1})$.

Besides, according to Lemma 26, the minimum norm solution $W^*_\gamma(u) = \arg\min_W \ell_\gamma(u, W)$ is bounded by $L^*_w = \mathcal{O}(1)$. Moreover, according to (Oymak & Soltanolkotabi, 2019), GD on linear regression converges to the closest minimizer to the initialization. Therefore, the iterates of PBGD satisfies

$$
\|W^k\| \le \|W^k - \text{Proj}_{\mathcal{W}^*_\gamma(u^k)}(W^0)\| + \|\text{Proj}_{\mathcal{W}^*_\gamma(u^k)}(W^0)\| \le \mathcal{O}(1)(1-\beta\mu_w)^{T/2} + L^*_w
$$

where the bound is independent of $\gamma$. Then according to Theorem 7, the $\epsilon$-stationary point of the penalized objective with $\gamma = \mathcal{O}(\epsilon^{0.5})$ recovers an $(\epsilon, \epsilon)$ optimal point of the bilevel problem. Therefore, the iteration complexity of PBGD to achieve an $(\epsilon, \epsilon)$ optimal point is $TK = \mathcal{O}((\log \epsilon^{-1})^2)$.

## H   NUMERICAL EXPERIMENTS

In this section, we provide numerical results for global convergence of PBGD in representation learning and data hyper-cleaning.

## H.1 REPRESENTATION LEARNING

Considering the overparameterized and wide neural network case, we choose $N = 30, N' = 20, m = 40, n = 10, h = 300$. First, we respectively generate data matrix $X_{\text{trn}} \in \mathbb{R}^{N \times m}, X_{\text{val}} \in \mathbb{R}^{N' \times m}$ from Gaussian distribution $\mathcal{N}(5, 0.01)$ and $\mathcal{N}(-3, 0.01)$ to model different cluster of data. Then we generate optimal $W_1^* \in \mathbb{R}^{m \times h}, W_2^* \in \mathbb{R}^{h \times n}$ from Gaussian distribution $\mathcal{N}(0, 0.01)$ and $\mathcal{N}(2, 0.01)$, respectively. Moroever, we generate the optimal weight under validation dataset $\widetilde{W}_2^* \in \mathbb{R}^{h \times n}$ by $\widetilde{W}_2^* \sim \mathcal{N}(W_2^*, 0.001)$. This ensures that $\|\widetilde{W}_2^* - W_2^*\|$ is not too large with high probability, satisfying Assumption 2. Finally, we use $W_1^*, W_2^*, \widetilde{W}_2^*$ to generate the label matrix. For the bottom layer, we want both training label and validation clean label finds the shared optimal weight $W_1^*$, However, for the adaptation layer, they should exhibit distinct behaviors due to the different clusters. Specifically, the training labels should find the optimal weight $W_2^*$, while the validation label should find the optimal validation adaptation weight $\widetilde{W}_2^*$. Inspired by this, we generate the label matrix by

$$Y_{\text{trn}} \sim \mathcal{N}(X_{\text{trn}}W_1^*W_2^*, 0.01), \quad Y_{\text{val}} \sim \mathcal{N}(X_{\text{trn}}W_1^*\widetilde{W}_2^*, 0.01).$$

We test the PBGD in Algorithm 1 on this synthetic representation learning problem and plot the upper-level and lower-level relative error versus iteration in Figure 3. We measure upper-level relative error by $\mathsf{L}_{\text{val}}(W_1, W_2) - \mathsf{L}_{\text{val}}^*$ where $\mathsf{L}_{\text{val}}^* = \min_{W_1, W_2 \in \mathcal{S}(W_1)} \mathsf{L}_{\text{val}}(W_1, W_2)$, and lower-level relative error is measured by $\mathsf{L}_{\text{trn}}(W_1, W_2) - \mathsf{L}_{\text{trn}}^*(W_1)$. By the closed form of $\mathsf{L}_{\text{trn}}^*(W_1)$ in Lemma 16, lower-level relative error is accessible. On the other hand, $\mathsf{L}_{\text{val}}^* \approx \mathsf{L}_{\text{val}}(W_1^*, W_2^*)$ because $W_2^* \in \mathcal{S}(W_1^*)$ is feasible and $\|W_2^* - \widetilde{W}_2^*\| \leq \epsilon$ so that $\mathsf{L}_{\text{val}}(W_1^*, W_2^*)$ is closed to the unconstrained minimal value $\mathsf{L}_{\text{val}}(W_1^*, \widetilde{W}_2^*)$ which is also small. As $\mathsf{L}_{\text{val}}(W_1^*, W_2^*)$ is only an estimate of $\mathsf{L}_{\text{val}}^*$, there exists cases where $\mathsf{L}_{\text{val}}(W_1, W_2) - \mathsf{L}_{\text{val}}(W_1^*, W_2^*) < 0$, so in practice, we use $|\mathsf{L}_{\text{val}}(W_1, W_2) - \mathsf{L}_{\text{val}}(W_1^*, W_2^*)|$ to estimate the upper-level relative error. Since the convergence of lower-level relative error suggests $W_2 \to W_2^*$, the convergence of upper-level relative error will indicate that $W_1 \to W_1^*$.

We select the best stepsizes $\alpha, \beta$ and the number of inner loop $T_k = T$ by grid search. It can be observed from Figure 3 that PBGD converges almost at a linear rate to a certain accuracy, and the relative error decreases as $\gamma$ increases. The fluctuation in the upper-level error near convergence is due to the global convergence result in Theorem 1 not being strictly decreasing because of the $\mathcal{O}(\epsilon)$ error at each step. When upper-level and lower-level relative errors are sufficiently small, the additional $\mathcal{O}(\epsilon)$ error has larger impact, leading to a slight increase in error. However, the final error remains small, around $10^{-5}$ when $K$ and $\gamma$ is large enough. This validates Theorem 1 that by setting $\gamma$ large enough, PBGD globally converges to a target accuracy determined by $\gamma$ at almost linear rate.

The fluctuation in the upper-level error near convergence in representation learning is due to the global convergence result in Theorem 1 not being strictly decreasing because of the $\mathcal{O}(\epsilon)$ error at each step. When upper-level and lower-level errors are sufficiently small, the additional $\mathcal{O}(\epsilon)$ error has larger impact, leading to a slight increase in error. However, the final error remains small, around $10^{-5}$ when $K$ and $\gamma$ is large as $\gamma = 10,500$. This validates Theorem 1 that by setting $\gamma$ large enough, PBGD globally converges to a target accuracy $\epsilon$ inversely determined by $\gamma$ at almost linear rate.

## H.2 DATA HYPER-CLEANING

Considering the overparameterized linear regression with a small clean validation dataset and a large dirty training dataset, we choose $N = 100, N' = 10, m = 200, n = 10$. First, we respectively generate data matrix $X_{\text{trn}} \in \mathbb{R}^{N \times m}, X_{\text{val}} \in \mathbb{R}^{N' \times m}$ from Gaussian distribution $\mathcal{N}(5, 0.01)$ and $\mathcal{N}(-3, 0.01)$ to model different cluster of data. Then we generate optimal clean weight $W^* \in \mathbb{R}^{m \times n}$ from Gaussian distribution $\mathcal{N}(1, 0.01)$ and generate the clean label matrix for validation dataset as $Y_{\text{val}} \sim \mathcal{N}(X_{\text{val}}W^*, 0.001)$. For the training label matrix, we first generate optimal classification parameters $\psi(u_i) \sim \text{Bernoulli}(0.2)$ and then generate the label matrix as

$$Y_{\text{trn}} \sim \mathcal{N}(X_{\text{trn}}W^*, 0.01) + \psi(u) \odot \mathcal{N}(10, 10)$$

where $\psi(u) = [\psi(u_1); \psi(u_2); \cdots ; \psi(u_N)]$ and $\odot$ denotes the Hadamard product. This ensures the training dataset is polluted with probability 0.2.

We run PBGD in Algorithm 2 on this synthetic data hyper-cleaning problem and plot the upper-level and lower-level relative errors versus iteration in Figure 3. We measure the lower-level relative

error by $\ell_{\mathrm{trn}}(u, W) - \ell_{\mathrm{trn}}^*(u)$ with closed form in Lemma 23 and the upper-level relative error by $\ell_{\mathrm{val}}(W) - \ell_{\mathrm{val}}^*$ where $\ell_{\mathrm{val}}^* = \min_{u, W \in \mathcal{S}(u)} \ell_{\mathrm{val}}(W)$. We estimate $\ell_{\mathrm{val}}^* \approx \min_W \ell_{\mathrm{val}}(W) = \ell_{\mathrm{val}}(W^*)$ because there exists $u$ such that the selected data matrix $\sqrt{\psi_N(u)} X_{\mathrm{trn}}$ is almost full rank so that selected training dataset and validation dataset share a joint minimizer $W^*$.

We select the best stepsizes $\alpha, \beta, \tilde{\beta}$ and the number of inner loop $T_k = T$ by grid search. It can be observed from Figure 3 that PBGD converges almost at a linear rate to a certain accuracy, and the relative error, especially at the lower level, decreases as $\gamma$ increases. The final error remains small, around $10^{-5}$ when $K$ and $\gamma$ is large enough. This coincides with our Theorem 1 that PBGD globally converges to a target accuracy inversely determined by $\gamma$ at almost linear rate. Furthermore, the lower-level relative error is more sensitive to the choice of $\gamma$, as noted in Theorem 1, where the lower-level relative error $\epsilon_\gamma$ is inversely related to $\gamma$.

