# OpenReview forum: "Unlocking Global Optimality in Bilevel Optimization: A Pilot Study"
_ICLR.cc/2025/Conference — ICLR 2025 Poster_

### Official Review · Reviewer_FJVZ · 2024-10-26

**Soundness:** 4
**Presentation:** 3
**Contribution:** 3
**Rating:** 8
**Confidence:** 3

**Summary:**

This paper studies the global convergence rate of bilevel optimization. The main result is that if the penalized objective satisfies the PL condition, then the bilevel problem have almost linear global convergence rate if PBGD method is used to solve the problem. Then the authors give two applications: representation learning and data hyper-cleaning. These problems can be formulated as bilevel optimization problems, and their penalized objectives satisfy the PL condition. Thus, when applying PBGD algorithm, they should converge almost linearly. The preliminary computational results also support the theorem.

**Strengths:**

Clear Problem Statement: The authors articulate the limitations of existing methods, particularly those that only guarantee convergence to local minima or stationary points, which motivate them for pursuing global convergence.

Timeliness and Relevance: The paper proof the global convergent rate for a certain type of bilevel optimization problems. Given the increasing application of bilevel optimization in machine learning and high-stakes fields, this work has substantial relevance.

Theoretical Contribution: The authors provide sufficient conditions for achieving global optimality. By leveraging the penalty reformulation approach, the paper establishes an almost linear global convergent rate for some linear bilevel optimization problems.

Experimental Validation: The empirical results test on bilevel learning problems like representation learning and data hyper-cleaning. The preliminary computational results support the almost linear convergence theorem.

**Weaknesses:**

Assumptions and Limitations: While the paper claims global convergence for bilevel problems, it focuses primarily on linear models. Expanding the theoretical foundation to nonlinear models or other loss functions would improve the paper’s generalizability.

Comparative Analysis: While the paper mentions other approaches, a direct empirical comparison with state-of-the-art methods for bilevel optimization would strengthen its validation.

Connection between Theory and Experiment: the author should clearly specified the connections between the theory and experiment so that the experimental results can support the theory. For example: in section 6, the author should specific the choice of the step length and make sure that they satisfied the conditions stated in Theorem 2 and 3.

**Questions:**

1.	Major Concerns:

(a) In line 261, Danskin theorem is mentioned, then the gradient is calculated. Also, the variable $\omega$ is introduced later. I think it would be better to explain the connection and point out that the using Danskin theorem, the auxiliary variable $\omega$ will help us to find a good estimation of the gradient with respect to $u$.

(b) It may be better to put Algorithm 1 and 2 on Page 6 after the authors have summary these algorithms. It will give the readers a smooth reading experience.

(c) In section 6, you may want to specific the choice of $\alpha$ and $\beta$ and make sure that they satisfied the conditions stated in Theorem 2 and 3.

(d) If possible, adding more baseline methods would help readers better understand the convergence rate of the PBGD method. This is not necessary given the limited time.

2.	Minor Concerns:

(a) The sentence in line 199 is not very clear, please double check.

(b) There’s a “?” in line 309, please make sure it is correct.

(c) Misspelling in Line 973 and Line 2189. “invertiable” to “invertible”.

---

### Official Review · Reviewer_4hB6 · 2024-11-01

**Soundness:** 3
**Presentation:** 3
**Contribution:** 3
**Rating:** 6
**Confidence:** 3

**Summary:**

This paper propose a new bilevel optimization algorithm. This paper is generally very well written and provide plenty of theoretical results. Overall this paper is clear a good paper. If all these results are correct, this paper should be clearly accepted (However, I am inadequate to go through all proofs).

**Strengths:**

good. This paper is overall well written and provide plenty of theoretical results.

The proposed method also solves the neural network cases. That's especially good.

**Weaknesses:**

1. Experiments are not adequate.

2. Some fonts seem strange.

**Questions:**

1. What does the a pilot mean in the title?

2. Line 057, a benign landscape, is there a direct meaning for that?

3. Line 53, the  goal of this paper, this sentence is not important. Do not need to emp{}.

4. The numerical results seem too little? Does the proposed method outperform SOTA bi-level methods?

5. What are the best convergence results for bi-level optimization method before this paper?

6. line 414, what does \gamma to be O(\epsilon^{-0.5}) mean? If gamma is very very large (with a very large constant), can the algorithm still converge? What is the meaning of O(xx) here?

7. Will PL condition a bit too strong?

**Details Of Ethics Concerns:**

NA.

---

### Official Review · Reviewer_5tcw · 2024-11-02

**Soundness:** 4
**Presentation:** 3
**Contribution:** 2
**Rating:** 3
**Confidence:** 2

**Summary:**

The paper proposed two PL conditions, by satisfying which the global optimality of bi-level optimization can be achieved using simple algorithms like Gauss-Seidel.

**Strengths:**

Achieving Global optimality is an important property.

**Weaknesses:**

The paper's assumptions are very restrictive. For most bilevel optimization problems, the Joint and blockwise PL conditions cannot be guaranteed, and even checking these conditions can be challenging.  The representative problems illustrated in the paper are very specific simple cases. For example, only linear models can satisfy the assumption for representation learning.

**Questions:**

Can it be applied to a more general bi-level optimization with constraints in (1)?

---

> ### Author Response · Authors · 2024-11-27
>
> We sincerely thank the reviewer for the time and thoughtful review of our submission. We hope our response has addressed your concerns, and we welcome any further comments or questions you may have.

---

### Official Review · Reviewer_Dv2u · 2024-11-03

**Soundness:** 3
**Presentation:** 2
**Contribution:** 3
**Rating:** 6
**Confidence:** 3

**Summary:**

The paper explores global convergence in bilevel optimization, a crucial yet challenging objective due to the non-convexity and potential for multiple local solutions in bilevel problems. To address this, the authors propose sufficient conditions for global convergence and illustrate these in bilevel learning applications such as representation learning and data hyper-cleaning.

**Strengths:**

The paper offers conditions that ensure global convergence in bilevel optimization by generalizing the Polyak-Lojasiewicz (PL) condition.

**Weaknesses:**

While global optimality is underscored as essential, the precise definition or context of “global optimality” within this framework is unclear. A clear explanation of how this term is specifically applied in their method would strengthen the paper.

**Questions:**

1. Could the authors expand Section 1.1 with detailed theorems? The sentence following C3, “The joint and blockwise PL condition… are not assumptions, but the properties of the penalty reformulation,” is confusing. The authors should clarify the assumptions needed to establish global convergence rigorously.

2. In what specific way is “global optimality” used in the paper?

---

### Official Review · Reviewer_Bn7x · 2024-11-04

**Soundness:** 3
**Presentation:** 3
**Contribution:** 2
**Rating:** 5
**Confidence:** 3

**Summary:**

This paper presents a theoretical framework for achieving global convergence in bilevel optimization. The authors propose that a constrained reformulation generally yields a benign landscape, and they analyze the global convergence of penalized bilevel gradient descent (PBGD) algorithm for bilevel objectives under the proposed joint and blockwise PL conditions. The paper illustrates that the specific applications of representation learning and data hyper-cleaning can satisfy these PL conditions. Theoretical results are then supported by experiments conducted on these applications.

**Strengths:**

The main strength is that it is a pioneering work that studies the challenging and important problem of global convergence in bilevel optimization, a topic with substantial real-world relevance. The proposed analysis extends PL to both joint and blockwise PL conditions and verifies them on two application cases. Overall, the paper is well-organized and easy to follow.

**Weaknesses:**

I have several concerns and comments on the submission (please correct me if I am wrong):

1. The applicability of the developed theorem seems unclear. The proof closely dependent on and follow existing convergence theorems for PBGD, and it’s unclear whether the analysis could extend to other bilevel algorithms. The non-additivity of PL conditions poses a great challenge for applying the developed theorem and no practical solutions are provided. The two applications studied rely on linear models and strong convexity of loss, which is overly idealized and simplified.

1. Moreover,  in line 228 (Section 3), the authors mention that convergence analysis may need “fine-tuning per application,” but it remains unclear which parts of the analysis are generally hold, such as whether the iteration complexity $𝑂(log⁡(𝜖^{−1}))$ generally holds to other settings that satisfy PL conditions. It also mention that "This may also shed light on a broader range of bilevel problems involving sophisticated neural network architectures in machine learning", but the paper lacks clearly summaries practical takeaways got from the developed theorem for achieving global convergence in such complex applications with modern non-linear deep models.

1. The numerical analysis lacks depth and discussion on robustness. I am suggesting throughly evaluating how values of parameters $\alpha$, $\beta$, $\gamma$ are set theoretically as well as practically, and whether the observed results match theoretical expectations on the convergence rate. Also, exploring how slight violations of PL conditions affect convergence would help clarify the robustness.

1. Section 2 provides an example to illustrate the complexity of the nested objective $𝐹(𝑢)$ caused by the lower-level mapping $𝑆(𝑢)$, but it lacks rigorous analysis of how the constrained formulation reliably produces a more benign landscape and to what extend. A precise definition of benign landscape in the context of bilevel optimization is also helpful. The conclusion that constrained reformulation yields a benign landscape relies heavily on prior literature (lines 211-215) rather than in-depth analysis in this paper.

1. In line 373 (page 7), matrix $𝑊_3$ is introduced without a clear explanation.

**Questions:**

1. How should one choose between joint and blockwise PL conditions for a given application?
1. Could you please clarify which aspects of the convergence results would generalize to more complex settings like non-linear models?
1. What practical takeaways does this work provide for achieving global convergence in more complex bilevel applications?
1. How robust are the convergence results if the PL conditions are only approximately met?

---

> ### Author Response · Authors · 2024-11-27
>
> We sincerely thank the reviewer for taking the time to review our submission.
>
> To further address your concern, **Q5: Definition of benign landscape and the reason why penalty landscape is easier to yield a benign landscape**, we have provided additional visualizations based on the updated Example 1 in the revision.
>
> These visualizations illustrate how the lifted dimensionality helps the penalty function achieve a better landscape compared with the nested objective. We believe this is a critical insight that may worth future development. Hope this further clarifies your concerns, and we welcome any additional comments or questions you may have.

---

> ### Comment · Reviewer_Bn7x · 2024-11-29
>
> I appreciate the authors' efforts in addressing most of my concerns regarding the soundness of the theorem.
>
> Regarding W3, I remain unconvinced about how the hyperparameters were chosen in alignment with the theorem's development and how the robustness of the theoretical claims would hold under varying parameter values.
>
> Morever, while I acknowledge that I am not an expert in the theoretical aspects, I feel that it remains unclear how the results of this paper can be effectively leveraged for general practical applications in bilevel optimization. Could the authors clarify the practical value of the theorem in the context of bilevel optimization algorithm design and the connection of the theorem to practical insights or implementation suggestions? I believe a key impact of developing theorem analysis is to uncover insights that can effectively guide practical applications.

---

> ### Author Response · Authors · 2024-12-01
> **Response to Reviewer Bn7x**
>
> We thank the reviewer for the insightful feedback and we are glad that our response has solved most of your concerns. Below, we address your remaining concerns.
>
> ### **1. Hyperparameter Selection and Alignment with Theory**
>
> Thanks for your question. We demonstrate alignment between theory and practice in the following ways:
> - **Almost Linear Convergence**: PBGD, with appropriately chosen stepsizes, converges almost linearly to the optimal solution in representation learning and data hyper-cleaning, as shown in Figure 3. In this figure, the log of the optimality gap is plotted on the y-axis, and the resulting linear trend with respect to $K$ indicates the log of the optimality gap decreases linearly with $K$, confirming the linear convergence rate shown in Theorems 2–3.
> - **Impact of penalty constant $\gamma$**: Enlarging the penalty constant $\gamma$ reduces the target optimality gap $\epsilon$ in Figure 3, as predicted by the inverse proportionality established in Theorems 2–3. Besides, by choosing varying $\gamma$ from $0.1-500$ for representation learning ($50-100$ for data hyper-cleaning), PBGD all converges to the global optimum with error less than $10^{-3}$, suggesting the robustness of our results in terms of $\gamma$.
> - **Impact of stepsizes $\alpha$ and $\beta$**: While the theoretical upper bounds for stepsizes $\alpha,\beta$ provided in this paper primarily serve to demonstrate the existence of thresholds for convergence and may not be tight, they offer valuable theoretical guidance. In practice, although we cannot verify exact alignment with these bounds, we empirically identify stepsizes that ensure convergence, which also supports the existence of such thresholds in Figure 4. Specifically, stepsizes exceeding these thresholds (e.g. $\alpha=5e^{-10},\beta=1e^{-9}$) cause divergence, while those within the range maintain stability and convergence. Moroever, by choosing varying $\alpha$ and $\beta$ from $1e^{-10}-1e^{-12}$ and $5e^{-10}-1e^{-10}$ respectively, PBGD all converges to the global optimum with error less than $10^{-3}$, suggesting the robustness of our results in terms of $\alpha,\beta$.
> - **Impact of ratio of stepsizes $\alpha/\beta$**: Besides, the empirical threshold for $\alpha=5e^{-10}$ is smaller than $\beta=1e^{-9}$ when $\gamma=10$ for representation learning, which also aligns with the theoretical finding that the threshold $\frac{\alpha}{\beta}={\cal O}(1/\gamma)$ in terms of $\gamma$.

---

> ### Author Response · Authors · 2024-12-01
> **Response to Reviewer Bn7x (con't)**
>
> ### **2. Practical Value of the Theorems**
>
> Thank you for your question. Our work offers actionable insights for bilevel optimization with global convergence guarantee.
> - **Choose the level of Overparameterization**: Our analysis highlights that overparameterization is critical for ensuring a benign optimization landscape and achieving global convergence; see the model requirement in our paper (Line 373 for representation learning and Line 454-455 for data hyper-cleaning, and empirical verification in Line 2810 and Line 2850). This insight is particularly relevant for applications such as hyperparameter optimization, meta-learning, and adversarial robustness, where overparameterized models are common.
> - **Choose the type of bilevel algorithms**: Previous approaches to global convergence in bilevel optimization often relied on application-specific algorithmic designs, such as semidefinite relaxations, which limit their applicability to general scenarios (see **General Response G2 2)** for more details). In contrast, our work is the first to establish global convergence for PBGD—a generally applicable bilevel method with stationary convergence guarantees across a broad range of settings. Rather than requiring application-specific modifications on algorithm, our approach only requires theoretical proof adaptations to achieve global convergence, allowing practitioners to confidently apply PBGD and other first-order bilevel algorithms without the risk of getting stuck in local minima or saddle points. Besides, better landscape of penalty reformulation also suggest that using penalty-based gradient descent approaches may help converge to a better point than implicit-gradient-based approaches.
> - **Choose the update for different problem structure**: We propose two tailored update styles to address different bilevel coupling structures: Jacobi-style updates for jointly coupled variables and Gauss-Seidel-style updates for blockwise coupled variables. For problems where the upper- and lower-level variables share a common nature like representation learning, joint updates (Jacobi-style) are more beneficial. Conversely, when the upper- and lower-level variables serve distinct purposes like data hyper-cleaning, sequential updates (Gauss-Seidel-style) are more effective. These styles offer practical guidelines for selecting the most suitable update strategy based on problem characteristics. For instance, our study can be generalized to suggest that Gauss-Seidel-style updates are better suited for tasks like hyperparameter optimization, while Jacobi-style updates align well with meta-learning problem.
>
> By bridging theory and practice, we believe these points offer guidance for practitioners while staying grounded in our theoretical framework. We are happy to further elaborate on any of these points to better highlight the connection between theory and practice.

---

### Official Review · Reviewer_GMWV · 2024-11-04

**Soundness:** 3
**Presentation:** 3
**Contribution:** 2
**Rating:** 6
**Confidence:** 4

**Summary:**

This paper studies the convergence properties of a penalized bilevel gradient descent (PBGD) algorithm, aiming to obtain global optimal solutions of bilevel optimization problems under the joint and blockwise Polyak-Łojasiewicz (PL) conditions. The joint and blockwise PL conditions are validated in the context of two specific applications: representation learning and data hyper-cleaning. Numerical experiments are provided to substantiate the theoretical results.

**Strengths:**

1.	The study of convergence of bilevel algorithms to global solutions is an interesting topic, and this paper offers an approach.
2.	The paper includes concrete application examples that validate the assumptions necessary for establishing global convergence results.

**Weaknesses:**

1.	While the topic of global optimal convergence in bilevel optimization is engaging, the approach presented in this work does not appear as innovative as suggested by the title. The main idea relies on the joint/blockwise PL condition of the penalized objective $L_\gamma$. However, it is well known that when the PL condition holds, any stationary point is globally optimal, and the proximal-gradient method can achieve linear convergence to this global optimum (see, e.g., Hamed Karimi, Julie Nutini, and Mark Schmidt, Linear Convergence of Gradient and Proximal-Gradient Methods under the Polyak-Łojasiewicz Condition, ECML PKDD 2016). Furthermore, the convergence of PBGD to a stationary point of $L_\gamma$ under the PL condition has been well studied in existing literature (e.g., Bo Liu, Mao Ye, Stephen Wright, Peter Stone, BOME! Bilevel Optimization Made Easy: A Simple First-Order Approach, NeurIPS 2022, and Shen, Han, and Tianyi Chen, On Penalty-Based Bilevel Gradient Descent Method, ICML 2023). Thus, the approach in this work may lack novelty, and the contribution seems somewhat incremental.
2.	Although the authors have put considerable effort into verifying that the joint/blockwise PL condition can be satisfied in specific applications, such as representation learning and data hyper-cleaning, only very restricted cases are analyzed, with strong assumptions imposed. For instance, Assumption 2 in the representation learning setting and the assumption $X_{trn}X_{trn}^{\dagger}$ is a diagonal matrix in data hyper-cleaning narrow the applicability of the results and limit their general applicability. The theoretical analysis appears heavily dependent on these assumptions, raising doubts about whether the joint/blockwise PL condition would hold in broader or more practical cases, or even in other bilevel optimization applications.

**Questions:**

In Line 1226, why is the blockwise PL condition of $L_\gamma$ over $u$ sufficient to ensure the PL condition for $L^*_\gamma$?

---

### Meta-Review · Area_Chair_CghV · 2024-12-17

**Metareview:**

While the reviewers have expressed concerns about some of the applicability of the assumptions and the value of the numerical studies, the authors have substantively resolved most of these concerns during the rebuttal period, and the least positive reviewers have failed to comment on whether their concerns have been resolved. After reviewing the paper personally, I believe there are fundamental contributions to the field of bilevel optimization that are worthy of publication at this time.

**Additional Comments On Reviewer Discussion:**

See above.

---

### Decision · Program_Chairs · 2025-01-22

Accept (Poster)